# Tracking momentary fluctuations in human attention with a cognitive brain-machine interface

Abhijit M. Chinchani [1,4,8], Siddharth Paliwal [1,5,8], Suhas Ganesh[1,6], Vishnu Chandrasekhar[1,7], Byron M. Yu[2] & Devarajan Sridharan [1,3 ✉]

Selective attention produces systematic effects on neural states. It is unclear whether, conversely, momentary fluctuations in neural states have behavioral significance for attention. We investigated this question in the human brain with a cognitive brain-machine interface (cBMI) for tracking electrophysiological steady-state visually evoked potentials (SSVEPs) in real-time. Discrimination accuracy (d′) was significantly higher when target stimuli were triggered at high, versus low, SSVEP power states. Target and distractor SSVEP power was uncorrelated across the hemifields, and target d′ was unaffected by distractor SSVEP power states. Next, we trained participants on an auditory neurofeedback paradigm to generate biased, cross-hemispheric competitive interactions between target and distractor SSVEPs. The strongest behavioral effects emerged when competitive SSVEP dynamics unfolded at a timescale corresponding to the deployment of endogenous attention. In sum, SSVEP power dynamics provide a reliable readout of attentional state, a result with critical implications for tracking and training human attention.

---

[1] Centre for Neuroscience, Indian Institute of Science, Bangalore, KA, India. [2] Department of Biomedical Engineering, and Department of Electrical & Computer Engineering, Carnegie Mellon University, Pittsburgh, PA, USA. [3] Computer Science and Automation, Indian Institute of Science, Bangalore, KA, India. [4] Present address: University of British Columbia, 2329 West Mall, Vancouver, BC, Canada. [5] Present address: Stony Brook University, 100 Nicolls Rd, Stony Brook, NY, USA. [6] Present address: Verily Life Sciences, 269 E Grand Ave, South San Francisco, CA, USA. [7] Present address: Carnegie Mellon University, 319 Morewood Avenue, Pittsburgh, PA, USA. [8] These authors contributed equally: Abhijit M. Chinchani, Siddharth Paliwal. ✉email: sridhar@iisc.ac.in

Selective attention is an essential cognitive capacity that allows us to select and prioritize task-relevant information in the world around us. The effects of attention manifest in behavior as both an increase in perceptual accuracies and faster reaction times for task-relevant stimuli[1]. In the brain, attention alters various measures of neural information processing for attended targets, both at the single neuron level and at the population level[2–4]. For example, visuospatial attention induces modulations of various neuronal metrics, including an increase in firing rate, a decrease in firing rate Fano factor, or a decrease in noise correlation across the population, in the visual cortex[5–7].

While signature changes in the neural state are known to occur during attention, here, we asked the converse question: Do momentary fluctuations in neural signatures have behavioral relevance for attention? Specifically, are such changes of neural state sufficient indicators of changes of attention state? A few studies investigating the non-human primate cortex have addressed this question, and have shown that momentary fluctuations in attention correlate with neural activity states in primate visual areas[8–10]. For example, instantaneous metrics of attentional state were estimated from neuronal population activity in the visual cortex (e.g., projection along attention axis[8]), and fluctuations in these metrics correlated with animals' perceptual accuracy on a trial-by-trial basis.

Here, we investigated the link between momentary fluctuations in neural state and attention, in human participants. To investigate this link at millisecond timescales in the human brain, we employed electroencephalography (EEG). Investigations of human EEG signatures of visual attention fall into three broad categories, depending on the type of neural process being studied. First, the amplitude of alpha-band (8–14 Hz) oscillations is known to consistently modulate with attention: Attention to one visual hemifield produces a robust decrease in alpha oscillation power over the contralateral, relative to the ipsilateral, brain hemisphere[11,12]. Second, consistent effects of attention are observed on the amplitude of event-related potentials (ERPs), such as the N200 or the P300[13,14]. Third, attention also consistently affects steady-state visually evoked potentials (SSVEPs): oscillatory EEG potentials evoked by rhythmically flickering visual stimuli. Since the 1990s, an extensive literature has documented enhanced SSVEP power for stimuli in the attended hemifield[15–17]. In contrast to first two EEG signals (alpha oscillations and ERPs), SSVEPs have the unique advantage of being reliably evoked with high signal-to-noise ratio (SNR) from the occipital lobe and are persistent for the duration of the flickering visual stimulus[18]. SSVEPs, therefore, provided a potential neural marker for tracking momentary fluctuations in attention state.

While previous studies have extensively examined the relationship between SSVEP power and behavior offline, with post hoc analyses[15,17], such a correlational approach merely measures associations, and does not permit inferring direct relationships between neural states and behavior. In contrast, here, we adopt a more rigorous, interventional approach that enables inferring direct (mechanistic) relationships between neural signatures and behavior[19] (see "Discussion"). Specifically, we investigated the link between SSVEP states and behavior with an interventional cognitive brain–machine interface. Unlike typical brain–machine interfaces (BMIs), widely known for their potential to help disabled individuals[20–22], a cognitive brain–machine interface (cBMI) provides closed-loop neurofeedback based on neural signatures of cognitive processes, like attention[23–28] (see "Discussion"). Specifically, our cBMI tracked momentary fluctuations in SSVEP power in real-time, with millisecond precision (Supplementary Fig. 1a, b) and enabled performing temporally precise interventions, based on specific SSVEP power states.

With this cBMI platform, we addressed two key questions. First, we asked if fluctuations in SSVEP power would provide a reliable (sufficient) marker for subjects' attention. To answer this question, we tracked SSVEP power in real-time, and triggered visual stimuli (target and distractor) when SSVEPs reached high, versus low, power states (Fig. 1a). We then compared differences in behavioral performance (accuracies and reaction times) across these high and low SSVEP power states. Second, we trained participants to generate biased, cross-hemispheric competitive interactions in SSVEP representations across the visual field, using auditory neurofeedback. We asked if such a manipulation would produce changes in attention state, in real-time.

Our experiments on SSVEP power states, demonstrate that a change in neural state (SSVEP power) is sufficient to achieve a change in a specific behavioral state (accuracy, but not reaction time). We propose our cBMI platform as a useful tool for real-time tracking, and training, of human visuospatial attention.

## Results

**Tracking momentary fluctuations of SSVEP power with an interventional cBMI.** Two cohorts (A and B), comprising $n = 24$ subjects, performed these experiments. Results from cohort A are described in this and the next section, and those from cohort B in a later section. Details regarding the cognitive brain–machine interface (cBMI) are described in Methods.

Subjects ($n = 15$; cohort A) performed a cued orientation discrimination task, while we tracked their EEG-SSVEP power in real-time. Each trial began with a positive contrast fixation cross that was presented in the center of a gray screen (Fig. 1b; see "Methods" for detailed experimental procedures). Next, two pedestals appeared on each side of the fixation cross. Each pedestal was a plaid produced by superimposing square wave gratings oriented at +45° and −45° from the vertical. Either one (6/15 subjects) or both (9/15 subjects) pedestals flickered at a distinct frequency (Supplementary Table 1) to evoke SSVEPs at the corresponding frequency. After 1000 ms, a directed cue (central arrow) appeared, indicating the side to be attended. The cue indicated the side corresponding to the target stimulus with 100% validity (pseudorandom, and counterbalanced across hemifields). After a variable interval (determined by SSVEP power dynamics, see next), the pedestals disappeared and two stimuli—sinusoidal gratings oriented at either +45° or −45°— appeared briefly (75 ms) on either side of the fixation cross, concentric with the pedestals. Following this, the fixation cross and the cue changed in color. Subjects reported the orientation of the grating on the cued side (target) as being clockwise or counterclockwise of vertical, ignoring the grating on the uncued side (distractor). Subjects' scalp EEG was concurrently recorded from 41 electrodes over occipital cortex (Supplementary Fig. 1c; Methods). For triggering target and distractor gratings based on EEG SSVEP power, we employed the following two-step procedure.

First, we isolated and quantified SSVEP power from the underlying, noisy EEG data with the denoising source separation (DSS) technique[29] ("Methods"). Briefly, DSS identifies low-dimensional latent dimensions from high-dimensional, noisy sensor data. Each DSS latent dimension (Y) represents a linear combination of the raw EEG signals (X) from the occipital electrodes ($Y = XW_f$), where $W_f$ are the electrode weights for flicker frequency f. $W_f$ for each flicker frequency for an individual subject, and the associated spectra, are shown in Fig. 1c (population average, Supplementary Fig. 1c, d, "Methods"). Next, to estimate these dimensions without over-fitting and circularity, we conducted a baseline block before the actual experiment. We estimated SSVEP power in moving windows of 0.5 s, generated a

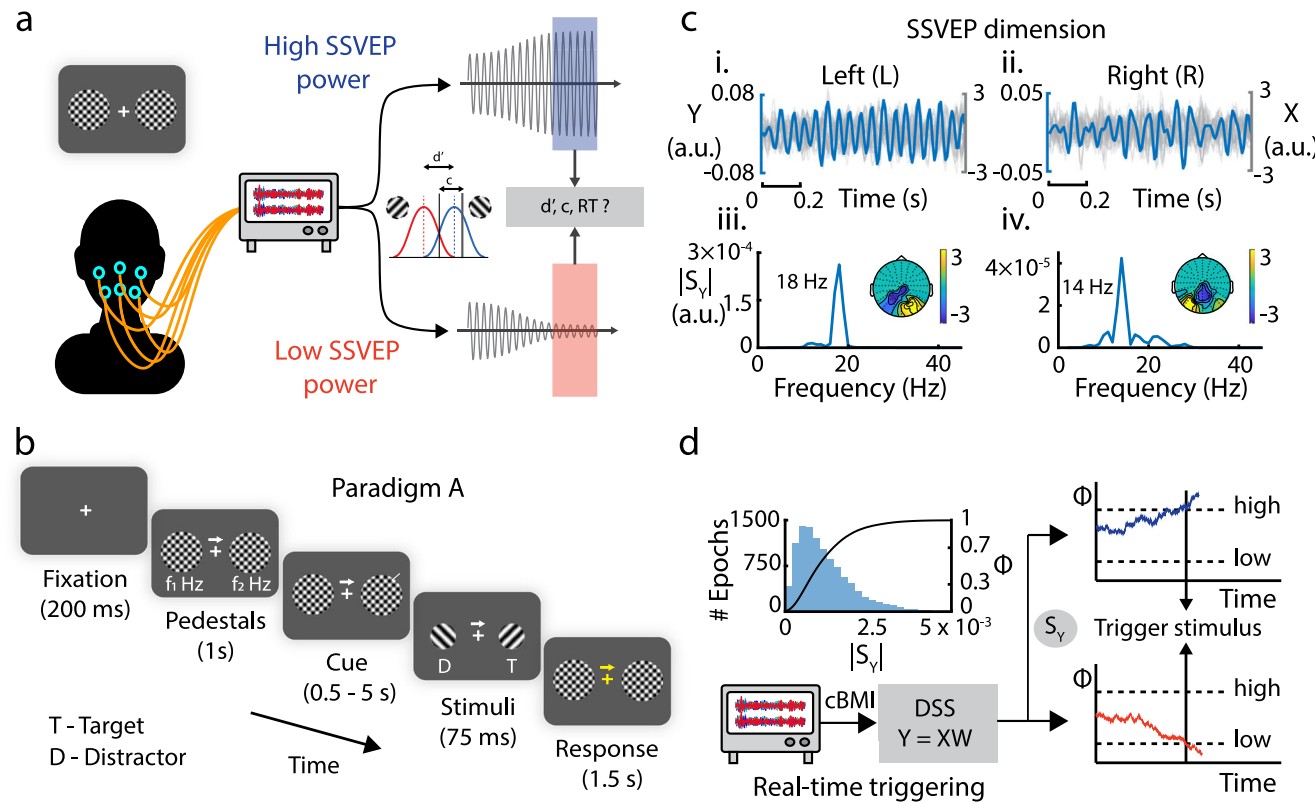

**Fig. 1 Tracking momentary fluctuations in SSVEP power with a real-time cBMI. a** Schematic depicting the cognitive brain–machine interface (cBMI) for investigating the effect of momentary fluctuations in EEG steady-state visually evoked potential (SSVEP) power on behavior (see text for details). Blue and red boxes: Momentary, high and low SSVEP power states, respectively. **b** Behavioral attention task (Paradigm A). Following an initial fixation epoch, two pedestals (plaids) flickering at distinct frequencies (e.g., f1 Hz and f2 Hz) appeared, one in each visual hemifield. After 1000 ms, a spatial cue (central arrow, 100% validity) indicated the side to be attended. After a variable interval (determined by SSVEP power states, see next), the pedestals disappeared and two stimuli—sinusoidal gratings oriented at either +45° or −45°—appeared briefly (75 ms) in their place. Subjects reported the tilt of the grating on the cued side (target) while ignoring the grating on the uncued side (distractor). **c** SSVEP dimensions extracted using denoising source separation (DSS) for an exemplar subject. (Left, i and iii) SSVEP dimension for the flickering pedestal in the left hemifield (14 Hz). (i) Mean (blue solid line) DSS time-series (Y) across epochs, in the SSVEP dimension. Gray lines: Preprocessed EEG signal (X) for a random epoch. (iii) Spectrum ($|S_Y|$) of the DSS time-series. (iii, Inset) Topographic plot of the SSVEP spatial dimension (W). (Right, ii and iv) Same as in the left column, but for the flickering pedestal in the right hemifield (18 Hz). **d** Schematic of real-time stimulus triggering protocol. (Top left) Blue histogram (left y-axis): Baseline distribution of SSVEP power ($|S_Y|$) for a representative subject for a flicker frequency of 18 Hz. Black line (right y-axis): CDF of SSVEP power or SSVEP power index (Φ). (Bottom) The recorded EEG data (X) were preprocessed and projected, in real-time, onto the SSVEP DSS dimension (W), estimated in a baseline session. The projected data (Y) was used to estimate Φ in real-time. Target and distractor gratings were presented when Φ crossed either a pre-set high threshold (high-Φ trials, top, solid blue line) or a low threshold (low-Φ trials, bottom, solid red line). Dashed, horizontal lines: high and low thresholds. Arrows: time of stimulus presentation.

baseline distribution of SSVEP for each flicker frequency (Fig. 1d, upper-left), and computed its cumulative distribution function (CDF) ("Methods"). The CDF value—which we call a normalized SSVEP power index or Φ—varies between 0.0 and 1.0 (Fig. 1d, upper-left) and provides a normalized measure of SSVEP power at each instant. Φ represents a normalized measure of the strength of SSVEP power modulation that accounts for variations in baseline SSVEP power across individual participants.

In the cBMI attention experiment, the SSVEP power index evoked by the pedestal on the cued (target) or uncued (distractor) side was computed and tracked, for each trial, in real-time (closed-loop delays: mean +/− std: 21.1 +/− 5.9 ms, Supplementary Fig. 1e; SSVEP power computed in a 500 ms window). When Φ for the respective pedestal reached a predetermined high or low threshold value (interleaved trials), the presentation of the target and distractor gratings was triggered simultaneously (Fig. 1d, right); we call these high-Φ and low-Φ trials, respectively. If Φ did not reach either the high or low thresholds within a maximum duration (4 s after cue onset), the grating stimuli were presented,

regardless of the value of Φ; these non-triggered trials (31.5 ± 3.7% of trials, mean ± s.e.m., across $n = 15$ subjects) were rejected from subsequent behavioral analyses.

Such an interventional cBMI provides two distinct advantages over correlational post hoc analysis approaches. First, the interventional cBMI enables decoupling and isolating SSVEP effects on behavioral metrics—accuracy (d′) and reaction times (RT)—from those of other neural processes. We examine 4 directed, graphical models, each of which depicts a distinct mechanism for the influence of SSVEP power (or Φ) on the behavioral metrics of d′ and RT (Supplementary Fig. 2). If a change in Φ were sufficient to induce a change in a behavioral metric (d′ or RT), we would expect to necessarily observe a change in the respective behavioral metric across high-Φ and low-Φ trials. The specific behavioral metric affected, then, enables us to distinguish between the 4 models; further details are provided in Methods (section on "Modeling interventional versus correlational approaches"; Supplementary Fig. 2). Second, the interventional cBMI enables targeted data collection for

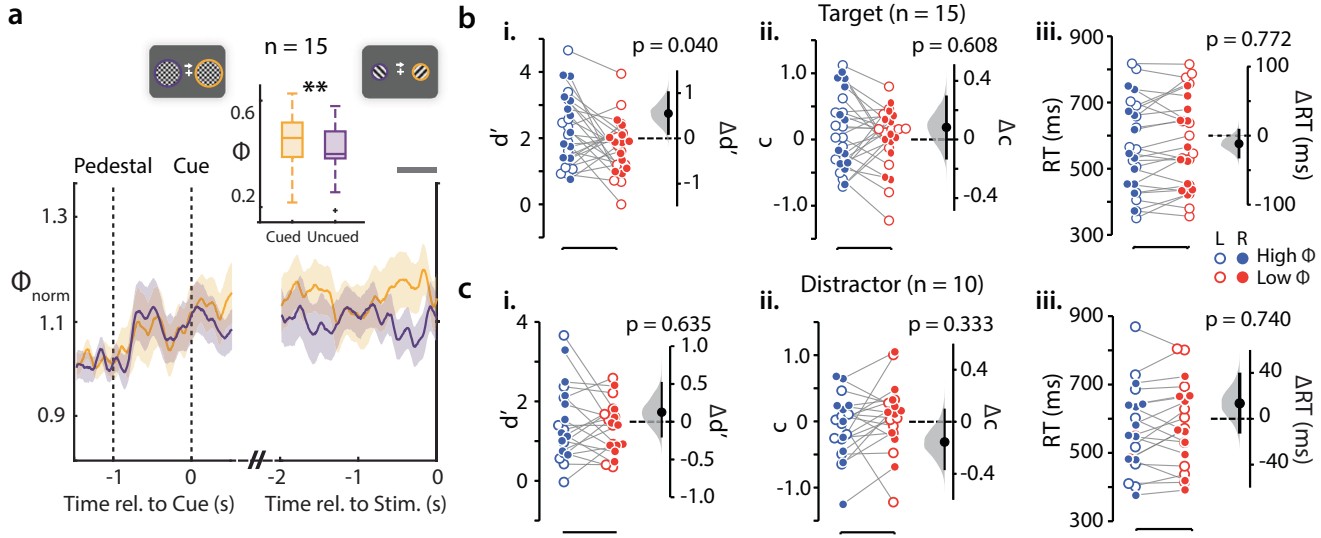

**Fig. 2 Momentary SSVEP power fluctuations predict target discrimination accuracy (d′). a** Time evolution of $\Phi_{norm}$ (fractional change in $\Phi$ over baseline) for cued (yellow) and uncued (purple) trials, pooled across all subjects ($n = 15$) and frequencies, time locked to cue onset (left half) or to grating stimulus presentation (right half) on non-triggered trials (see text for details). Shaded region: s.e.m. across subjects. (Inset) Mean $\Phi$ for cued (yellow) and uncued (purple) trials in the interval −500 ms–0 ms before stimulus onset (dark gray horizontal bar). Central mark: median, edges: 25th and 75th percentiles, the whiskers: most extreme points, +: outliers. \*\*$p < 0.01$. **b** (i) Discrimination accuracy (d′) for high-$\Phi$ (blue) and low-$\Phi$ (red) trials for the target-side triggered trials, in which grating presentation was triggered based on the cued (target) side SSVEP power index ($\Phi$) for all ($n = 15$) subjects. Open and filled circles: Trials triggered based on left- and right-hemifield $\Phi$ values, respectively. Black dot and error bar: Mean effect size and 95% confidence interval, respectively, for the difference between d′ across high-$\Phi$ and low-$\Phi$ trials. Gray histogram: bootstrap sampling distribution for effect size. Dashed horizontal line: Datum indicating zero effect size. (ii) Same as panel i, but for choice criterion (c). (iii) Same as panel i, but for reaction times. **c** Same as in (**b**), but comparing discrimination accuracy (d′, i), choice criterion (c, ii), and reaction time (iii) across high-$\Phi$ and low-$\Phi$ trials for the distractor-side triggered trials, in which grating presentation was triggered based on the uncued (distractor) side SSVEP power index ($\Phi$). Other conventions are the same as in panel **b**.

addressing our question of interest viz., the relationship between SSVEP power states and behavioral metrics. We tested if the wide differences in SSVEP power states we obtained in our experiments could have been achieved with offline (post hoc) analyses, without such a real-time cBMI. We estimate that in the latter case (post hoc analyses) we would have required several orders of magnitude more trials than we conducted (~11x trials for paradigm A and ~69x trials for paradigm B) (see Methods, section on "Online stimulus triggering versus post hoc analyses"; also see Peixoto et al.[30]).

In sum, we developed an EEG-based cognitive brain–machine interface (cBMI) that tracked SSVEP power levels in real-time with a closed-loop delay of tens of milliseconds. Target (and distractor) presentation was triggered simultaneously based on SSVEP power levels of either one of the two stimuli, in real-time. The cBMI enabled us to measure participants' behavioral performance when SSVEP oscillations were in divergent states of high or low power, at the target or distractor location. With this cBMI, we tested for putative links between momentary changes in neural states and behavior.

**Momentary fluctuations of SSVEP power predict target discrimination performance.** As a first step, we sought to recapitulate a ubiquitous finding: cueing of attention is widely known to induce a gain in SSVEP power (e.g., Morgan et al.[16]). Indeed, post-cue SSVEP power was higher at the cued location, compared to that at the uncued location (Fig. 2a, cued $\Phi = 0.48 \pm 0.09$, uncued $\Phi = 0.42 \pm 0.07$, median ± sd, $p = 0.006$, Wilcoxon signed-rank test, non-triggered trials) (see Methods section on "Analysis of cueing-induced gain of SSVEP power"); the difference was not significant in the pre-cue epoch (cued $\Phi = 0.44 \pm 0.08$, uncued $\Phi = 0.44 \pm 0.07$, $p = 0.967$). These results

provided electrophysiological evidence indicating that subjects were paying attention to the cued side during this task. We quantified, next, behavioral performance metrics for discriminating orientations at the target location (Fig. 1a, inset), for trials triggered based on the (cued) target's SSVEP power index.

Discrimination accuracy (d′) was significantly higher for the high-$\Phi$ compared to the low-$\Phi$ trials (Fig. 2bi; d′: ANOVA, main effect, high-$\Phi$ versus low-$\Phi$, $F(1, 48) = 4.47$, $p = 0.040$; Bayes Factor (BF) = 3.36; $n = 15$ subjects; see Methods section on "Statistics and reproducibility"). On the other hand, there was no significant difference in criteria across high-$\Phi$ and low-$\Phi$ trial types (Fig. 2bii; criterion: ANOVA, main effect, high-$\Phi$ versus low-$\Phi$, $F(1, 48) = 0.27$, $p = 0.608$; BF = 0.20). Moreover, we observed no evidence for or against significant differences in reaction times across high-$\Phi$ and low-$\Phi$ trials (Fig. 2biii; RT: ANOVA, main effect, high-$\Phi$ versus low-$\Phi$, $F(1, 48) = 0.08$, $p = 0.772$; BF = 0.96). Median cue-target intervals were not significantly different between high-$\Phi$ and low-$\Phi$ trials (Supplementary Fig. 3a, left; CTI: high-$\Phi$ = 1840 [1007 2132] ms, median [95% CI], low-$\Phi$ = 1573 [949 2208] ms, $p = 0.252$; Wilcoxon signed-rank test), suggesting that these behavioral differences were not due to systematic differences in cue-target intervals across the two trial types.

Next, we asked if the higher discrimination accuracy for high-$\Phi$ trials was due to a spatially selective modulation of SSVEP power at the target location. An alternative possibility is that global modulations of SSVEP power at both the target and distractor locations, possibly due to alertness (or arousal), were responsible for these effects.

We tested these hypotheses directly with additional experiments. In a subset of participants ($n = 10$), we also triggered stimulus presentation when the distractor SSVEP power index reached the same high or low thresholds; these distractor SSVEP

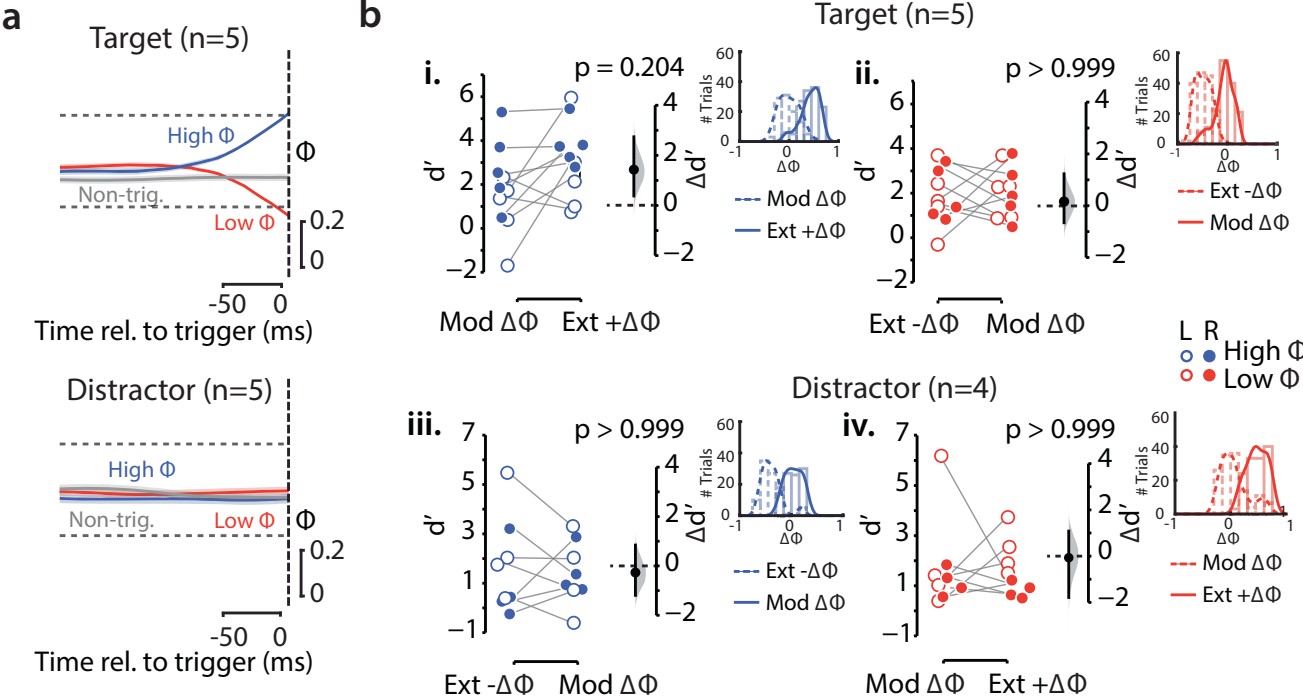

**Fig. 3 Effect of difference of SSVEP power across hemifields (ΔΦ) on target d′. a** Mean target Φ (top) and distractor Φ (bottom) traces for high-Φ (blue) and low-Φ (red) target triggered trials, as well as non-triggered trials (gray), time-locked to the presentation of the grating stimuli (dashed vertical line). Shaded region: s.e.m. **b** (i) d′ values (dots) based on a median split of ΔΦ for high-Φ, target side triggered trials. Extreme (Ext) +ΔΦ and moderate (Mod) ΔΦ refer to ΔΦ values greater than and less than the median, respectively. Open and filled circles: Trials triggered based on left- and right-hemifield Φ values, respectively. Black dot and error bar: Mean effect size and 95% confidence interval, respectively, for the difference between d′ across Ext +ΔΦ and Mod ΔΦ trials. Gray histogram: bootstrap sampling distribution for effect size. Dashed horizontal line: Datum indicating zero effect size. (Inset) ΔΦ distribution for Extreme +ΔΦ (light, solid line) and Moderate ΔΦ (light, dashed line) subsets for high Φ trials. Dark lines: Curve fits based on a kernel density estimate. (ii–iv) Same as in the top left panel, but showing median split results for low-Φ target side triggered trials, high-Φ distractor side triggered trials and low-Φ distractor side triggered trials, respectively.

triggered trials were interleaved, in equal proportion, with target triggered trials ("Methods"). In this case, we observed no apparent difference in d′, criterion or reaction time across the high-Φ and low-Φ trials (Fig. 2ci, distractor SSVEP triggered: d′: ANOVA, main effect, high-Φ versus low-Φ, $F(1, 36) = 0.23$, $p = 0.635$; BF = 0.17; Fig. 2cii, criterion: ANOVA, main effect, high-Φ versus low-Φ, $F(1,36) = 0.96$, $p = 0.333$; BF = 0.12; Fig. 2ciii, RT: ANOVA, main effect, high-Φ versus low-Φ, $F(1, 36) = 0.11$, $p = 0.740$; BF = 0.14; $n = 10$ subjects). These results confirmed that the observed behavioral enhancement of target d′ was selective to SSVEP power modulation at the target location.

To summarize, perceptual accuracy (d′) for discriminating target orientations increased during epochs of high target SSVEP power. The effect was selectively associated with modulation of SSVEP power at the target location, but not at the distractor location. Interestingly, we observed no reliable effects of SSVEP power states on reaction times. These results suggest that neural mechanisms of attention that enhance perceptual accuracy may be distinct from those that facilitate reaction times (see "Discussion").

**Local SSVEP fluctuations rather than global SSVEP power differences correlate with accuracy effects.** There is some debate in the literature about whether attention resources across the two visual hemifields are independent[31,32]. While some studies suggest that sensory processing resources for the attended hemifield (e.g., as indexed by target SSVEP power) are allocated independently of the unattended hemifield (e.g., as indexed by distractor SSVEP power)[31,33], other studies suggest that attention produces

a biased competition for sensory resources across hemifields[32,34]. While these hypotheses are not mutually exclusive, the results of Paradigm A support the former hypothesis: distractor side Φ states did not systematically predict target d′ variations. In other words, the behavioral effects on d′ were mediated, at least in part, by local, not global, SSVEP power fluctuations. To further confirm this hypothesis, we performed two additional analyses.

First, we plotted the Φ traces, separately for the target and the distractor sides, for the high-Φ and low-Φ target SSVEP triggered trials (Fig. 3a; $n = 5$ subjects with flickering stimuli and high SNR on both sides, "Methods"). Based on this analysis, we observed no evidence for or against systematic modulation of mean Φ on the distractor side across the target high-Φ and low-Φ trials (distractor Φ value: high-Φ trials = $0.46 \pm 0.15$, low-Φ trials = $0.51 \pm 0.14$, median ± sd, $p = 0.125$, Wilcoxon signed-rank test; BF = 1.36) (Supplementary Fig. 3b). In additon, trial-wise analysis revealed no evidence for correlation between target-Φ and distractor-Φ either in the low-Φ trials ($r = -0.05$, $p = 0.24$) or in the high-Φ trials ($r = -0.04$, $p = 0.40$) (Methods, section on "Trial-wise analysis of correlation between target-Φ and distractor-Φ"; see also trial-averaged correlations in Supplementary Fig. 3c; low-Φ trials, $r = 0.03$, $p = 0.91$; high-Φ trials, $r = 0.09$, $p = 0.75$).

Second, we tested if target discrimination accuracy would vary with the strength of biased, global competition between target and distractor representations, at a fixed level of target (or distractor) SSVEP power. For this, we computed the difference between the SSVEP power indices across hemifields ($\Delta\Phi = \Phi_{target} - \Phi_{distractor}$), just before stimulus presentation (at the time of threshold crossing), separately for the target triggered and distractor-

triggered trials, and separately for the high-$\Phi$ and low-$\Phi$ trials (Fig. 3b). For each trial type, we divided the trials into two subsets, based on a median split (Methods section on "Analysis of difference of SSVEP power across hemifields"). For the target triggered high-$\Phi$ trials and the distractor triggered low-$\Phi$ trials, the first subset of trials (values above the median) comprised trials typically with large positive values of $\Delta\Phi$ (extreme $+ \Delta\Phi$) whereas the second subset (values below the median) comprised of $\Delta\Phi$ values around zero (moderate $\Delta\Phi$) (Fig. 3b, i and iv). Conversely, for the target triggered low-$\Phi$ trials and the distractor triggered high-$\Phi$ trials, the first subset of trials (values above the median) comprised $\Delta\Phi$ values around zero (moderate $\Delta\Phi$) whereas the second subset (values below the median) comprised typically with large negative values of $\Delta\Phi$ (extreme $-\Delta\Phi$) (Fig. 3b, ii and iii). We confirmed that the $\Delta\Phi$ distributions for every pair of subsets was significantly different from each other (Fig. 3b, insets; $p < 0.001$, for 4 pairwise comparisons, Kolmogorov-Smirnov test). We then tested if d′ modulations would co-vary systematically with $\Delta\Phi$ for each trial type.

Accuracies did not vary systematically with $\Delta\Phi$ across each pair of subsets, both when the data were analyzed together ($F(1,56) = 0.85$, $p = 0.360$, ANOVA, $n = 5$ subjects for target triggered trials, and $n = 4$ for distractor triggered trials), and in post hoc comparisons, when each of the four trial types were compared separately ($p > 0.05$, estimation statistics).

In other words, modulations of target SSVEP power were not accompanied by congruent modulations of distractor SSVEP power in the opposite hemifield, a result consistent with earlier studies. Moreover, once local SSVEP power was fixed at specific (threshold) values, there was no evidence for a significant change in d′ with global differences in SSVEP power across the visual hemifields. In other words, sensory processing for the stimulus in the attended hemifield appears to be, at least, in part, independent of processes occurring in the opposite hemifield, a result consistent with earlier studies[34,35].

**Biased, global competitive interactions also provide a robust marker for discrimination accuracy effects.** Our findings, so far, suggest that attentional resources in the two hemifield are, at least partly, independent. Yet, previous behavioral studies suggest that the behavioral effect of spatial attention—in particular, those on d′ enhancement—are mediated by biasing competitive selection for sensory resources, globally, across the visual field[1,36–39]. We asked, therefore, if inducing biased competition among target and distractor sensory representations across visual hemifields would also produce robust behavioral effects on d′. We, therefore, designed a neurofeedback experiment to directly test the behavioral effects of biasing global competitive interactions between target and distractor representations (Fig. 4a).

Subjects ($n = 9$; cohort B) performed an orientation discrimination task (paradigm B), with a structure largely similar to the previous task design (see Methods, section on "cBMI paradigm B"), except for the following key difference: we triggered stimulus presentation not based on the local SSVEP power on each side but based on the global difference of SSVEP power ($\Delta\Phi$) across the two hemifields ($\Delta\Phi$; Fig. 4a), incorporating auditory neurofeedback.

For each trial in this experiment, one hemifield was selected pseudorandomly as a target side and the other, as the distractor side; target and distractor sides were counterbalanced, with equal probability, across the left and right hemifields. We triggered the presentation of the grating stimuli when the difference in SSVEP power index between the target and distractor sides ($\Delta\Phi = \Phi_{target} - \Phi_{distractor}$) reached a prespecified, participant-specific, threshold measured in an earlier

practice session ("Methods"). Participants reported the tilt of the grating indicated by a post hoc response probe (Fig. 4b); the target and distractor sides were probed with equal probability. As in the previous paradigm, if $\Delta\Phi$ did not reach threshold within a maximum interval (7.5 s after cue onset) stimulus was presented, regardless; these non-triggered trials ($23.9 \pm 4.3\%$, $n = 9$ subjects) were not considered for further behavioral analyses. This task enabled quantifying subjects' orientation discrimination accuracy (d′) on each side when the global SSVEP power difference between the target and distractor sides reached extreme values (Fig. 4c).

In order to trigger stimulus presentation, participants needed to reliably identify the target and distractor sides on each trial. Because the target and distractor sides were switched pseudorandomly, unbeknownst to the participants, we discovered, in initial pilot experiments, that participants found it difficult to identify these sides, by trial and error alone. To enable participants to quickly identify the target and distractor sides, we provided continuous auditory feedback: the auditory feedback frequency was scaled linearly for positive values of $\Delta\Phi$ and reached a frequency floor (500 Hz) for zero or negative $\Delta\Phi$ ("Methods"). The auditory feedback enabled participants to reliably perform the task as evidenced by the >6x more trials with positive ($87.0 \pm 2.4\%$) as compared to negative $\Delta\Phi$ ($13.0 \pm 2.4\%$) ($p = 0.004$, Wilcoxon signed-rank test). Moreover, $\Delta\Phi$ values were systematically higher for this paradigm (Fig. 4d, upper), as compared to paradigm A (Supplementary Fig. 4a), confirming the effectiveness of the auditory neurofeedback at inducing global differences in SSVEP power across the visual hemifields ($\Delta\Phi$: paradigm A $= 0.29 \pm 0.29$, paradigm B $= 0.68 \pm 0.17$, $p < 0.001$).

First, we tested if this neurofeedback paradigm induced competitive interactions across the visual hemifields. Indeed, in this paradigm, target-$\Phi$ and distractor-$\Phi$ varied in opposite directions—at trend clearly visible in trial-average traces over time (Fig. 4d, lower). In addition, trial-wise analysis revealed clear evidence for anti-correlation between the target-$\Phi$ and distractor-$\Phi$ ($r = -0.16$, $p < 0.001$; Methods, section on "Trial-wise analysis of correlation between target-$\Phi$ and distractor-$\Phi$"; see also trial-averaged correlations in Supplementary Fig. 4b; $r = -0.84$, $p < 0.001$). In other words—when subjects sought to achieve large differences between target-$\Phi$ and distractor-$\Phi$, based on auditory neurofeedback—robust competitive interactions were generated between the neural representations in each hemifield.

This cBMI neurofeedback paradigm revealed robust effects of $\Delta\Phi$ on behavioral accuracies. Discrimination d′ was significantly higher for the target compared to the distractor (Fig. 4ei, Supplementary Fig. 4ci, $p = 0.006$, d′: target $= 1.72 \pm 0.24$, distractor $= 1.41 \pm 0.28$, median $\pm$ sd; $p = 0.006$, BF $= 18.28$, $n = 9$ subjects, paired test, estimation statistics). Interestingly, we observed that criteria were closer to zero, on average, for target as compared to distractor judgments (Fig. 4eii, criterion: target $= 0.03 \pm 0.19$, distractor $= 0.23 \pm 0.25$, $p = 0.004$, BF $= 12.47$). As before, no significant differences were observed in reaction times for target versus distractor responses (Fig. 4eiii, RT: target $= 1006 \pm 252$ ms, distractor $= 991 \pm 247$ ms, $p = 0.335$, BF $= 0.17$).

To summarize: With an auditory neurofeedback paradigm, we trained subjects to generate biased competitive interactions in SSVEP power between the visual hemifields. Orientation discrimination accuracies (d′), but not reaction times, were robustly modulated when stimuli were presented when global differences in SSVEP power across hemifields reached extreme values. Specifically, d′ was significantly higher for stimuli that were presented on the side of comparatively higher SSVEP power. In other words, biasing global competitive interactions among stimulus representations across the visual hemifields provided a

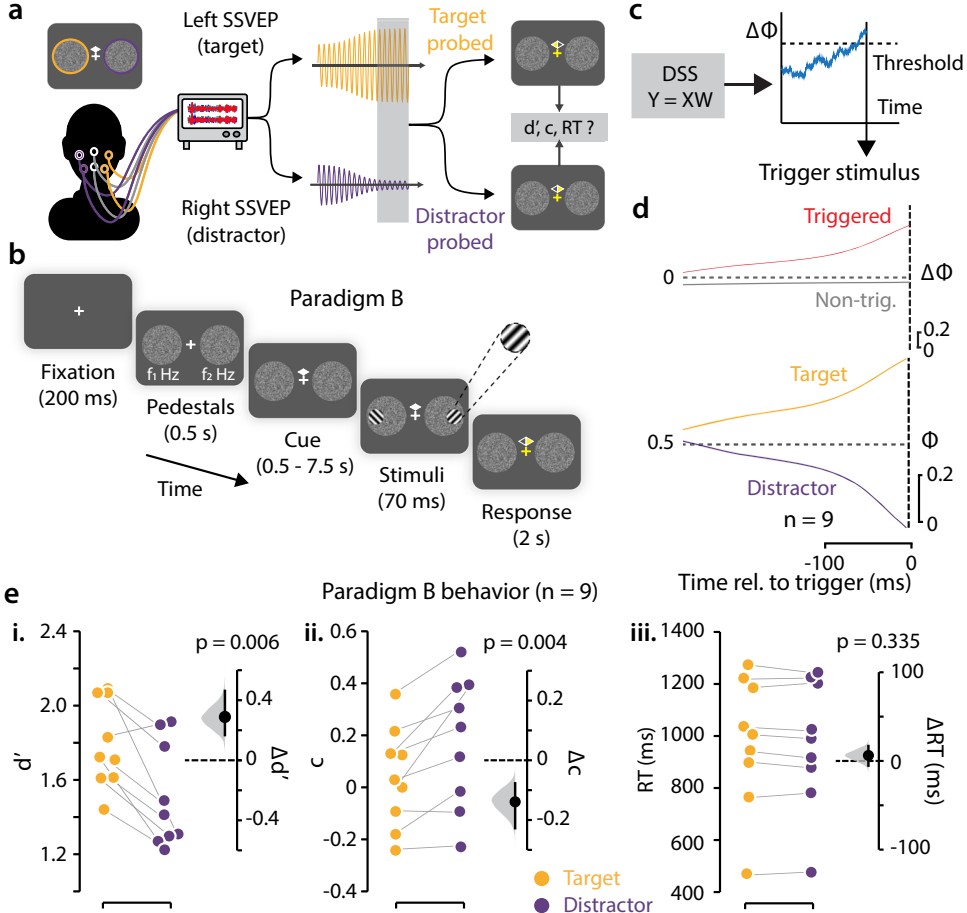

**Fig. 4 A neurofeedback paradigm confirms ΔΦ as a robust marker for attention. a** Schematic of a neurofeedback-based cBMI to investigate the effects of difference in SSVEP power across hemifields on behavior. Yellow and purple traces: Momentary, target and distractor SSVEP power states, respectively. **b** Behavioral neurofeedback task (Paradigm B). Key task details are similar to those of paradigm A (Fig. 1b). On each trial, the pedestal on one side was predesignated as a target and the other, as a distractor. Participants were provided auditory neurofeedback based on the difference of SSVEP power between the target and distractor (ΔΦ), and stimulus presentation was triggered based on ΔΦ values (see next). On each trial, one of the two sides was (randomly) probed for stimulus orientation report (filled yellow triangle). Other differences with paradigm A are discussed in the text ("Methods"). **c** Schematic of real-time stimulus triggering protocol. The recorded EEG data was preprocessed and projected, in real-time, onto the SSVEP DSS dimension (W), estimated in a baseline session (left). The projected data (Y), computed separately for the target and distractor pedestals, was used to estimate ΔΦ in real-time. When ΔΦ crossed a preset high threshold, grating presentation was triggered (right). **d** (Top) Mean ΔΦ traces for triggered (red) and non-triggered trials (gray), averaged across subjects, time locked to the triggering of the grating stimuli (dashed vertical line). Shaded region: s.e.m. (Bottom) Mean Φ traces on the target (yellow) and distractor (purple) side. Other conventions are the same as in the top panel. **e** (i) Discrimination accuracy (d′) for target-side probed (yellow) and distractor-side probed (purple) trials for Paradigm B ($n = 9$). Black dot and error bar: Mean effect size and 95% confidence interval, respectively, for the difference between d′ across target and distractor probed trials. Gray histogram: bootstrap sampling distribution for effect size. Dashed horizontal line: Datum indicating zero effect size. (ii) Same as panel i, but for choice criterion (c). (iii) Same as panel i, but for reaction times.

reliable marker for behavioral effects on discrimination accuracy, but not on reaction times.

**Slowly ramping dynamics of competitive interactions yield stronger behavioral effects**. The previous results demonstrate that robust effects on discrimination accuracy occur when global (cross-hemifield) difference in SSVEP power (ΔΦ) reach large values. Yet, large ΔΦ values also entail a high Φ value at the target location on target-side probed trials, and a low distractor Φ value at the distractor location on distractor-side probed trials. It is possible, then, that the effects observed in Paradigm B occurred due to extreme (high or low) co-fluctuations in Φ for the stimulus on the respective side (target or distractor, respectively), and were independent of Φ for the stimulus on the other side, as in Paradigm A.

To distinguish these possibilities further, we analyzed the dynamics of Φ and ΔΦ in Paradigm B immediately prior to stimulus triggering. Typically, endogenous attention is deployed at a timescale of about 200–300 ms[1] and sustaining attention for many seconds is known to impair orientation discrimination performance[40]. If either Φ or ΔΦ dynamics were to represent a neural signature of attention, we hypothesized that attention's behavioral timescales would be faithfully reflected in the respective metric's timescales as well. For this, we tested whether specific patterns in the dynamics of local SSVEP power (Φ), or its global difference across hemifields (ΔΦ), at distinct timescales would yield systematically different perceptual accuracies (Fig. 5a).

First, we divided trials into two categories—fast-ramp versus slow-ramp—based on the rate of change of Φ, just before threshold crossing, across trials of Paradigm B ("Methods"). The

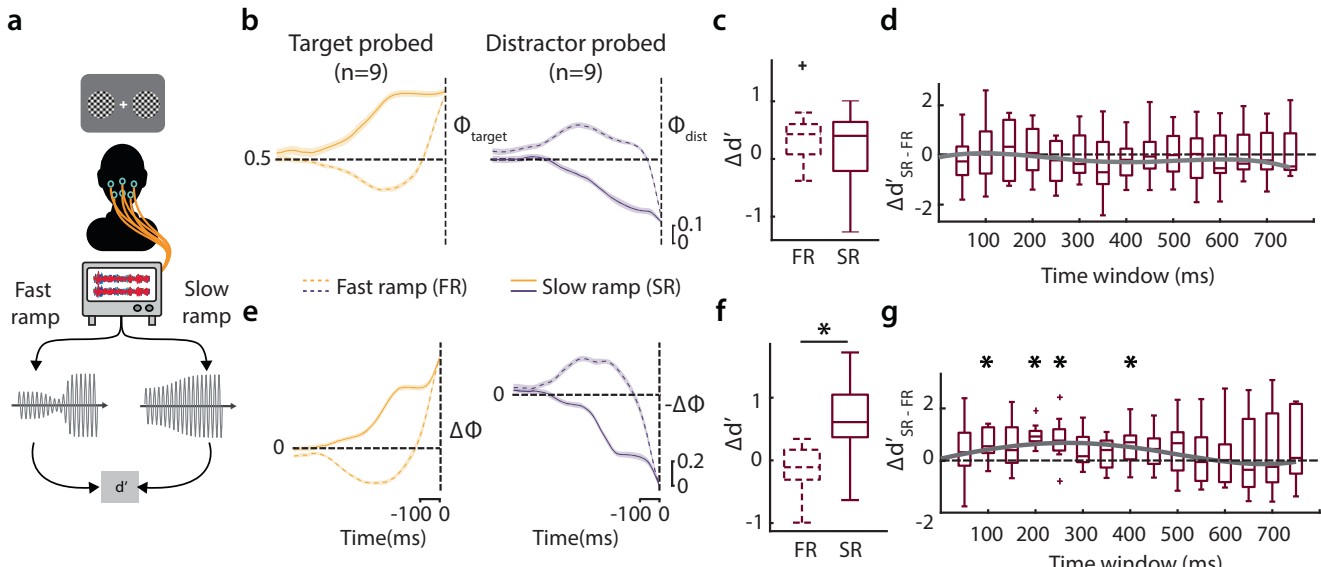

**Fig. 5 Slow-ramping dynamics of ΔΦ yield stronger d′ enhancements. a** Schematic depicting the analysis protocol for investigating behavioral differences between trials with fast or slow-ramping Φ (or ΔΦ) dynamics. **b** Mean Φ traces for the target-side (left, yellow) and distractor-side (right, purple) probed trials on the target and distractor side, respectively, leading up to grating presentation (dashed vertical line). Dashed and solid lines: fast-ramping trials ($n = 9$) and slow-ramping trials ($n = 9$), respectively. Trials pooled across subjects for Paradigm B. Shaded regions: s.e.m. **c** Difference in perceptual accuracy (Δd′) between target-side and distractor-side probed trials, computed separately for fast-ramping (FR) and slow-ramping (SR) subsets. Dashed and solid outline boxes: fast-ramping and slow-ramping trials, respectively. Central mark: median, edges: 25th and 75th percentiles, the whiskers: most extreme points, +: outliers. **d** The difference in Δd′ between slow-ramping and fast-ramping trials (Δd′$_{SR-FR}$) at different time-windows used for computing the rate of variation of Φ before threshold crossing. Central mark: median, edges: 25th and 75th percentiles, the whiskers: most extreme points, +: outliers. Solid gray line: Quartic polynomial fit. **e** Same as in **b**, but showing mean ΔΦ (Φ$_{target}$ − Φ$_{distractor}$) trace for the target-side (left, yellow) and −ΔΦ (Φ$_{distractor}$ − Φ$_{target}$) trace for the distractor-side (right, purple) probed trials. Other conventions are the same as in panel **b**. **f** Same as in **c**, but Δd′ for fast-ramping and slow-ramping trials based on ΔΦ dynamics. Other conventions are the same as in panel **e**. *$p < 0.05$. **g** Same as in **d**, but Δd′$_{SR-FR}$ for fast-ramping and slow-ramping trials based on ΔΦ dynamics. *$p < 0.05$.

average Φ traces for each of these sets of trials are shown in Fig. 5b (solid lines: slow ramp; dashed lines: fast ramp); the left panel shows target-Φ dynamics from target-side probed trials whereas the right panel shows distractor-Φ dynamics from distractor-side probed trials (for the converse dynamics, see Supplementary Fig. 5a). We then computed the difference in d′— Δd′—between these two sets of trials. Note that this quantity, Δd′, measures the difference in d′ for the probed stimulus between trials when the SSVEP power index was high, compared to when it was low, at the probed location. We tested if this d′ modulation was different between fast-ramp and slow-ramp trials.

Interestingly, when we divided trials based on ramping dynamics of Φ, we observed no significant differences between the two categories (Fig. 5c, Δd′: fast-ramp Φ = 0.43 ± 0.54, slow-ramp Φ = 0.40 ± 0.69, median ± sd, $p = 0.496$, Wilcoxon paired signrank test; BF = 0.19; $n = 9$; rate of change computed in a time-window extending from $t_s − 250$ ms to $t_s$, where $t_s$ is the stimulus onset time). In fact, regardless of the duration of the time-windows tested for computing the rate of change of Φ, there was no systematic difference in d′ modulation across fast-ramp and slow-ramp trials (Fig. 5d, Supplementary Fig. 5c).

Next, we divided trials based on ramping dynamics of ΔΦ (global difference of SSVEP power indices; average ΔΦ traces in Fig. 5e); in this case, the left panel shows positive ΔΦ (Φ$_{target}$ − Φ$_{distractor}$) dynamics when the target side was probed whereas the right panel shows negative ΔΦ (Φ$_{distractor}$ − Φ$_{target}$) dynamics when the distractor side was probed (for target and distractor Φ dynamics based on ΔΦ split, see Supplementary Fig. 5b). Remarkably, in this case, we observed a significantly higher Δd′ for slow-ramp trials as compared to fast-ramp trials (Fig. 5f; Δd′: fast-ramping = −0.11 ± 0.44, slow-ramping = 0.61

± 0.66, median ± sd, $p = 0.039$, Wilcoxon paired signrank test; BF = 5.52, $n = 9$). These Δd′ differences showed graded changes across time-windows used for computing rate of change: d′ modulations were strongest for time-windows of 200–250 ms duration before stimulus onset (Fig. 5g) and weakened at much shorter (e.g., 50 ms) or much larger (e.g., 600–800 ms) durations (Fig. 5g, Supplementary Fig. 5d).

In sum, distinct dynamics of global, cross-hemifield differences in SSVEP power (ΔΦ) were strongly predictive of accuracy (d′) modulations. In particular, slow-ramping dynamics of ΔΦ were predictive of d′ differences between the target and distractor locations, with strongest effects occurring at a timescale matching that of the deployment of endogenous attention[40]. Taken together, these results indicate that a gradual change in SSVEP power differences across hemifields represents a robust neural signature of attention's effects on discrimination accuracy.

## Discussion
The precise neural mechanisms by which attention enables adaptive behavior, remain heavily researched[41–44]. It is increasingly clear that attention is not a unitary phenomenon[36,38,45–47]. Identifying necessary and sufficient neural correlates is key to arriving at a consensus definition for attention, and for understanding how it is implemented in the brain.

Our findings provide key evidence that addresses this challenge. With an EEG-based cognitive brain–machine interface (cBMI) that tracks SSVEP power in real-time, we demonstrate that orientation discrimination accuracies, but not reaction times, were reliably enhanced when targets were presented during epochs of high, compared to low SSVEP power. Generating

biased competition for SSVEPs globally, across visual hemifields, with a cBMI neurofeedback paradigm we observed similar, robust effects on perceptual accuracies, but not reaction times. Our study identifies, therefore, specific, sufficient neural correlates of at least one behavioral component of attention: enhanced perceptual accuracy.

Voluntary control of neural activity in the brain has a long history, spanning several decades[48,49]; earliest EEG neurofeedback[50,51]. The primary use of such neurofeedback training is in motor brain–machine interfaces (BMI), where the goal is for subjects to gain control over the neural activity in the motor cortex that is the basis for neuroprosthetic applications[20–22]. While such BMIs have found extensive application in neurorehabilitation and for studying motor control, more recent applications of BMIs have sought to address basic questions about the neural underpinnings of cognitive processes (cBMIs).

Attention-based cBMIs are typically developed for one of at least two purposes: as a platform for tracking and training attention in naive subjects, and/or as a tool for understanding neural mechanisms of attention. The former purpose is, arguably, more common, particularly, in human studies[23,26–28], whereas the latter is emerging as a powerful tool for understanding attention mechanisms[24,52–54].

Yet, in previous cBMI studies, subjects' performance was gauged by their ability to readily achieve target neural states (e.g., high firing rates[27] or low alpha power[54]). In contrast, in our study, achieving specific SSVEP power states was necessary for triggering stimulus presentation, but was not directly related to task performance; behavioral performance was measured with an independent orientation judgment task. In this sense, our interventional cBMI paradigm dissociates subjects' ability to control their neural states with the effect of those neural states on behavioral metrics, such as accuracy and reaction times.

Despite a rich literature on EEG correlates of attention, relatively few studies have shown a direct correspondence between trial-by-trial fluctuations of EEG signatures and behavior. For example, Störmer et al. showed that subjects' object tracking performance was better, on average, on trials with larger attentional modulation of SSVEP power[17]. Another study demonstrated an increase in SSVEP power for targets relative to distractors on correct trials alone[55]. Yet another study showed that the time course of reaction time changes tracked the time course of SSVEP power modulations in a feature-based attention task[15]. Many studies have used the single-trial ERP analyses to link ERP changes and behavior[56–58]. Other studies have examined how behavioral performance varies on individual trials when stimuli are presented at specific phases of endogenous EEG oscillations like theta-band (4–8 Hz) or beta-band (16–35 Hz) oscillations[59,60].

Yet, nearly all previous studies examined the relationship between EEG signatures and behavior offline, with post hoc analyses. Such post hoc analyses possess a key limitation, in terms of interpretation: Neural fluctuations, on a trial-by-trial level, may be correlated with behavioral fluctuations not because of a direct relationship between the two variables, but simply because of common underlying sources, driving each. Such a correlational approach merely measures the strength of association between two variables, without any basis for inferring direct, mechanistic relationships. For example, while correlational analyses may indicate that a change in behavioral state (e.g., accuracy or reaction time) is typically accompanied by a change in neural state (e.g., SSVEP power), such an analysis cannot establish the necessity or sufficiency of this association.

In contrast to such a correlational approach, here, we adopted an interventional approach that provides a stronger basis for inferring direct relationships between neural signatures and behavior[19]. Unlike a correlational approach, an interventional approach seeks to achieve control over one of the variables being tested to measure the necessity of association between the two variables. With such an approach, stronger claims can be made regarding a direct causal relationship. For example, an experimental intervention on neural states (e.g., SSVEP power) can indicate whether a change in neural state (e.g., SSVEP power) is sufficient to achieve a change in behavioral state (e.g., accuracy or reaction time). For this reason, interventional approaches occupy a higher position on the ladder of causation (Judea Pearl's causal metamodel)[19] as compared to post hoc correlational approaches, and permit stronger inferences about direct (causal) relationships (see also Methods, section on "Modeling interventional versus correlational approaches").

We tested the link between EEG signatures and attention with such an interventional approach. We performed an experimental intervention on neural states by tracking target SSVEP power in real-time, and when it reached predetermined states, measured the effects on behavioral performance metrics (d′, RT). Our results indicate a direct link between attention's effects on perceptual accuracy and SSVEP power in the human brain. Moreover, these changes in perceptual accuracy were associated with spatially localized SSVEP power fluctuations, as would be expected to accompany visuospatial selective attention, rather than from global changes in physiological state, such as arousal[61–63].

Second, it is well known that—at least in some settings—attention can bias competition for resources across the visual hemifields[37,64,65]. We, therefore, tested the hypothesis that generating biased competition for neural representations across the visual hemifields would also produce attention-like effects on perceptual accuracy. Our cBMI neurofeedback paradigm revealed that a difference in the SSVEP power (ΔΦ) across hemifields provided a reliable marker for accuracy (d′) modulations. To decouple the effects of changes in ΔΦ from those on changes in Φ, we examined the dynamics of each metric (speed of ramping to threshold), and the corresponding effects on behavior. We observed that slow, but not fast, ramping dynamics of ΔΦ at timescales ~200–300 ms reliably predicted d′ differences between the target and distractor locations; these ramping dynamics match the timescale of deployment of endogenous attention[1]. Taken together, these results indicate that biased, globally competitive, neural dynamics, unfolding at a timescale typically associated with the deployment of voluntary attention, are a sufficient neural marker for endogenous visuospatial attention.

Although, a priori, we sought to divide trials based on the speed of ramping of ΔΦ (or Φ), trials with both slow and fast dynamics exhibited temporal features beyond those characterizing the speed of ramping alone. For example, fast ramping ΔΦ trials showed a marginally negative slope of ramping, favoring the distractor side, followed by a large positive slope, and rapid rise toward the target side (Fig. 5e, left), potentially indicating that attention was initially deployed toward the distractor side and rapidly reoriented toward the target location on these trials. By contrast, slow-ramping ΔΦ trials showed a gradual increase toward the target location followed by a steady levels of activity prior to crossing the threshold, potentially indicative of sustained attention, for a few hundred milliseconds, toward the target location on these trials. Future studies could characterize in detail these, and other distinct types of, SSVEP power dynamics, and their consequences for behavior.

The results based on our SSVEP state-locked stimulus presentation paradigm deviate from two key observations in literature. First, previous studies have documented systematic effects of SSVEP power on reaction times (RT). For instance, Störmer et al.

showed that trials with larger modulations of SSVEP power showed correspondingly lower RTs[15,17]. On the other hand, we observed no systematic effects of SSVEP power modulations on RTs, in either task paradigm (Figs. 2b, c, 4e). In both of these tasks, the response window was fixed, motivating subjects to respond quickly (1.5–2 s, Figs. 1b, 4b). Mean RTs were in a range typically observed in psychophysics tasks (paradigm A: 578 ± 32 ms, paradigm B: 1017 ± 111 ms, mean ± s.e.m.). Nonetheless, because we did not explicitly instruct subjects to respond quickly, it is possible that, RT effects arising from SSVEP power modulations were not as pronounced as those reported in previous studies.

Yet, an alternative and, arguably, more plausible explanation is as follows: In previous studies, attention could have engaged multiple and, potentially, dissociable mechanisms, which were responsible for the distinct effects on accuracies, RTs, and SSVEP power. Post hoc trial-wise analyses cannot decouple these effects, in practice, because attention may have engaged these distinct mechanisms in a highly correlated manner (see Methods, section on "Modeling interventional versus correlational approaches"). On the other hand, our interventional cBMI approach provided a direct test of this hypothesis by revealing no systematic effects on RT when stimuli were triggered at particular SSVEP power states. Our results are in line with literature which suggests that attention mechanisms that modulate perceptual accuracy are distinct from those that modulate reaction times[12,66,67].

Second, previous literature suggests that attentional resources across the left and right visual hemifields are separate and, potentially, independent; this dissociation has been reported in many studies including human psychophysics[31,33], human electrophysiology[34,35], and non-human primate electrophysiology[8]. In line with these findings, in Paradigm A, target discrimination d′ was not systematically modulated either by distractor SSVEP power states ($\Phi_{distractor}$), or by the global difference of SSVEP power across hemifields ($\Delta\Phi$), suggesting that SSVEP power modulations may occur independently across brain hemispheres, by default, and independently affect performance across the hemifields during cued attention tasks. Yet, in Paradigm B, when subjects were trained using neurofeedback, to increase the difference of SSVEP power across hemifields ($\Delta\Phi$)— and stimuli were triggered based on this neural signature of biased stimulus competition[32,65]—we observed robust differential modulations of d′ across the hemifields, with Bayes Factor values around 18.0.

We propose the following hypothesis that reconciles our findings with those in previous literature. SSVEP power is modulated by a combination of processes: These include both local selection processes that operate independently across visual hemifields—by engaging dissociable neural processing resources in each brain hemisphere—as well as global selection mechanisms that involve biased competition for neural resources between hemispheres. Under normal conditions, the former set of processes (local selection) dominate. But, when subjects were explicitly provided with neurofeedback based on the difference in SSVEP power across hemifields, global selection processes became strongly engaged. In other words, our neurofeedback paradigm engendered global, spatial selection by inducing biased competition for neural resources across brain hemispheres, and thereby produced robust effects on behavioral accuracies. Whether such a neurofeedback cBMI can be employed to train subjects to produce behavioral effects that are considerably stronger than those reported in standard attention tasks, remains to be explored.

Our study leaves open several interesting avenues for future exploration. Can specific EEG signatures be identified as necessary neural correlates of attention? What are the sufficient neural correlates of attention's signature effects on reaction times? Can the activity of particular brain regions be voluntarily controlled with cBMI neurofeedback to identify their specific, and potentially distinct roles, in attention control? Could patients with attention deficits be identified by their ability to control these distinct signatures? Our attention-based cBMI provides a viable platform for addressing these questions and may enable devising remedial measures for treating and managing attention disorders.

## Methods

**Ethics declaration**. Twenty-four subjects (9 females; age range: 20–28 years; median age: 23 years) with no known history of neurological disorders and with normal or corrected-to-normal vision participated in the experiment. All participants provided written, informed consent, and all experimental procedures were approved by the Institute Human Ethics Committee at the Indian Institute of Science, Bangalore.

**Cognitive brain–machine interface (cBMI) based on EEGs**. Conventional brain–machine interfaces (BMI) measure brain activity, typically from populations of neurons, to decode subjects' intended actions and actuate prosthetic controllers (e.g., robot arms). Cognitive BMIs (cBMIs) are an emerging technology that acquires neural activity linked to cognitive processes, in real-time, for closed-loop neurofeedback. Here, we designed and constructed an EEG-based interventional cBMI for tracking visuospatial attention. We describe, next, the configuration of the cBMI system.

The cBMI system broadly comprises of three modules: (a) EEG acquisition hardware, (b) an EEG data processing system, and (c) a stimulus/feedback presentation system (Supplementary Fig. 1). We acquired EEG data with a high-density 128-channel acquisition system (ActiveTwo, Biosemi Inc.). The processing system (Intel Core i7 CPU, 16 GB RAM, running Windows 10) imported these EEG data using the Fieldtrip acquisition software (see next). EEG data were processed to obtain SSVEP dimensions with high signal-to-noise (see section on "EEG and behavioral data analysis"). The neurofeedback signal was computed and routed to the presentation system by means of a shared drive located on the presentation system. Finally, the presentation system (Intel Core i5 CPU, 8GB RAM, running Windows 7) presented accurately timed visual stimuli and neurofeedback signals on the monitor. Stimuli and neurofeedback were presented using Psychtoolbox 3[68–70] on Matlab (version: 2015b). Concurrently with stimulus/neurofeedback display, the presentation system also routed task-specific neurofeedback event markers to the EEG acquisition system using a parallel port. The closed-loop delay between the EEG data acquisition and neurofeedback was measured using the following approach: We measured the time between specific EEG events recorded by the acquisition hardware and the neurofeedback event marker times received by the acquisition system. We tested different acquisition software and acquisition parameters; the procedure for identifying combinations that minimized closed-loop delay are described below (section on "Optimizing closed-loop delay"). We used FieldTrip software[71] with a sampling rate of 128 Hz, which yielded a fixed overhead delay (Supplementary Fig. 1b) of ~10 ms. Overall, in our real-time experiments, the closed-loop delay was <50 ms across all trials (21.1 ± 5.9 ms, mean ± std, across trials; Supplementary Fig. 1e).

**Optimizing closed-loop delay**. As the cBMI system (Supplementary Fig. 1a) works in real-time, the closed-loop delay of the acquisition block was a crucial parameter that determined the latency of the neurofeedback. We measured the closed-loop delay with four acquisition softwares: ActiView, Lab Streaming Layer, OpenVibe, and Fieldtrip; using a procedure described above (section "Cognitive brain–machine interface (cBMI) based on EEGs"). We measured this delay by systematically varying the data packet size from 4 to 64 samples, across three different sampling frequencies (128 Hz, 256 Hz, and 512 Hz, Supplementary Fig. 1b). In each case, we fit a line to the closed-loop delay as a function of packet size, of the form:

$$D = mp + c \qquad (1)$$

Where $D$ is the closed-loop delay, $p$ is the packet size, $m$ is the slope and $c$ is an offset (y-intercept) that determines the overhead for each software associated with reading data from the acquisition hardware into the processing system. Among the 4 softwares compared, Fieldtrip provided the least overhead of 10.9 ± 0.5 ms.

**EEG and behavioral data analysis**. To study the link between EEG SSVEP power states and attention, we tested two behavioral paradigms using the cBMI platform. We describe next, EEG and behavioral analyses common to the two paradigms.

**EEG data acquisition and minimal preprocessing**. EEG data was recorded from 41 occipital scalp electrodes out of a total of 128 electrodes. The data was streamed for real-time analyses with Fieldtrip software at 128 Hz. Data sampled at 4.096 kHz was concurrently stored for offline analyses.

Offline analysis was performed with custom scripts written in Matlab (version: 2017a and version: 2019a). Spectral analyses were performed with the Chronux toolbox (version: 2.12[72]). SSVEP topoplots—topographical plots showing SSVEP dimensions (e.g., Fig. 1c, inset)—were visualized with the EEGLab toolbox (version: 13.6.5b[73]). For these offline analyses, EEG data was preprocessed using the Statistical Correction of Artifacts in Dense-array Studies (SCADS[74]) approach. First, the data was filtered from 2 to 43 Hz using an FIR filter of order 19. The filter was designed using Matlab's signal processing toolbox. Noisy electrodes were identified by visual inspection and also with an automated algorithm that identifies outlier signal values based on their maximum amplitude, maximum gradient, and standard deviation of EEG signal amplitude (±3 standard deviation of the respective median); noisy electrodes, thus were removed from further analysis. Finally, the EEG data were re-referenced to the common average, and z-scored. All of these steps (filtering, electrode rejection, re-referencing and z-scoring) were also performed also for the real-time cBMI session except that noisy electrodes were identified beforehand, in a baseline session preceding the actual real-time experiment (see section on "Baseline EEG session").

**Denoising source separation (DSS)**. To quantify SSVEP power evoked by the flickering pedestals, we sought to first isolate the electrodes that maximally expressed power at the SSVEP frequencies, as compared to other frequencies. For this, we employed an algorithm for latent source identification and dimensionality reduction: Denoising source separation (DSS). DSS (also called Joint Decorrelation) identifies linear latent dimensions embedded in high-dimensional noisy electrode data, that maximize power in specific frequencies of interest[29,75]. Briefly, the algorithm involves whitening the electrode signals followed by rotating the data along a direction that maximizes the variance for the desired feature, here, power at each SSVEP. In detail: Each DSS latent dimension ($Y$) comprises a linear combination of the raw EEG signals ($X$; $Y = XW$), where $W$ represents the weight or projection (or analysis) matrix that converts between electrodes and latent dimensions. The algorithm consists of the following steps: $X$ is first whitened with a succession of linear transformations:

$$Z = XPN \qquad (2)$$

Where $N = 1/\sqrt{D}$, and $P$ and $D$ are the eigenvector and eigenvalue matrices, respectively, obtained by eigendecomposition of the covariance matrix of $X$ ($C_o = X^T X$). In other words, $X$ is rotated along the principal components using the rotation matrix, $P$, and further scaled using the scaling matrix, $N$ to have uniform variance across dimensions. The whitening step enables estimating latent dimensions that are unbiased by differences in signal strength among dimensions in the electrode space. Next, the whitened data are filtered in the frequency domain with a bias filter[29] centered at the fundamental frequency of each SSVEP (each flicker frequency), such that

$$Z_f = LZ \qquad (3)$$

Where $L$ is the bias filter matrix. Eigen decomposition of the covariance matrix of $Z_f$ ($C_1 = Re(Z_f^T Z_f)$) provides a second rotation matrix $Q$, whose columns indicate dimensions expressing maximum variance for the SSVEP frequency of interest; columns of $Q$ are ordered by proportion of explained variance. The projection matrix $W$ is then constructed as:

$$W = PNQ \qquad (4)$$

This achieves the transformation from the raw data, $X$ into the desired latent dimensions, $Y$ that maximally express SSVEP power (e.g., Fig. 1c). The projection vector indicating the SSVEP dimension for each flicker frequency and subject was identified with visual inspection, normalized by its $L_2$ norm, averaged across subjects, and visualized in Supplementary Fig. 1c.

**SSVEP SNR calculation**. The signal-to-noise ratio (SNR) for these DSS components (Supplementary Fig. 1d) was estimated with the data from a baseline EEG session (see section on "Baseline EEG session"). EEG data were minimally preprocessed and epoched into non-overlapping epochs of 1 s duration; trailing epochs of <1 s duration were NaN padded. The epoched data were projected onto the DSS latent dimension (projection matrix W), and time-averaged across all trials for each participant. Spectral power was estimated with multi-taper spectral estimation (one Slepian taper)[72,76,77]. SSVEP SNR for each DSS dimension was then calculated as the ratio of the power at the SSVEP frequency, $f$, and average power in a frequency band from $f − 10$ to $f + 10$ Hz in increments of 1 Hz, excluding the SSVEP frequency itself. Due to low SSVEP SNR, data from 4 out of the 30 DSS dimensions in paradigm A (data from one hemisphere each for 4 subjects) were post hoc rejected (Supplementary Fig. 1d, upper); whereas no dimensions from paradigm B were rejected due to low SNR (Supplementary Fig. 1d, lower).

**Signal detection theory (SDT)**. We employed signal detection theory[78] to measure orientation discrimination accuracy in both paradigms. In our experiment, participants had to report whether the target grating, on each trial, was oriented clockwise (rightward tilt) or counterclockwise (leftward tilt), relative to the vertical meridian. Participants' responses were organized into a 2 × 2 stimulus response contingency table. Clockwise responses were arbitrarily designated as hits and misidentifications: Hit rates corresponded to the proportion of trials in which

clockwise target orientations were correctly identified as clockwise, whereas misidentification rates corresponded to the proportion of trials in which counterclockwise target orientations were incorrectly reported as clockwise. We employed a conventional, one-dimensional SDT model to estimate discrimination accuracy (d′) and decision criterion (c), for the participant based on these hit rates (H) and misidentification rates (M), as follows:

$$d' = \varphi^{-1}(H) - \varphi^{-1}(M) \qquad (5)$$

$$c = -1 \backslash 2 \left( \varphi^{-1}(H) + \varphi^{-1}(M) \right) \qquad (6)$$

Where $\varphi^{-1}$ is the probit function defined as the inverse of the cumulative distribution function of the standard normal distribution. Here, d′ measures the subject's ability to discriminate clockwise from counterclockwise orientations (discriminability of the clockwise versus counterclockwise signal distributions, Fig. 1a inset); larger values of d′ index higher discrimination accuracy. The criterion measures the bias for reporting one orientation over the other (Fig. 1a inset); positive values of the criterion represent a higher bias for reporting clockwise orientation.

**Baseline EEG session**. The objectives of the baseline EEG session were two-fold: first, to identify latent EEG dimensions that expressed the strongest SSVEP power, and second, to develop an individual-specific baseline distribution of SSVEP power values for later application in the cBMI session.

For the first objective, we used a dimensionality reduction algorithm called Denoising Source Separation (DSS[29,75]). Briefly, DSS identifies latent dimensions in the multidimensional space of EEG electrodes that express SSVEP oscillations of a specified frequency with high signal-to-noise ratio (see section "EEG and behavioral data analysis", DSS). For this session, we recorded EEG data from 41 occipital electrodes (Supplementary Fig. 1c) overlapping early visual regions, which have been consistently identified in literature as the neural source of SSVEP oscillations[15–18]. EEG data from 500 ms after the onset of the flickering pedestals until their offset was minimally preprocessed for outlier removal (see sections "EEG and behavioral data analysis", "EEG data acquisition and minimal preprocessing"), and epoched into non-overlapping epochs of 1 s duration; trailing epochs of <1 s duration were NaN padded. These data were used to construct the DSS projection matrix $W_f$ separately for each flickering frequency of SSVEP employed in each experiment. We refer to the matrices $W_f$ as spatial dimensions in the main text, in the sense that these matrices indicate the projection weights across electrodes for the DSS dimension that maximizes SSVEP power at each frequency $f$ (see for example, Fig. 1c main text and Supplementary Fig. 1c).

For the second objective, we estimated a normalized SSVEP power index, separately for each flicker frequency. EEG data was preprocessed offline, as described above, and projected onto the DSS dimension for the respective SSVEP frequency. To simulate the SSVEP power distribution during the actual real-time experiment, data were epoched into successive 500 ms bins (64 samples) offset by 1 sample (7.8 ms). We then applied multi-taper spectral estimation (one Slepian taper[72]). SSVEP power estimates were then averaged (smoothed) across eight successive bins; the distribution of these smoothed values of SSVEP power during this baseline session for a representative subject (18 Hz flicker frequency) are shown in Fig. 1d. To obtain a normalized SSVEP power measure for each participant, we computed the cumulative distribution function (CDF) of this SSVEP power distribution using a non-parametric kernel density estimator (Kernel Probability Distribution Object - MATLAB). The CDF value—which we call a normalized SSVEP power index or Φ—varies between 0.0 and 1.0 and provides a normalized measure of SSVEP power with a temporal resolution of ~500 ms. By accounting for inter-individual variations in SSVEP power, Φ provides a measure of the strength of SSVEP power modulation that can be directly compared across participants. This normalized SSVEP power index was used in the subsequent real-time cBMI session to track modulations in SSVEP power.

In addition, the baseline session served to estimate individual-specific best SSVEP frequencies. For 5/15 participants we used fixed frequencies of 14 Hz and 18 Hz for the SSVEPs (Supplementary Table 1). These frequencies were counterbalanced between hemifields across participants. Perhaps because these were not ideal frequencies for evoking SSVEPs for these participants, signal-to-noise ratios (SNR) were poor for SSVEPs in 4/10 DSS dimensions estimated (Supplementary Fig. 1d). Therefore, for the remaining 10 participants (who also ran distractor-triggered trials, see next section), we ran around 40 pilot trials by testing different SSVEP frequencies in the range of 13–18 Hz (step: 1 Hz), and selected the frequency that provided the strongest SSVEP response for that participant (Supplementary Table 1); these were found to be either at 13 Hz or at 15 Hz. With this procedure, in all cases, DSS dimensions exhibited good SNR values for the SSVEPs (Supplementary Fig. 1d). For this subset of participants, the stimulus on the other side was either non-flickering (0 Hz, 6/10 participants) or flickered at 25 Hz (4/10 participants); these latter frequencies were chosen to be sufficiently apart from the 13 Hz/15 Hz flicker frequency, so to ensure reliable isolation of SSVEP power at these frequencies. The stimulus tagged with individual-specific flicker frequencies (SSVEP stimulus) was presented with equal probability across left and right visual hemifields, and cues were counterbalanced across the SSVEP stimuli and the other (non-flickering or 25 Hz) stimulus such that either could be a target or distractor with equal probability. By this careful task

design, we avoided systematic spatial biasing of attention toward any one side or any one type of stimulus in our experiments.

**Sample size estimation**. Power analysis for sample size was performed for Cohort A with data from the first $n = 5$ subjects. Mean discrimination accuracy (d′) for the high and low SSVEP power was computed for the 5 subjects (6 SSVEP dimensions) and used as representative of the effect size. Sample sizes were determined based on $\alpha = 0.01$ and $\beta = 0.2$. This analysis estimated a sample size of $n = 15$. Power analysis for sample size was performed for Paradigm B, separately, with data from the first $n = 5$ subjects. This analysis estimated a sample size of $n = 20$. Experiments were conducted until $n = 11$ subjects' data was acquired. Data collection was suspended following the onset of the coronavirus pandemic. Nonetheless, Paradigm B revealed sufficiently robust effects even with the fewer subjects tested (e.g., Fig. 4).

**cBMI paradigm A**. The first paradigm (paradigm A) sought to identify changes in behavioral metrics—reaction time, d′, and criterion—accompanying divergent states of SSVEP power at the target location.

Fifteen healthy adult human participants (six females) with an age range of 20–26 years (mean = 22.6 yrs, std = 1.8 yrs) with normal or corrected-to-normal vision and no known history of neurological disorders were included in this experiment. Participants gave written, informed consent prior to their participation in the experiment, and were monetarily compensated for their time.

Participants performed the task in an isolated dark room. The participant's head was positioned 60 cm away from the monitor on a chin rest. The task was presented on a 24-inch, contrast-calibrated LCD monitor with a resolution of 1920 × 1080 at 144 Hz screen refresh rate. Stimulus presentation was performed with Psychtoolbox version 3.0[68–70] with Matlab version 2015b (Mathworks Inc., Natick, MA). Two keys of a five key response box (RB-540, Cedrus) were used to record participants' responses. A neutral 50% gray background was maintained throughout these experiments. Each trial began with a central positive contrast (white) fixation cross (0.5 dva [degrees in visual angle] diameter), presented in the center of the screen. Participants were instructed to fixate on the cross throughout each trial. Subjects' fixation was monitored with an infrared eye-tracker and gaze position was sampled at 500 Hz and stored for offline analysis. Gaze was tracked binocularly with an infrared eye tracker (SMI iViewX), and stored at 500 Hz sampling rate for offline analysis (see section on "Eye-tracking"). The flickering pedestals consisted of a pair of circular plaids (5.3 dva diameter) that were generated by superimposing two diagonal gratings (spatial frequency: 1.0 cpd [cycles per degree], oriented at 45° and −45° relative to the vertical meridian. The centers of each pedestal were positioned also at an azimuthal eccentricity of 5.3 dva away from the central fixation cross along the horizontal meridian. Each pedestal flickered at a distinct frequency to evoke oscillations at a fundamental frequency identical with that of the stimulus; such evoked oscillations are known as Steady-State Visually Evoked Potential or SSVEPs[18]. SSVEPs used for individual participants are tabulated in Supplementary Table 1. Details regarding these specific choices of SSVEPs are provided in the subsection on "Baseline EEG session" below. The discriminanda (target and distractor; Fig. 1b) consisted of sine-wave gratings (spatial frequency: 0.6 cpd) with a diameter of 2.1 dva and oriented at either +45° or −45° relative to the vertical meridian, presented concentrically with the pedestal stimuli.

Participants performed a two-alternative forced choice (2-AFC), grating orientation discrimination task. Each trial began with the fixation cross appearing at the center of the screen for 200 ms, followed by the two flickering pedestals, one on each hemifield. Each pedestal flickered at a distinct frequency to generate two distinct SSVEPs over opposite hemispheres of the brain (see Methods, section on "EEG and behavioral data analysis", DSS). After an interval of 1000 ms, a directed cue (central arrow, 0.5 dva length) appeared above the fixation cross, indicating the side to be attended. The cue had a validity of 100% and the cue indicated either the left or the right hemifield pseudorandomly with equal probability of 0.5 (counterbalanced across the left and right) across trials. A variable interval later (duration described in the next section), the pedestals disappeared, and target and distractor stimuli (oriented diagonal gratings) flashed briefly for ~75 ms on each side, concentric with the pedestals. Following this, the fixation cross changed color indicating the beginning of the response epoch, which lasted for 1.5 s. Participants reported the orientation of the grating on the cued side (target) as being clockwise (45°) or counterclockwise (−45°) of vertical, ignoring the grating on the uncued side (distractor). Participants reported their responses with distinct response keys for clockwise versus counterclockwise orientations.

Each experiment comprised of four sessions—a behavioral training session, a behavioral staircasing session, a baseline EEG session, and a real-time cBMI session —all conducted within the span of a single day. Typically, the entire experiment lasted for about 2.5 h for each participant. Each of the 4 sessions is described, next.

The first session was the training session. In this session, participants were familiarized with the task, including the stimuli and response protocol. Participants performed the cued orientation discrimination task (described above) without concurrent EEG recordings, and received behavioral feedback (on screen) after each trial indicating whether their response was correct or incorrect. For these trials, following flickering pedestal presentation (1 s), the cue-target interval (CTI, or the interval between cue presentation and the appearance of target stimuli) was

set to a minimum of 1.5 s and a maximum of 4.0 s. The variable interval between the minimum and maximum CTI was sampled from an exponential distribution, with a mean of 1.0 s. Participants typically performed around 80–100 trials in the training session to familiarize themselves with the task.

The next session was the staircasing. In the staircasing session, participants performed the same task as in the training session (without concurrent EEG recordings), but were not provided behavioral feedback regarding their accuracy after each trial. In this session, the contrast of the target and distractor stimuli were staircased to achieve an overall accuracy of 70%. All other task parameters were identical to the training session. Participants typically performed 60–120 trials in the staircasing session.

Staircasing was followed by the baseline EEG session. In this session subjects performed a behavioral task that was identical to the one in the staircasing session except that, in this case, the contrast of the target and distractor gratings were fixed to their values determined during the staircasing session. In addition, EEG data were sampled at 4.096 kHz and downsampled to 128 Hz and stored for offline analyses. The objectives of the baseline EEG session were two-fold: first, to identify latent EEG dimensions that expressed the strongest SSVEP power, and second, to develop an individual-specific baseline distribution of SSVEP power values for later application in the cBMI session; these are described in the Methods, section on "Baseline EEG session".

Real-time cBMI was the final session. In this real-time cBMI session, EEG data was streamed in real-time sampled at 128 Hz (Fieldtrip toolbox), and stored in in a real-time buffer of duration 500 ms (64 samples); the buffer was updated with each new sample so that it always contained the last 500 ms of streaming data. EEG data were minimally preprocessed (see Methods, section on "EEG data acquisition and minimal preprocessing"). These data were then projected in real-time onto a latent dimension with DSS (projection matrix $W_f$), which was estimated separately for each SSVEP flicker frequency during the baseline block. Power at the corresponding SSVEP frequency was estimated using multi-taper spectral estimation (one Slepian taper). SSVEP power estimates were averaged across eight successive time points and, as described in the previous subsection, used to compute the SSVEP power index ($\Phi$) based on the CDF of the baseline SSVEP power distribution. Separate $\Phi$ values were computed for the two SSVEP frequencies, such that we were able to track $\Phi$ for both the cued (target) and uncued (distractor) pedestals in real-time on each trial.

On each trial, when the $\Phi$ for the target pedestal reached predetermined high or low threshold values, the presentation of the target and distractor gratings was triggered; we call these trials, respectively, high-$\Phi$ and low-$\Phi$ trials. Each trial was pseudorandomly designated as either a high-$\Phi$ or low-$\Phi$ trial, a priori, and these designations were counterbalanced across the cued hemifields. For the first 4/15 participants (sub-group I), high-$\Phi$ and low-$\Phi$ thresholds were set to 0.8 and 0.2, respectively, but resulted in a high proportion of non-triggered trials in pilot experiments. Consequently, thresholds were relaxed to 0.7 and 0.3 for the remaining 11/15 participants (sub-group II). No significant interaction effects between behavioral parameters and sub-groups were observed (d′: main effect of high-$\Phi$ vs low-$\Phi$: $F(1,48) = 7.58$, $p = 0.008$, main effect of group: $p = 0.752$, interaction effect: $p = 0.088$; criterion: main effect of high-$\Phi$ vs low-$\Phi$: $F(1,48) = 0.09$, $p = 0.761$, main effect of group: $p = 0.529$, interaction effect: $p = 0.890$; RT: main effect of high-$\Phi$ vs low-$\Phi$: $F(1,48) = 0.01$, $p = 0.916$, main effect of group: $p = 0.005$, interaction effect: $p = 0.806$), indicating that modulations (across high-$\Phi$ and low-$\Phi$ trials) were not different across the two sub-groups for any behavioral parameter. Therefore, parameters across all 15 participants were combined for subsequent analyses. To avoid very short cue-target intervals, target presentation never occurred at time-windows of <500 ms from cue onset. In addition, target presentation always occurred if the respective threshold had not been reached within 4 s of cue onset, regardless of the value of $\Phi$. We term these latter trials as non-triggered trials (31.5 ± 3.7% of all trials), and these were not considered for further analyses. For a subset of $n = 10$ (out of 15) participants, we also triggered stimulus presentation based on the $\Phi$ value on the distractor side on 50% of the trials, again pseudorandomly selected a priori, and counterbalanced across the high-$\Phi$ and low-$\Phi$ threshold conditions. Subjects typically ran 6 blocks of 60–64 trials each. For one subject (A6, Supplementary Table 1) the session was terminated after 5 blocks, owing to the subject's inability to continue the experiment.

Following target and distractor presentation, participants reported the orientation of the grating on the cued side (target), ignoring the grating on the uncued side (distractor) within a fixed response window (1.5 s). Participants' behavioral accuracy—d′ (see section on "Momentary SSVEP power fluctuations predict discrimination performance" in Results)—as well as reaction times and decision criteria, were measured, and compared across the high-$\Phi$ and low-$\Phi$ threshold conditions (e.g., Fig. 2b, c; see section on "Statistical tests").

**Eye-tracking**. Gaze was tracked binocularly with an infrared eye tracker (SMI iView X Hi-Speed) throughout the experiment, sampled at 500 Hz, and stored for offline analysis. To ensure that subjects' gaze remained stable on the fixation cross at the time of stimulus presentation, we employed a 600 ms window (−500 ms to +100 ms with respect to task stimuli onset) for trial rejection based on eye-tracking. Trials in which gaze deviated from the central fixation cross by more than ±1 dva along the azimuth during this window of stability were rejected from

further analysis. Trials in which blinks occurred or the gaze position was lost for more than 100 ms in this stability window were also rejected. Overall, $8.3 \pm 2.4\%$ of trials across all participants were excluded due to poor gaze fixation in this paradigm.

**Analysis of cueing-induced gain of SSVEP power (Fig. 2a).** We tested whether cueing of spatial attention induced a gain in SSVEP power. For this, we analyzed non-triggered trials alone in paradigm A ($n = 15$ subjects), because in this subset of trials SSVEPs did not systematically diverge to extreme values (high or low) on the cued side. For each subject, the change in the SSVEP power index ($\Phi$) over baseline was calculated by dividing the instantaneous $\Phi$ with the mean $\Phi$ in a baseline interval $-1500$ ms to $-1000$ ms before the onset of the cue. Figure 2a shows the mean trace of this fractional change in $\Phi$ over baseline ($\Phi_{norm}$) for cued and uncued trials across all subjects in paradigm A ($n = 15$). The post-cue difference in $\Phi$ between cued and uncued conditions was calculated in a window from $-500$ ms to 0 ms prior to stimulus triggering, and compared across the cued and uncued hemifields using Wilcoxon signed-rank test (Fig. 2a, inset). In addition, we also compared $\Phi$ between cued and uncued conditions during the pre-cue epoch, in a window from $-500$ ms to 0 ms prior to cue onset.

**Trial-wise analysis of correlation between target-$\Phi$ and distractor-$\Phi$.** We computed trial-wise correlations between target-$\Phi$ and distractor-$\Phi$. For this analysis, we considered target and distractor-$\Phi$ values computed in 4 windows of 250 ms duration centered at uniformly spaced time points ranging from 0 to 1000 ms prior to stimulus triggering (duration: $+/-125$ ms or $+/-16$ samples at fs = 128 Hz). For statistical analyses, the Pearson correlation coefficient was computed between the z-scored (trial-wise) target-$\Phi$ and distractor-$\Phi$ values, pooled over all trials and participants. Since the trial-wise correlation analysis has a large number of data points (i.e., >400 data points, 4 points per trial and >25 trials per participant), only the trial-averaged target and distractor-$\Phi$ values were plotted for these time windows across participants in Supplementary Fig. 3c and Supplementary Fig. 4b, for clarity of visualization.

**Analysis of difference of SSVEP power across hemifields ($\Delta\Phi$).** We tested whether the difference in SSVEP power indices between the target (cued) and the distractor (uncued) hemifield ($\Delta\Phi = \Phi_{Target} - \Phi_{Distractor}$), rather than the power index ($\Phi$) would provide a more reliable marker for modulations of behavioral accuracy in paradigm A. This analysis was performed offline, as a precursor to the next online experiment (paradigm B). EEG data were analyzed offline using a procedure identical to that applied for the real-time data, except that in this case, the difference of SSVEP power index values between the target and the distractor hemifields ($\Delta\Phi$) was computed for each trial at the time of threshold crossing, separately for each condition: (i) target triggered vs distractor triggered, (ii) high-$\Phi$ vs low-$\Phi$, and (iii) left vs right hemifield SSVEP (Fig. 3b). For each subject, we divided trials in each condition into two subsets, based on a median split of $\Delta\Phi$. For the target triggered high-$\Phi$ trials and the distractor triggered low-$\Phi$ trials, trials with $\Delta\Phi$ values above the median, typically with large positive values of $\Delta\Phi$, were labeled as extreme $+\Delta\Phi$ trials, whereas the trials with $\Delta\Phi$ values below the median, typically $\Delta\Phi$ values around zero, were labeled as moderate $\Delta\Phi$ trials (Fig. 3b, i and iv, respectively). Conversely, for the target triggered low-$\Phi$ trials and the distractor triggered high-$\Phi$ trials, the subset of trials with $\Delta\Phi$ values above the median were labeled as moderate $\Delta\Phi$; whereas the subset with $\Delta\Phi$ values below the median was labeled as extreme $-\Delta\Phi$ (Fig. 3b, ii and iii, respectively). We tested whether the $\Delta\Phi$ distributions for every pair of subsets were significantly different from each other with pairwise Kolmogorov-Smirnov (KS) tests. We then tested if d′ modulations co-varied systematically with $\Delta\Phi$ for each trial type using two kinds of statistical tests. Firstly, we performed an n-way ANOVA with d′ as the dependent variable and $\Delta\Phi$, $\Phi$ (high-$\Phi$ vs low $\Phi$ trials), hemisphere (left and right), and type of triggering (target vs distractor) as independent factors. Secondly, in post hoc comparisons, each of the four trial types were analyzed separately using estimation statistics[79]. d′ values for left and right hemifields were pooled for each of the 4 trial types, before performing KS tests and computing estimation statistics.

For these analyses, and those in Supplementary Fig. 3b, we used data from $n = 5$ participants (subject indices A1, A12, A13, A14, and A15, Supplementary Table 1) for whom the SSVEP on both sides exhibited high SNR on both sides and could, therefore, be reliably analyzed. Although 9 participants were tested with flickering stimuli on both sides, for 4 participants, the SSVEP on either the target or the distractor SSVEP DSS dimension exhibited low SNR (Supplementary Fig. 1d) precluding the reliable estimation of $\Delta\Phi$; therefore, data from the remaining 5 participants were included in these analyses. Data from 4 out of these 5 participants were used for target-triggered trials, as one participant (subject A1, Supplementary Table 1) was not tested with distractor side triggering.

**cBMI paradigm B.** The second paradigm (paradigm B) sought to identify changes in behavioral metrics accompanying large differences in SSVEP power across hemifields ($\Delta\Phi$). Task and training details were closely similar to paradigm A; key differences are indicated below.

Eleven healthy adult human participants (4 females; age range: 20–28 yrs; mean: 24.2 yrs, std dev: 2.9 yrs) with normal or corrected-to-normal vision and no known history of neurological disorders were included in this experiment. Participants gave written, informed consent prior to their participation in the experiment, and were monetarily compensated for their time. Two subjects who participated in paradigm A's experimental session, also participated in paradigm B's experiment (Supplementary Table 1; B10 and B11 were the same participants as A4 and A5). In the main text, we report results including only the $n = 9$ participants who were unique to paradigm B (Figs. 4, 5, main text). In the Supplementary Information, we report analyses including all participants in this cohort ($n = 11$, Supplementary Fig. 4c).

Task and stimulus configuration for Paradigm B was closely similar to paradigm A except for the following key differences. In the previous paradigm, participants had provided feedback that it was challenging to perform the task for prolonged durations due to the bright plaid pedestals and the overall bright (50% contrast) background. For this paradigm, therefore, we incorporated the following two changes. First, instead of flickering plaids as pedestals, we used flickering noise masks (uniform spatial noise, uncorrelated across pixels). The size and locations of these pedestals were identical to that of the previous paradigm. Second, the background contrast of the screen was reduced to 10%, with the screen refresh rate set at 100 Hz. Based on their feedback, participants considered this display much easier to view for prolonged durations. For this paradigm flicker frequencies were selected to be 14 Hz and 18 Hz for all participants, and, for each participant, were fixed for each hemifield (e.g., 14 Hz for the left hemifield pedestal and 18 Hz for the right hemifield pedestal). Flicker frequencies were counterbalanced between hemifields across participants.

In addition, because of the comparatively high d′-s observed in the previous paradigm, we made minor changes to the target and distractor stimuli to make their orientation discrimination itself more challenging. Target and distractor stimuli were 20% contrast Gabor gratings (std: 0.8 dva; spatial frequency: 1.5 cpd) embedded within the noise patch, and were displaced toward the outer edge of the pedestals along the horizontal meridian such that their centers were at $\pm6.6$ dva azimuth from the fixation cross. Both the neutral cue and response probe comprised of two isosceles triangles (0.6 dva base, 0.9 dva height) rotated and joined at the base (Fig. 4b), positioned 1.1 dva above the fixation cross. On each trial, the neutral cue comprised of two filled triangles, whereas the response probe comprised of a single filled triangle; the filled triangle indicated the side probed for response (Fig. 4b).

The task protocol was identical with that in paradigm A except for the following minor differences (Fig. 4b): (i) pedestals were presented for 500 ms before the cue onset; (ii) a neutral cue was presented on each trial above the fixation cross; (iii) the maximum cue-target interval was increased to 7.5 s; (iv) target and distractor stimuli were presented for 70 ms. These differences were driven by practical or task design considerations. For example, CTI increased because in pilot experiments, we observed that triggering stimuli based on extreme values of $\Delta\Phi$ was more challenging than that based on $\Phi$ alone. For example, mean trigger times for Paradigm A were significantly lesser than that for Paradigm B (Paradigm A: $2754 +/- 263$ ms; Paradigm B: $3410 +/- 383$ ms, $p < 0.001$, Mann Whitney U test) despite a comparable percentage of trials being triggered across both paradigms (Paradigm A: $68.49 +/- 14.38\%$; Paradigm B: $76.07 +/- 12.98\%$, $p = 0.1$, Mann Whitney U test). The pedestal duration was decreased, to keep the overall length of the trial shorter. Stimulus duration was marginally decreased, so as to have it similar to the previous paradigm while accommodating a 100 Hz screen refresh rate. Neutral cues were part of the task design for the present paradigm (see next). On each trial, the stimulus on one of the two hemifields was designated the target and the other designated the distractor. Neurofeedback was provided based on the difference of SSVEP power index across target and distractor hemifields ($\Delta\Phi = \Phi_{Target} - \Phi_{Distractor}$; see next). Response probes were equally likely to be the target side or distractor side; we term these the target-side probed and distractor-side probed trials, respectively. Participants were instructed to report the orientation of the grating on the response probed side. Distinct response keys were used for clockwise versus counterclockwise grating reports; this response mapping was counterbalanced across participants. In this paradigm, $5.5 \pm 1.4\%$ of trials were excluded due to poor gaze fixation.

Paradigm B comprised of 5 sessions spanning two days. On the first day, participants performed training and staircasing behavioral sessions. On the second day, participants performed a baseline EEG session, a $\Delta\Phi$ threshold session, and a neurofeedback session. The details for each session are described next:

The first session was the training session. Participants were trained on the behavioral task in Fig. 4b. The training protocol, without concurrent EEG recordings, was as described in paradigm A. Participants typically performed over 100–150 trials in the training session to familiarize themselves with the task.

The next session was staircasing. Participants performed the same task as in the training session. Staircasing was performed, as in paradigm A, except that in this case the orientation of stimuli was staircased to achieve an overall accuracy of 75%. Participants typically performed about 200 trials in the staircasing session.

Next session was the baseline EEG session. The goal of this session, as in paradigm A, was to obtain latent dimensions that maximized SSVEP power with DSS, and to estimate a distribution of SSVEP power for each flicker frequency, based on which the SSVEP power index ($\Phi$) was computed. The task in this session was the same as that in the staircasing session except that the orientation of target and distractor gratings was fixed to a specific value obtained from the staircasing session. Moreover, all trials were cued with 100% valid cues, counterbalanced

across the left and right hemifields. The goal of cueing was to obtain estimates of $\Phi$ when participants were attending to one side versus another (as in the actual neurofeedback paradigm, see next).

The baseline session was followed by $\Delta\Phi$ threshold session. In pilot neurofeedback experiments, we had observed considerable variability among subjects in their ability to trigger target and distractor stimuli based on uniformly high $\Delta\Phi$ thresholds. The goal of this session was, therefore, to obtain an approximate estimate of $\Delta\Phi$ thresholds for individual subjects to be used in the next neurofeedback session. The task for this session was identical with that for the baseline EEG session (cued orientation discrimination), except that this session comprised 32 trials with cues counterbalanced across left and right hemifields. With this paradigm, we constructed a distribution of $\Delta\Phi$ ($=\Phi_{Target} - \Phi_{Distractor}$) for SSVEPs and the threshold was set as the 98th percentile of the $\Delta\Phi$ distribution, subject to a minimum value of 0.75. The goal was to come up with an approximate value for $\Delta\Phi$ value that would require cognitive effort on the part of each participant, but not a value so high as to be impossible to achieve. In 4/11 participants, these thresholds had to be reduced slightly to make stimulus triggering less challenging. The distribution of the final thresholds used for all experiments across subjects, as well as the proportion of trials triggered across subjects, is plotted in Supplementary Fig. 4d, left and right.

The final session was the Neurofeedback cBMI session. The task in this session was the same as in the baseline EEG session above, except that no directed cues were presented. Rather, 500 ms after the onset of the pedestals, neutral cues (see section "cBMI paradigm B") were presented on each trial, which were uninformative about which side was the target for attention. In each trial, one of the two hemifields was pseudorandomly designated the target side, and the other side, as the distractor side. We tracked the difference of SSVEP power indices—across visual hemifields—between the target ($\Phi_{Target}$) and distractor ($\Phi_{Distractor}$) sides in real-time; we term this difference $\Delta\Phi$. When this difference exceeded a predetermined threshold, determined in the $\Delta\Phi$ threshold session, the presentation of both target and distractor stimuli were triggered.

Participants were not directly instructed (cued) regarding the target side. Instead, the participants received continuous auditory feedback whose frequency scaled linearly with $\Delta\Phi$ when this value was positive, and was fixed to a default value (500 Hz) when $\Delta\Phi$ was zero or negative. Specifically, we used the following piecewise linear transformation:

$$A_f = L_f + \lfloor \Delta\Phi \rfloor_+ (H_f - L_f) \qquad (7)$$

Where $\Delta\Phi = \Phi_{Target} - \Phi_{Distractor}$, where $L_f$ and $H_f$ represent the lowest (500 Hz) and highest (1500 Hz) possible frequencies provided as feedback, respectively and $\lfloor x \rfloor_+$ represents positive rectification. Auditory feedback was provided binaurally at a sampling rate of 44,100 Hz, in real-time using custom scripts based on MATLAB's audio toolbox. Participants were informed prior to the session that covert spatial attention would enable them to modulate the neurofeedback, but had to work out, by trial and error, which hemifield the feedback was based on. This resulted in this task being more challenging, and each trial being more prolonged than the previous paradigm. The maximum cue-target interval until trials were forcefully terminated (by stimulus presentation) was, therefore, increased to 7.5 s for this task. On each trial, feedback was terminated when $\Delta\Phi$ crossed the threshold.

For 3/9 participants presented in the main text, grating presentation occurred immediately upon $\Delta\Phi$ crossing threshold; for the remaining 6/9 participants, grating presentation occurred following a 60 ms delay of this threshold crossing. We introduced this delay for empirical reasons in the latter set of participants: to avoid stimulus onset from being coincident with the cessation of the auditory neurofeedback, because the latter terminated precisely at the time of threshold crossing. For both sets of participants, behavioral effects on normalized $d'$ were closely similar: there was a significant main effect of task condition (target-side probed versus distractor-side probed, $F(1,14 = 21.06, p = 0.0004$, ANOVA), but not of the timing ($p > 0.99$), nor an interaction effect ($p = 0.107$). A permutation test, by shuffling the timing label 1000 times across participants while retaining all other labels intact, also revealed no significant evidence for an effect of timing on $d'$ differences between target-side and distractor-side probed trials ($p = 0.169$). Consequently, we pooled together these two groups of subjects for subsequent analyses. Subjects typically ran 7 blocks of 64 trials each. For two subjects (B2, B5, Supplementary Table 1) the session was terminated after 5 blocks, owing to the subjects' inability to complete the experiment.

As in the baseline session, following target and distractor presentation, participants reported the orientation of the grating on side probed for response (Fig. 4b), within a fixed response window (2.0 s). For this paradigm, we compared behavioral metrics (reaction time, $d'$, and criterion) across target-side probed and distractor-side probed trials (see section on "Statistics and reproducibility"). In other words, we tested if inducing extreme differences in $\Phi$ across the visual hemifields would induce corresponding differences in behavioral metrics across these hemifields.

### Analysis of SSVEP power dynamics

We investigated how the dynamics of $\Phi$ (or $\Delta\Phi$) affected behavioral modulations of $d'$. Specifically, we asked whether the rate of change of $\Phi$ (or $\Delta\Phi$) immediately before threshold crossing would influence

sensitivity modulation—computed as the difference in discrimination accuracy ($\Delta d'$) at a location between trials for which the SSVEP power index was high at that location, versus trials for which the power index was high at the opposite location (Fig. 5c, f; Supplementary Fig. 5). The analysis below describes the procedure for measuring $\Phi$ dynamics; an identical procedure was used for measuring $\Delta\Phi$ dynamics also.

SSVEP power indices ($\Phi$) on the target and distractor side were estimated offline, using procedures described in an earlier section (see section on "Analysis of difference of SSVEP power across hemifields"). We considered 15 time-windows—ranging from 50 ms to 750 ms (in steps of 50 ms)—prior to threshold crossing. In each time-window, the difference between the $\Phi$ value at the time of threshold crossing and the minimum value of $\Phi$ within that (respective) window was computed. This provided an estimate of the rate of change of $\Phi$ (slope) prior to the threshold crossing at different timescales, ranging from 1.3 Hz (750 ms window) to about 20 Hz (50 ms window). For each participant, trials were divided into two categories: fast-ramping and slow-ramping trials, corresponding to the top-most and bottom-most quartiles (top 25% and bottom 25%), respectively, of the rates of change of $\Phi$. $d'$ was estimated separately for these fast- and slow-ramping trials, for all participants. For this analysis, we used data from $n = 9$ subjects who participated in Paradigm B alone.

### Modeling interventional versus correlational approaches

We present 4 directed graphical models—each of which reflects a distinct, possible mechanism for attention's effect on $d'$ and RT modulation (Supplementary Fig. 2a). In these models, A represents the cognitive process of attention, $\Phi$ represents SSVEP power modulation, M and N represent alternative neural processes (e.g., alpha or beta oscillations), distinct from SSVEP generating mechanisms, and $d'$ and RT reflect behavioral effects on sensitivity and reaction times, respectively. Note that, in general, each neural process can be engaged by multiple cognitive processes other than attention (e.g., arousal) and can also be susceptible to internal, neural noise. Similarly, neural processes, other than those shown here, could influence each behavioral metric.

Each of these 4 models differs in the way it links SSVEP power ($\Phi$), as well as the other neural processes, to the behavioral metrics of $d'$ and RT. In the first model, SSVEP power changes do not directly influence either $d'$ or RT. In the second and third models, SSVEP power changes drive RT and $d'$ effects, respectively. In the fourth model, SSVEP power changes directly affect both $d'$ and RT. For ease of reference, we call these the "no effect" model, the "RT effect" model, the "$d'$ effect" model, and the "both effects" model.

These 4 models cannot be distinguished with correlational approaches alone. In conventional tasks that employ correlational approaches, the common source process (A) represents the strongest source of common variation among all of the neural processes (M, N, $\Phi$) and, thereby, the behavioral metrics ($d'$, RT). As a result, correlations occur between neural processes and behavioral metric that are causally influenced by attention. These are the correlations, for example, between $\Phi$ and $d'$ as well as between $\Phi$ and RT, that previous studies have reported[15,17]. We demonstrate this result with simulations based on a simple mathematical formulation in Supplementary Fig. 2b–e, subpanel ii.

In an interventional approach, SSVEP power ($\Phi$) is forced to be at fixed, predetermined high or low values on each trial. If a change in $\Phi$ were sufficient to induce a change in a behavioral metric ($d'$ or RT), we would expect to observe an obligatory change in the respective behavioral metric value across high-$\Phi$ and low-$\Phi$ trials. The specific behavioral metric affected, then, enables us to distinguish between the 4 models. Again, we demonstrate this result with simulations in Supplementary Fig. 2b–e, subpanel iii.

Although in our experiments, we did not have the ability to directly intervene on SSVEP power and set it to arbitrary values, we followed an indirect approach to achieve interventional control. We triggered stimulus presentation when $\Phi$ (Paradigm A) or $\Delta\Phi$ (Paradigm B) reached specific, predetermined high or low values. Because of the causal relationship between the source process A (e.g., attention), and neural processes (M, N, $\Phi$) (Supplementary Fig. 2b–e, subpanel i), fixing $\Phi$ to a particular value does not guarantee that A is also fixed to a specific value. Consequently, other neural processes, like M and N, are free to vary even if $\Phi$ is at a fixed value at the time of grating presentation. By this intervention on stimulus timing our experiment examines the effect of decoupling fluctuations in $\Phi$ from fluctuations in the other neural processes on behavioral metrics. Our results support the $d'$ effect model—changes in $\Phi$ produce obligatory changes in $d'$, but not in neural process N and, consequently, not in RT.

### Comparison of online triggering of stimuli versus post hoc analyses

We could have tested for the effect of SSVEP power on behavior using an offline approach: using a conventional task design and post hoc analyses, as has been done in previous studies[15–17]. We have discussed key differences between these offline approaches and our online (real-time) cBMI approach in the "Discussion". Nevertheless, we seek to compare here the number of trials that would need to be collected, had we used the offline approach, to achieve the same level of difference in SSVEP power index across high-$\Phi$ or low-$\Phi$ trials in paradigm A or the same level of SSVEP power index difference across hemifields ($\Delta\Phi$). In the real-time triggering task, $68.5 \pm 3.7\%$ and $76.1 \pm 4.3\%$ of trials were triggered in paradigm A and B, respectively, and used for the subsequent behavioral analysis. To estimate

the number of trials that would have crossed the threshold in the last 50 ms of the trial (same in real-time triggering experiment) in a conventional task, we considered the baseline block for both paradigms. For each paradigm. We simulated 10,000 trials with trial intervals randomly sampled from an exponential distribution (mean = 1 s, with trial times of 1.5–4 s for paradigm A and 1–8 s for paradigm B) and computed the $\Phi$ (or $\Delta\Phi$) traces by sampling trials, with replacement, in the baseline block of the respective paradigm. For each condition in paradigm A (target- or distractor-side triggering x high-$\Phi$ versus low-$\Phi$) and for each condition in paradigm B (target- or distractor-side probe), we calculated how many trials crossed the threshold in the last 50 ms of each simulated trial. For paradigm A, $\Phi$ crossed the high threshold on the target- and distractor-sides in $5.2 \pm 1.2\%$ and $3.3 \pm 0.6\%$ of the simulated trials, respectively, whereas $\Phi$ crossed the low threshold on the target- and distractor-sides in $7.5 \pm 1.2\%$ and $7.4 \pm 1.0\%$ of the simulated trials, respectively. For paradigm B, $\Delta\Phi$ crossed the threshold for the target-side and distractor-side probed trials on $1.2 \pm 0.3\%$ and $1.0 \pm 0.2\%$ trials, respectively. Thus, to achieve 68.5% and 76.1% triggering in paradigm A and B, respectively, we estimate that, on average, ~11-fold more trials for paradigm A (~4000 trials) and ~69-fold more trials for paradigm B (~30,000 trials) are necessary.

**Statistics and reproducibility**. Two-way ANOVAs were used to compare behavioral (psychometric and psychophysical) parameters for target and distractor-triggered trials in Fig. 2b, c. AVOVAs were performed with behavioral measures (d′, criterion, and RT) as dependent variables and $\Phi$ levels (high-$\Phi$ and low-$\Phi$) and hemisphere (left and right) as the two independent factors. A non-parametric Wilcoxon paired signed-rank test was employed for the following analyses: (i) SSVEP power index ($\Phi$) between cued and uncued conditions (Fig. 2a, inset), (ii) median cue-target intervals between high-$\Phi$ and low-$\Phi$ trials (Supplementary Fig. 3a), (iii) mean $\Phi$ on the distractor side between high-$\Phi$ and low-$\Phi$ target triggered trials (Supplementary Fig. 3b), (iv) $\Delta$d′ between fast-ramping versus slow-ramping trials (Fig. 5c, d, f, g and Supplementary Fig. 5c, d). Pearson correlations were used to compute correlation coefficients for the following analyses: (i) correlations between simulated $\Phi$ and behavioral metrics (d′ and RT) for correlational and interventional approaches (Supplementary Fig. 2a–d), (ii) correlations between target and distractor $\Phi$ values (Supplementary Figs. 3c and 4b). Kolmogorov-Smirnov tests were used to test for differences between $\Delta\Phi$ distributions (Fig. 3b, insets). A 4-way ANOVA was performed (Fig. 3b) with d′ as the dependent variable and $\Phi$ levels (high-$\Phi$ and low-$\Phi$), $\Delta\Phi$ levels, trigger type (target vs distractor), and hemisphere (left and right) as independent factors. Estimation statistic analog of paired t-test, statistical method which computes effect size and tests for its significant difference from zero[79], were used to test differences in behavioral metrics (d′, criterion, and RT) between target and distractor probed trials (Fig. 4e and Supplementary Fig. 4c). Significance levels of the p-values (asterisks) follow the convention: *$p < 0.05$, **$p < 0.01$, ***$p < 0.001$. Sample size estimation is described in the Methods section titled "Sample size estimation". In addition, behavioral differences were analyzed using the Bayes Factor (BF). BF for the one-tailed t-test was computed using JZW priors[80]. A BF of 3 (10) or higher represents substantial (strong) evidence in favor of the effect of interest. All the statistical tests in the main text and their results are summarized in Supplementary Table 2. We conducted two sets of experiments; key results from paradigm A were replicated in paradigm B.

**Reporting summary**. Further information on research design is available in the Nature Portfolio Reporting Summary linked to this article.

## Data availability

All data necessary for reproducing all figures in the paper have been deposited into an opensource online repository[81] for ready inspection and replication of the results. Moreover, source data for the graphs in the main text are also provided in an excel file as Supplementary Data 1. The source data (raw EEG) will be made available upon request to the corresponding author.

## Code availability

All code necessary for reproducing all figures in the paper have been deposited into an opensource online repository[81] for ready inspection and replication of the results.

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

## Acknowledgements

This research was supported by a CMU-IISc BrainHub grant (jointly to D.S. and B.Y.), the following grants to D.S.—Wellcome Trust-Department of Biotechnology India Alliance Intermediate fellowship [IA/I/15/2/502089], Pratiksha Trust award [FG/SMCH-19-2047], India-Trento Programme for Advanced Research (ITPAR) grant, Department of Biotechnology-Indian Institute of Science Partnership Program grant, Tata Trusts grant—and the following grants to B.M.Y.—NSF NCS BCS 1533672 and 1734916, NIH R01 HD071686, NIH CRCNS R01 NS105318, NIH CRCNS R01 MH118929, NIH R01 EB026953, and Simons Collaboration on the Global Brain 543065. D.S. wishes to acknowledge a DRDO grant [SR/CSI/44/2008] awarded to Prof. Veni Madhavan and a IRHPA grant [IR/SO/LR-002/2006] awarded to Profs. Vijayalakshmi Ravindranath and Govindan Rangarajan, that funded key instrumentation for this project.

## Author contributions

D.S. and B.M.Y. conceived and designed the study; A.C. and S.G. designed the EEG cBMI system; A.C. and S.P. conducted the experiments for paradigms A and B, respectively; A.C., S.P., and V.C. analyzed the data; D.S. wrote the manuscript with inputs from all authors.

## Competing interests

Devarajan Sridharan is a Research Consultant at Google. The other authors declare no competing interests.
