## [Peer Review File · Communications Biology]

Reviewers' comments:

Reviewer #1 (Remarks to the Author):

This study investigates whether and how fluctuations of brain activity as indexed by momentary changes of SSVEP power influence behavioral performance parameters reflecting the operation of visual attention. To this end, the authors develop, what they call, a cognitive brain machine interface (cBMI), that continuously tracked momentary SSVEP-power elicited by flickering plaids (pedestrians) in the left and right visual. When SSVEP-power (expressed as an index Φ) exceeds or undershoots certain thresholds, trials containing an attention-directing arrow-cue followed by a target and a distractor (oriented gratings) are triggered. This allows the authors to analyze target discrimination performance (accuracy, RT, d' , choice criterion) as a function of momentary SSVEP-power. Data of two experimental paradigms (A,B) are reported, both containing a training session, a staircase session to determine target contrast for 70% response accuracy, a baseline session to find strongest SSVEP response and to determine baseline SSVEP power, and finally a real-time cBMI session. Paradigm B adds a neurofeedback cBMI session, where observers are not cued to the target side but receive auditory feedback reflecting the momentary power-difference between hemifields ($\Delta\Phi$), requiring them to figure out the target side by themselves.

It is reported that a higher SSVEP power at the target side is associated with higher accuracy of target discrimination. RT, however, is not affected. Accuracy is also higher with increasing $\Delta\Phi$, which the authors suggest reflects cross-hemispheric competitive interactions underlying endogenous focusing. Overall, it is concluded that the cBMI approach reported here allows for a successful read-out of brain states underlying attention.

This is a very interesting and innovative study that aims at establishing a link between brain-electric activity and attention via an interventional approach that goes beyond the typical limitations of correlational analyses. The experiments are carefully designed and the data analysis is appropriate and well performed. Overall, I find the observation that momentary increases of SSVEP-power predict better target discrimination convincing, and I think the manuscript demonstrates the substantial potential of the SSVEP-based cBMI approach for investigating neural mechanisms of visual attention very clearly. Nonetheless, as detailed below, I have some issues with the authors' interpretations in terms of competitive interactions as well as with some features of the data.

Issues:

(1) The conclusion about competitive neural dynamics is not entirely borne out by the data. Results of the hemifield-selective analyses (Paradigm A) yield a clear link between target- Φ and accuracy. However, at the same time, distractor- Φ doesn't vary with the target- Φ in opposite direction (see analysis at p. 13/14 lines 252-258), suggesting that the SSVEP fluctuation of the distractor is independent of that of the target. The across-hemifield analysis ($\Delta\Phi$) is therefore equivalent to the hemifield-selective analysis, with $\Delta\Phi$ being driven by the increase of target- Φ . More critically, the absence of such opposite-direction dependency in paradigm A suggests that the size of $\Delta\Phi$ does not really reflect competition between hemifields as suggested by the authors (Discussion at p. 28, lines 570-572, Abstract). For this to be the case, the distractor- Φ should reduce when target- Φ increases. In paradigm B the authors decide to exclusively analyze $\Delta\Phi$ without considering Φ independently for the target and distractor. However, it remains unclear whether $\Delta\Phi$ in paradigm B is indicative of competition because the observed pattern of results can arise from a SSVEP fluctuation solely on the target side. The authors should provide a hemifield-selective analysis for the paradigm B data and show that Φ varies in opposite direction for the target and distractor side. If not, the authors should take back their interpretations in terms of competitive interactions.

(2) Ramping analysis at p.23 and Figure 5 is somewhat unclear. The ramping effect of Φ shown in Figure 5B is literally identical for the target and the distractor. Where does the $\Delta\Phi$ in Figure 5E come from? I guess, the reader needs some more explanation here.

Also, I am not sure whether slow versus fast ramping is the appropriate characterization of the response pattern. There is a peculiar and much more prominent difference between slow and fast ramping trials. The latter shows an initially negative slope ramping in the opposite direction to the slow ramping trials, suggesting some sort of inconsistent ramping on those trials. This could mean that fast ramping trials Φ initially show a relative increase on the distractor side, or alternatively that Φ decreases first on the target relative to the distractor side. Both patterns would eventually account for a lesser d' , but the fast versus slow distinction is somewhat missing the point. Finally, at the time of grating presentation, $\Delta\Phi$ is the same for fast and slow ramping trials. If $\Delta\Phi$ predicts performance, shouldn't there be no performance difference between FR and SR, contrary to what is shown in Figure 5F?

(3) The reasoning regarding the absence of RT effects in the discussion on p. 29 is not entirely plausible to me. In the present experiments, fluctuations of Φ were found to link with accuracy but not with RT. Other studies using correlational approaches, however, have reported effects on both accuracy and RT. To account for the absence of RT differences in the present study, the authors suggest that attention engages distinct mechanisms (all reflected in SSVEP power fluctuations), causing distinct effects on d' and RT. Because attention engages those mechanisms in a highly correlated manner, correlational approaches may not be able to separate their effects and therefore see effects on d' and RT all the time. But is not clear how this would explain why RT effects are not seen here. If attention employs these mechanisms in a highly correlated manner, their effect should play out in interventional approaches as well.

(4) Figure 2 A: The response to the pedestrian is bigger for subsequently cued versus uncued trials even before the cue. It is not clear whether the time range before cue onset was subjected to statistical validation? If the difference turns out to be significant, it would pose a problem for data interpretation. The authors should comment on this in the manuscript.

Reviewer #2 (Remarks to the Author):

Tracking momentary fluctuations in human attention with an "interventional" cognitive brain-machine interface

In a study based on two experiments, the authors examined the effects of spatial attention on behavior. They used a novel brain-machine-interface approach that online extracted SSVEP amplitudes for left and right presented stimuli as an ongoing marker of concurrently deployed spatial attention and triggered the presentation of task-relevant, tilted gratings during high-SSVEP-amplitude (i.e. high attention) or low-amplitude (i.e. low attention) states. The authors present results suggesting a better (but not faster) detection of the orientation of the target grating when stimuli are presented during high-amplitude states as compared to low-amplitude states.

While I found the general aim of the study quite exciting and found many aspects of it thoughtful and well implemented and the figures very pleasing, there are quite a few aspects that hinder the interpretation of the robustness and validity of the findings.

One major aspect concerns the experimental parameters of the studies. The authors reported of many aspects of the study that were changed during the course of the data acquisition and thus differed across uneven subgroups of participants. Given the low number of participants, such changes and their impact on potential results were not evaluated and cannot be accounted for.

- In experiment 1: for many participants left and right stimuli were presented differently (13 and 15 Hz vs 25 Hz or static presentation) than the intended presentation with 14 and 18 Hz for left and right or right and left stimuli. These changes overall create physically and perceptually different conditions

across participants. Especially the overall luminance of static stimuli is different as compared to flickering stimuli creating potential differences in the spatial biasing of attention due to luminance differences for left and right stimuli.

- Stemming from the different stimulation frequencies: not all subjects contribute equally to the data plotted in figure 2 A. Data with presumably differing SNR is entered, depending on the stimulation regime and thus on whether SSVEP time courses could be extracted for cued and uncued SSVEPs simultaneously or during different trials only or for the cued stimulus only.
- In addition, for 5 out of 15 participants of experiment 1, the stimulus presentation was triggered by the SSVEP amplitude for the cue side only, while no trials were present for which the stimulus presentation was triggered by the distractor SSVEP amplitude. This should affect the total trial number analyzed (and thus the SNR) or the temporal structure of the experiment.
- As described in line 888 onwards: the ratio of high- Φ and low- Φ thresholds was altered across subjects. Why was it done, if it had no effect, as stated in line 890 and how was a potential difference tested?
- In experiment 2 the timing of the grating presentation was changed for some subjects. As a side note: the authors mention, potential differences were tested here but don't report the nature of these tests.

Related to the point above: there are text passages that imply consistent experimental parameters for the whole cohort, while in fact, this is not the case. Or it is not clear how missing or different data from subjects with different parameters are treated. This is misleading:

- line 148: "two flickering pedestals appeared"; not true for all subjects
- line 150: "each flickered at a distinct frequency": not true for all subjects
- line 198/199 "Target (and distractor) presentation was triggered": not true for all subjects
- line 254: "Mean Φ on the distractor side was not significantly different across the target high- Φ and low- Φ ": Calculation not for all subjects possible. Here it is quite critical as the statistics calculated, was calculated only for a subset of participants but generalized to the effects of the whole experiment.
- Figure 2: A "pooled across all subjects (n=15)" not the same for all subjects
- Figure 3: Parts of the figure (A, B) are based on data from 5 subjects and some for 10. Why is this the case?

Concerning the discussion of the findings of both experiments, the author claim that a d' modulation is measurable when target stimuli presentation is based on the differences between SSVEPs in both hemisphere (hinting at some global competition) while this is not the case when stimuli are triggered by different SSVEP levels of the distractor.

While this would be a very interesting finding, this claim can not easily be made given the data:

- The crucial test for this claim is not available in the experiment/results: while for experiment 1 the impact of a distractor can be evaluated for different levels of distractor SSVEP amplitudes, this crucial contrast is not possible in experiment 2 and is also not presented in figure 4 for instance. Here, if I understood correctly, the stimulus presentation was triggered when the SSVEP lateralization reached an upper threshold and stimuli were either presented on the side with the high SSVEP amplitude (target-cued trials) or on the other side (distractor-cued trials). Hence for distractor processing in experiment 2, there is no statement possible in whether it varies according to the level of SSVEP lateralization. Most crucially in the second experiment, the SSVEP lateralization triggering the stimulus presentation was not manipulated orthogonally to the attentional focus, hindering the examination of crucial contrast of interest.
- There are other differences in the design that could also be responsible for potential effects:
- different stimuli, a different way of having participants shift their attention (top-down cue vs ongoing auditory feedback), different lengths of trials
- I was also wondering how a pattern of "extreme, global differences in SSVEP power across hemifields" [line 413] could be differentiated from a pattern with an upregulated SSVEP amplitude for the attended side and a non-regulated SSVEP amplitude for the unattended side (more capturing the findings from experiment 1).

Another aspect concerns the statistical analyses and interpretations based thereon:

- For the analysis of potential differences of behavioral parameters between the high and low SSVEP state via Wilcoxon signed-rank test (as displayed in figure 2) the values from left and right hemisphere-derived data are treated as independent samples. This, however, violates the test's assumption that all paired observations are drawn randomly and independently as they stem from the same subject and I have serious doubts that this is a valid procedure. Given that the main finding is not significant when not relying on this procedure (as seen in SI Figure S2), renders the main message of the manuscript shaky. As a side note: why is N = 26, shouldn't it be 30?
- A similar procedure for the statistical analysis is shown in figure 3C
- Given the differences between both experiments (see above), I'm also unsure whether pooling data from both experiments for the slope analysis across time bins is a good idea.
- Throughout the absence of a significant difference is treated and interpreted as the absence of a significant effect. Null-effects in a conventional significance testing do, however, not provide actual evidence for the null hypothesis (in contrast for instance to Bayesian analysis approaches, see Rouder et al., 2009) as the design may just be underpowered to detect effects below a specific effect size. In general, using a Bayesian framework (or other likelihood approaches) for the statistical tests could help to substantiate whether the null findings support the null hypothesis or whether evidence is pointing in neither direction of the null- or the alternative hypothesis.

One aspect that may be of interest for the interpretation of the data: were the target stimuli for all different conditions (cued high vs cued low vs uncued high vs uncued low) presented at comparable time points throughout each trial?

Minors:

- page 4 line 46: it is not mentioned for which measure the Fano Factor decreased
- page 9 line 153: what is a brief interval?
- page 10 line 179 onwards: while I think it's valid to say that data seems to be transferred and processed near real-time with neglectable delay, the SSVEP power calculation relies on the integration of data from a time window of 500 ms, if I understood correctly, and is thus delayed.
- Figure 3: What does "PDF" stand for?
- Discussion line 592 onwards: I don't get why the cBMI approach does not suffer from multiple mechanisms engaged in creating behavior. This point could be elaborated.
- line 654: What happened to the noisy electrodes?
- line 706: What is meant by clockwise and counterclockwise orientations. I only know this term from describing motion directions.
- line 764: participants with poor SNR are still in the dataset?
- line 926: How was this analysis window derived?
- line 1001: What does "challenging" mean?
- line 1050: What does "had to be relaxed" mean?
- SI Figure S2 D: why are there only 8 green data points instead of 10?

###

References:

Rouder, J. N., Speckman, P. L., Sun, D., Morey, R. D., & Iverson, G. (2009). Bayesian t tests for accepting and rejecting the null hypothesis. *Psychonomic Bulletin & Review*, 16(2), 225–237.
<https://doi.org/10.3758/PBR.16.2.225>

Reviewers' comments:

Reviewer #1 (Remarks to the Author):

This study investigates whether and how fluctuations of brain activity as indexed by momentary changes of SSVEP power influence behavioral performance parameters reflecting the operation of visual attention. To this end, the authors develop, what they call, a cognitive brain machine interface (cBMI), that continuously tracked momentary SSVEP-power elicited by flickering plaids (pedestrians) in the left and right visual. When SSVEP-power (expressed as an index Φ) exceeds or undershoots certain thresholds, trials containing an attention-directing arrow-cue followed by a target and a distractor (oriented gratings) are triggered. This allows the authors to analyze target discrimination performance (accuracy, RT, d' , choice criterion) as a function of momentary SSVEP-power. Data of two experimental paradigms (A,B) are reported, both containing a training session, a staircase session to determine target contrast for 70% response accuracy, a baseline session to find strongest SSVEP response and to determine baseline SSVEP power, and finally a real-time cBMI session. Paradigm B adds a neurofeedback cBMI session, where observers are not cued to the target side but receive auditory feedback reflecting the momentary power-difference between hemifields ($\Delta\Phi$), requiring them to figure out the target side by themselves. It is reported that a higher SSVEP power at the target side is associated with higher accuracy of target discrimination. RT, however, is not affected. Accuracy is also higher with increasing $\Delta\Phi$, which the authors suggest reflects cross-hemispheric competitive interactions underlying endogenous focusing. Overall, it is concluded that the cBMI approach reported here allows for a successful read-out of brain states underlying attention.

This is a very interesting and innovative study that aims at establishing a link between brain-electric activity and attention via an interventional approach that goes beyond the typical limitations of correlational analyses. The experiments are carefully designed and the data analysis is appropriate and well performed. Overall, I find the observation that momentary increases of SSVEP-power predict better target discrimination convincing, and I think the manuscript demonstrates the substantial potential of the SSVEP-based cBMI approach for investigating neural mechanisms of visual attention very clearly.

Thank you.

Nonetheless, as detailed below, I have some issues with the authors' interpretations in terms of competitive interactions as well as with some features of the data.

We address each issue, point-by-point, below.

Issues:

(1) The conclusion about competitive neural dynamics is not entirely borne out by the data. Results of the hemifield-selective analyses (Paradigm A) yield a clear link between target- Φ and accuracy. However, at the same time, distractor- Φ doesn't vary with the target- Φ in opposite

direction (see analysis at p. 13/14 lines 252-258), suggesting that the SSVEP fluctuation of the distractor is independent of that of the target. The across-hemifield analysis ($\Delta\Phi$) is therefore equivalent to the hemifield-selective analysis, with $\Delta\Phi$ being driven by the increase of target- Φ . More critically, the absence of such opposite-direction dependency in paradigm A suggests that the size of $\Delta\Phi$ does not really reflect competition between hemifields as suggested by the authors (Discussion at p. 28, lines 570-572, Abstract). For this to be the case, the distractor- Φ should reduce when target- Φ increases.

The reviewer is correct. We seek to provide two clarifications on this point.

First, the results of paradigm A indeed show that, by default, target- Φ and distractor- Φ are uncorrelated; an increase (decrease) in target- Φ does not result in an obligatory decrease (increase) in distractor- Φ . We show in revised Figure 3A (main text) and SI Figure S3C, below. In addition, trial-wise analysis revealed no evidence for correlation between target- Φ and distractor- Φ either in the high- Φ trial ($r=-0.01$, $p=0.90$) or in the low- Φ trials ($r=0.06$, $p=0.13$).

Revised Figure 3A (left) and SI Figure S3C (right)

These results are consistent with the interpretation that, by default, attentional resources in the two visual hemifields (and brain hemispheres) are independent and do not compete. We further elaborate on these results and discuss them in the revised manuscript (see next).

Second, to address the point raised by the reviewer, we have performed a new analysis. Unlike in conventional behavioral tasks, in our cBMI Paradigm A variations in $\Delta\Phi$ (difference in Φ across hemifields) occur primarily due to variations in Φ on the non-triggered side; this is because the Φ on the triggered side was set to fixed, threshold values. For example, on “target-

side triggered” trials, target- Φ reached a preset, fixed threshold at the time of stimulus presentation. Therefore, for each trial type (High- Φ or Low- Φ), $\Delta\Phi$ variations occurred largely due to changes in Φ on the distractor side alone, for that trial type. Conversely, on “distractor-side triggered” trials, distractor- Φ reached a preset fixed, threshold at the time of stimulus presentation, and $\Delta\Phi$ variations occurred largely due to changes in Φ on the target side alone (revised Fig. 3B).

With this understanding, we repeated the analysis shown in Figure 3 by dividing trials into extreme values of $+\Delta\Phi$ and $-\Delta\Phi$ based on a median split, separately, for each trial type (target-triggered, High- Φ ; target-triggered, Low- Φ ; distractor-triggered, High- Φ ; distractor-triggered, Low- Φ). We observed no systematic effect of $\Delta\Phi$ on d' variation $\Delta\Phi$ ($F(1,56) = 0.85$, $p=0.360$, ANOVA). These results indicate that, in a standard cued attention setting, there is no evidence for competition between SSVEPs or attentional resources across the two hemispheres.

Revised Figure 3B

In paradigm B the authors decide to exclusively analyze $\Delta\Phi$ without considering Φ independently for the target and distractor. However, it remains unclear whether $\Delta\Phi$ in paradigm B is indicative of competition because the observed pattern of results can arise from a SSVEP fluctuation solely on the target side.

The authors should provide a hemifield-selective analysis for the paradigm B data and show that Φ varies in opposite direction for the target and distractor side. If not, the authors should take back their interpretations in terms of competitive interactions.

Thank you for this suggestion. We have now performed this analysis for Paradigm B data also. In this paradigm, we observe indeed that target- Φ and distractor- Φ vary in opposite directions over time. We show this in revised Figure 4D and SI Figure S4B, below. In addition, trial-wise analysis revealed clear evidence for anti-correlation between the target- Φ and distractor- Φ ($r = -0.37$, $p < 0.001$).

In other words – when subjects sought to achieve large differences between target- Φ and distractor- Φ , based on auditory neurofeedback – competitive interactions between the hemifields were indeed generated.

Revised Figure 4D (left) and SI Figure S4B (right)

We have heavily amended the appropriate Results section to reflect these findings, as follows, when discussing Paradigm A:

“Local SSVEP fluctuations rather than global SSVEP power differences correlate with accuracy effects

There is some debate in the literature about whether attention resources across the two visual hemifields are independent^{31,32}. While some studies suggest that sensory processing resources for the attended hemifield (e.g., as indexed by target SSVEP power) are allocated independently of the unattended hemifield (e.g., as indexed by distractor SSVEP power)^{31,33}, other studies suggest that attention produces a global competition for sensory resources across hemifields^{32,34}. While these hypotheses are not mutually exclusive, the results of Paradigm A support the former hypothesis: distractor side Φ states did not systematically predict target d' variations. In other words, the behavioral effects on d' were mediated, at least in part, by local, not global, SSVEP power fluctuations. To further confirm this hypothesis, we performed two additional analyses.

First, we plotted the Φ traces, separately for the target and the distractor sides, for the high- Φ and low- Φ target SSVEP triggered trials (Fig. 3A; $n=5$ subjects with flickering stimuli and high SNR on both sides, Methods). We observed no systematic modulation of mean Φ on the

distractor side across the target high- Φ and low- Φ trials (distractor Φ value: high- Φ trials = 0.53 ± 0.06 , low- Φ trials = 0.51 ± 0.07 , $p=0.125$, Wilcoxon signed-rank test; $BF=1.36$) (SI Fig. S3B). In addition, trial-wise analysis revealed no evidence for correlation between target- Φ and distractor- Φ either in the high- Φ trials ($r=-0.01$, $p=0.90$) or in the low- Φ trials ($r=0.06$, $p=0.13$) (Methods, section on *Trial-wise analysis of correlation between target- Φ and distractor- Φ* ; see also trial-averaged correlations shown in SI Fig. S3C).

Second, we tested if target discrimination accuracy would vary with the strength of global competition between target and distractor representations, at a fixed level of target (or distractor) SSVEP power. For this, we computed the difference between the SSVEP power indices across hemifields ($\Delta\Phi = \Phi_{\text{target}} - \Phi_{\text{distractor}}$), just before stimulus presentation (at the time of threshold crossing), separately for the target triggered and distractor triggered trials, and separately for the high- Φ and low- Φ trials (Fig. 3B). For each trial type, we divided the trials into two subsets, based on a median split (Methods section on *Analysis of difference of SSVEP power across hemifields*). For the target triggered high- Φ trials and the distractor triggered low- Φ trials, the first subset of trials (values above the median) comprised trials typically with large positive values of $\Delta\Phi$ (“extreme + $\Delta\Phi$ ”) whereas the second subset (values below the median) comprised of $\Delta\Phi$ values around zero (“moderate $\Delta\Phi$ ”) (Fig. 3B, upper-left and lower-right respectively). Conversely, for the target triggered low- Φ trials and the distractor triggered high- Φ trials, the first subset of trials (values above the median) comprised of $\Delta\Phi$ values around zero (“moderate $\Delta\Phi$ ”) whereas the second subset (values below the median) comprised typically with large negative values of $\Delta\Phi$ (“extreme - $\Delta\Phi$ ”) (Fig. 3B, upper-right and lower-left respectively). We confirmed that the $\Delta\Phi$ distributions for every pair of subsets was significantly different from each other (Fig. 3B, insets; $p<0.001$, for 4 pairwise comparisons, Kolmogorov-Smirnov test). We then tested if d' modulations would co-vary systematically with $\Delta\Phi$ for each trial type.

Accuracies did not vary systematically with $\Delta\Phi$ across the first and second subsets, both when the data were analyzed together ($F(1,56) = 0.85$, $p=0.360$, ANOVA, $n=5$ subjects for target triggered trials, and $n=4$ for distractor triggered trials), and in *post hoc* comparisons, when each of the four trial types were compared separately ($p>0.05$, estimation statistics).

In other words, modulations of target SSVEP power were not accompanied by congruent modulations of distractor SSVEP power in the opposite hemifield, a result consistent with earlier studies^{34,35}. Moreover, once local SSVEP power was fixed at specific (threshold) values, d' did not modulate with global differences in SSVEP power across the visual hemifields. In other words, sensory processing for the stimulus in the attended hemifield appears to be, at least, in part, independent of processes occurring in the opposite hemifield.”

And, as follows, when discussing Paradigm B:

“First, we tested if this neurofeedback paradigm induced competitive interactions across the visual hemifields. Indeed, in this paradigm, target- Φ and distractor- Φ varied in opposite directions – at trend clearly visible in trial-average traces over time (Fig. 4D, lower). In addition, trial-wise analysis revealed clear evidence for anti-correlation between the target- Φ and

distractor- Φ (SI Fig. S4B) ($r = -0.37$, $p < 0.001$; Methods, section on *Trial-wise analysis of correlation between target- Φ and distractor- Φ*). In other words – when subjects sought to achieve large differences between target- Φ and distractor- Φ , based on auditory neurofeedback – robust competitive interactions were generated between the neural representations in each hemifield.”

(2) Ramping analysis at p.23 and Figure 5 is somewhat unclear. The ramping effect of Φ shown in Figure 5B is literally identical for the target and the distractor. Where does the $\Delta\Phi$ in Figure 5E come from? I guess, the reader needs some more explanation here.

We regret the lack of clarity with the presentation.

We clarify that in the original submission’s Figure 5B, we had presented the ramping effect of Φ for two distinct sets of trials in the left and right panels. In the left panel, we showed Φ dynamics for the pedestal on the side of the stimulus that was subsequently probed (“probed” stimulus). In the right panel, we showed Φ dynamics for the pedestal on the side opposite the probed stimulus (“unprobed” stimulus). As a result, the dynamics were closely similar because they show the build-up of SSVEP power either on the side of the probed stimulus, or on the side opposite to it, on distinct sets of trials. Φ dynamics from the other side were not shown in the earlier version of the figure, to avoid clutter – but we have addressed this in the revision.

In the revision – to avoid this potentially confusing description – rather than combining data from the two paradigms we now show this figure with only data from Paradigm B (also in line with a suggestion from Reviewer #2). All statistics are based on this paradigm alone. Therefore, in the revised manuscript, the left panel of Figure 5B shows target- Φ dynamics from *target-side probed* trials of Paradigm B whereas the right panel shows distractor- Φ dynamics from *distractor-side probed trials* of Paradigm B. Corresponding dynamics for the complementary set of trials are shown in SI Figure S5A.

Similarly, in Figure 5E, the left panel now shows $\Delta\Phi$ ($\Phi_{\text{target}} - \Phi_{\text{distractor}}$) dynamics for *target-side probed* trials (left) and the right panel shows $-\Delta\Phi$ ($\Phi_{\text{distractor}} - \Phi_{\text{target}}$) dynamics for *distractor-side probed* trials (right).

Revised Figure 5

Figure 5

Revised SI Figure S5

SI Figure S5

We have revised the text to more clearly describe these results, as follows:

“First, we divided trials into two categories – “fast-ramp” versus “slow-ramp” – based on the rate of change of Φ , just before threshold crossing, across trials of Paradigm B (Methods). The average Φ traces for each of these sets of trials are shown in Figure 5B (solid lines: slow ramp; dashed lines: fast ramp); the left panel shows target- Φ dynamics from target-side probed trials whereas the right panel shows distractor- Φ dynamics from distractor-side probed trials (for the converse dynamics, see SI Fig. S5A). We then computed the difference in d' – $\Delta d'$ – between these two sets of trials. Note that this quantity, $\Delta d'$, measures the difference in d' for the probed stimulus between trials when the SSVEP power index was high, compared to when it was low, at the probed location. We tested if this d' modulation was different between fast-ramp and slow-ramp trials.

Interestingly, when we divided trials based on ramping dynamics of Φ , we observed no significant differences between the two categories (Fig. 5C, $\Delta d'$: fast-ramp $\Phi = 0.43 \pm 0.54$, slow-ramp $\Phi = 0.40 \pm 0.69$, $p=0.496$, Wilcoxon paired signrank test; $BF=0.42$; rate of change computed in a time-window extending from $t_s - 250$ ms to t_s , where t_s is the stimulus onset time). In fact, regardless of the duration of the time-windows tested for computing the rate of change of Φ , there was no systematic difference in d' modulation across fast-ramp and slow-ramp trials (Fig. 5D, SI Fig. S5C).

Next, we divided trials based on ramping dynamics of $\Delta\Phi$ (global difference of SSVEP power indices; average $\Delta\Phi$ traces in Fig. 5E); in this case, the left panel shows positive $\Delta\Phi$ ($\Phi_{\text{target}} - \Phi_{\text{distractor}}$) dynamics when the target side was probed whereas the right panel shows negative $\Delta\Phi$ ($\Phi_{\text{distractor}} - \Phi_{\text{target}}$) dynamics when the distractor side was probed (for target and distractor Φ dynamics based on $\Delta\Phi$ split, see SI Fig. S5B). Remarkably, in this case, we observed a significantly higher $\Delta d'$ for slow-ramp trials as compared to fast-ramp trials (Fig. 5F; $\Delta d'$: fast-ramping = -0.11 ± 0.44 , slow-ramping = 0.61 ± 0.66 , $p=0.039$, Wilcoxon paired signrank test; $BF=2.80$). These $\Delta d'$ differences showed graded changes across time-windows used for computing rate of change: d' modulations were strongest for time-windows of 200-250 ms duration before stimulus onset (Fig. 5G) and weakened at much shorter (e.g. 50 ms) or much larger (e.g. 600-800 ms) durations (Fig. 5G, SI Fig. S5D).

In sum, distinct dynamics of global, cross-hemifield differences in SSVEP power ($\Delta\Phi$) were strongly predictive of accuracy (d') modulations. In particular, slow-ramping dynamics of $\Delta\Phi$ were predictive of d' differences between the target and distractor locations, with strongest effects occurring at a timescale matching that of the deployment of endogenous attention⁴⁰. Taken together, these results indicate that a gradual change in SSVEP power differences across hemifields represents a robust neural signature of attention's effects on discrimination accuracy.”

Also, I am not sure whether slow versus fast ramping is the appropriate characterization of the response pattern. There is a peculiar and much more prominent difference between slow and fast ramping trials. The latter shows an initially negative slope ramping in the opposite direction to the slow ramping trials, suggesting some sort of inconsistent ramping on those trials. This could mean that fast ramping trials Φ initially show a relative increase on the distractor side, or alternatively that Φ decreases first on the target relative to the distractor side. Both patterns would eventually account for a lesser d' , but the fast versus slow distinction is somewhat missing the point.

We appreciate the distinction mentioned by the reviewer. *A priori*, we had sought to divide trials based on “fast” versus “slow” ramping dynamics based on the rate of change of Φ just before triggering the stimulus. Yet, trials with “fast” dynamics appear to possess temporal features beyond those characterizing the speed of ramping alone. We discuss this now in the revised Discussion, as follows:

“Our cBMI neurofeedback paradigm revealed that a difference in the SSVEP power ($\Delta\Phi$) across hemifields provided a reliable marker for accuracy (d') modulations. To decouple the effects of changes in $\Delta\Phi$ from those on changes in Φ , we examined the dynamics of each metric (speed of ramping to threshold), and the corresponding effects on behavior. We observed that slow, but not fast, ramping dynamics of $\Delta\Phi$ at timescales ~ 200 - 300 ms reliably predicted d' differences between the target and distractor locations; these ramping dynamics match the timescale of deployment of endogenous attention¹. Taken together, these results indicate that global, competitive neural dynamics, unfolding at a timescale typically associated with the deployment of voluntary attention, are a sufficient neural marker for endogenous visuospatial attention.

Although, *a priori*, we sought to divide trials based on the speed of ramping of $\Delta\Phi$ (or Φ), trials with both “slow” and “fast” dynamics exhibited temporal features beyond those characterizing the speed of ramping alone. For example, fast ramping $\Delta\Phi$ trials showed a marginally negative slope of ramping, favoring the distractor side, followed by a large positive slope, and rapid rise toward the target side (Fig. 5E, left), potentially indicating that attention was initially deployed toward the distractor side and rapidly reoriented toward the target location on these trials. By contrast, slow ramping $\Delta\Phi$ trials showed a gradual increase toward the target location followed by a steady levels of activity prior to crossing the threshold, potentially indicative of sustained attention, for a few hundred milliseconds, toward the target location on these trials. Future studies could characterize in detail these, and other distinct types of, SSVEP power dynamics, and their consequences for behavior.”

Finally, at the time of grating presentation, $\Delta\Phi$ is the same for fast and slow ramping trials. If $\Delta\Phi$ predicts performance, shouldn't there be no performance difference between FR and SR, contrary to what is shown in Figure 5F?

On average, $\Delta\Phi$ indeed predicts performance when data were pooled across all types of ramping dynamics. The more fine-grained analysis in Figure 5F reveals that it is the “slow” ramping trials, rather than the “fast” ramping trials, in which this effect manifests.

(3) The reasoning regarding the absence of RT effects in the discussion on p. 29 is not entirely plausible to me. In the present experiments, fluctuations of Φ were found to link with accuracy but not with RT. Other studies using correlational approaches, however, have reported effects on both accuracy and RT. To account for the absence of RT differences in the present study, the authors suggest that attention engages distinct mechanisms (all reflected in SSVEP power fluctuations), causing distinct effects on d' and RT. Because attention engages those mechanisms in a highly correlated manner, correlational approaches may not be able to separate their effects and therefore see effects on d' and RT all the time. But is not clear how this would explain why RT effects are not seen here. If attention employs these mechanisms in a highly correlated manner, their effect should play out in interventional approaches as well.

We further clarify this key point.

Briefly, the interventional approach enables decoupling attention's effects on SSVEPs from those on other neural processes and, thereby, distinguishes the contribution of SSVEP fluctuations to behavioral effects on d' and RT. This idea is best illustrated by examining 4 directed, graphical models, each of which depicts a distinct mechanism for the influence of SSVEP power (Φ) on the behavioral metrics of d' and RT (figure below). If a change in Φ is sufficient to induce a change in a behavioral metric (d' or RT), we would expect to necessarily observe a change in the respective behavioral metric across high- Φ and low- Φ trials. The specific behavioral metric affected, then, enables us to distinguish between the 4 models.

We have revised the Results and Supplementary Information, and added a new SI Fig. S2, to more fully explain this point, as follows:

Revised Supplementary Information

SI Figure S2

Figure S2. Results auxiliary to Figure 1 (main text).

(*Topmost*) Time course of source signal ($A(t)$) (left) and intervened signal ($I(t)$) in (right).

Equations describing the relationships between the neural processes M , N , Φ and source signal A for the Correlational approach (left) and Interventional approach (right). Here, $A(t)$ represents the attention signal, $\Phi(t)$ is the SSVEP power (CDF normalized, see text for details), M and N are alternative neural processes distinct from Φ , d' is discrimination accuracy, and RT is reaction time. $e_x(t)$ represent additive noise, for the respective neural process (X) drawn from a unit normal distribution independently at each time instant.

- A. (*Top row*) Schematic depicting the No-effect model of attention's effect on d' and RT . (*Middle row*) Scatter plot showing relationship between Φ and RT (left sub-panel, red data) or between Φ and d' (right sub-panel, blue) for the Correlational approach. (*Bottom row*) Same as in the middle panel, but for the Interventional approach. In each panel, correlation coefficients (r) and significance values (p) are indicated above each panel.
- B. Same as in panel A, but for the RT -effect model.
- C. Same as in panel A, but for the d' -effect model.
- D. Same as in panel A, but for both-effects model.
- (B-D). Other conventions are the same as in panel A.

“Modeling interventional versus correlational approaches. We present 4 directed graphical models -- each of which reflects a distinct, possible mechanism for attention's effect on d' and RT modulation (SI Fig. S2, top row). In these models, A represents the cognitive process of attention, Φ represents SSVEP power modulation, M and N represent alternative neural processes (e.g. alpha or beta oscillations), distinct from SSVEP generating mechanisms, and d' and RT reflect behavioral effects on sensitivity and reaction times, respectively. Note that, in general, each neural process can be engaged by multiple cognitive processes other than attention (e.g. arousal) and can also be susceptible to internal, neural noise. Similarly, neural processes, other than those shown here, could influence each behavioral metric.

Each of these 4 models differs in the way it links SSVEP power (Φ), as well as the other neural processes, to the behavioral metrics of d' and RT . In the first model, SSVEP power changes do not directly influence either d' or RT . In the second and third models, SSVEP power changes drive RT and d' effects, respectively. In the fourth model, SSVEP power changes directly affect both d' and RT . For ease of reference, we call these the “no effect” model, the “ RT effect” model, the “ d' effect” model, and the “both effects” model.

These 4 models cannot be distinguished with correlational approaches alone. In conventional tasks that employ correlational approaches, the common source process (A) represents the strongest source of common variation among all of the neural processes (M , N , Φ) and, thereby, the behavioral metrics (d' , RT). As a result, correlations occur between neural processes and behavioral metric that are causally influenced by attention. These are the correlations, for example, between Φ and d' as well as between Φ and RT , that previous studies have reported^{1,2}. We demonstrate this result with simulations based on a simple mathematical formulation in SI Fig. S2 (middle row).

In an interventional approach, SSVEP power (Φ) is forced to be at fixed, predetermined high or low values on each trial. If a change in Φ were *sufficient* to induce a change in a behavioral metric (d' or RT), we would expect to observe an obligatory change in the respective behavioral metric value across high- Φ and low- Φ trials. The specific behavioral metric affected, then, enables us to distinguish between the 4 models. Again, we demonstrate this result with simulations in SI Fig. S2 (bottom row).

Although in our experiments, we did not have the ability to directly intervene on SSVEP power and set it to arbitrary values, we followed an indirect approach to achieve interventional control. We triggered stimulus presentation when Φ (Paradigm A) or $\Delta\Phi$ (Paradigm B) reached specific, predetermined high or low values. Because of the causal relationship between the source process A (e.g. attention), and neural processes (M, N, Φ) (SI Fig. S2, top row), fixing Φ to a particular value does not guarantee that A is also fixed to a specific value. Consequently, other neural processes, like M and N, are free to vary even if Φ is at a fixed value at the time of grating presentation. By this "intervention" on stimulus timing our experiment examines the effect of decoupling fluctuations in Φ from fluctuations in the other neural processes on behavioral metrics. Our results support the d' effect model -- changes in Φ produce obligatory changes in d' , but not in neural process N and, consequently, not in RT."

Revised Results

"Such an "interventional" cBMI provides two distinct advantages over correlational *post hoc* analysis approaches. First, the interventional cBMI enables decoupling and isolating SSVEP effects on behavioral metrics – accuracy (d') and reaction times (RT) – from those of other neural processes. We examine 4 directed, graphical models, each of which depicts a distinct mechanism for the influence of SSVEP power (or Φ) on the behavioral metrics of d' and RT (SI Fig. S2). If a change in Φ were sufficient to induce a change in a behavioral metric (d' or RT), we would expect to necessarily observe a change in the respective behavioral metric across high- Φ and low- Φ trials. The specific behavioral metric affected, then, enables us to distinguish between the 4 models; further details are provided in the SI (section on Modeling interventional versus correlational approaches; SI Figure S2)."

(4) Figure 2A: The response to the pedestrian is bigger for subsequently cued versus uncued trials even before the cue. It is not clear whether the time range before cue onset was subjected to statistical validation? If the difference turns out to be significant, it would pose a problem for data interpretation. The authors should comment on this in the manuscript.

In the high- Φ or low- Φ (all triggered) trials, it is possible that subjects were already paying attention to one side or another, even before cue onset to achieve extreme, supra-threshold values of SSVEP power. In this case Φ values could be already biased during the pedestrian. To avoid this confound, in the revised manuscript, we now present raw (unnormalized) Φ traces for the non-triggered trials alone, because in these trials SSVEPs did not achieve extreme values (high or low) on the cued side until the end of the trial. A statistical test comparing the SSVEP in the pedestal ($t=-500$ ms to 0 ms) before the cue onset revealed no significant difference in

SSVEP power across the cued and uncued locations ($p=0.967$, Wilcoxon signed rank test). By contrast SSVEP power in the post-cue window just before stimulus presentation ($t=-500$ ms to 0 ms prior to target and distractor onset) revealed evidence for significantly higher value at the cued, compared to the uncued, location ($p=0.006$).

We have revised the text as follows:

“As a first step, we sought to recapitulate a ubiquitous finding: cueing of attention is widely known to induce a gain in SSVEP power (e.g. Morgan et al., 1996¹⁶). Indeed, post-cue SSVEP power was higher at the cued location, compared that at the uncued location (Fig. 2A, cued $\Phi = 0.47 \pm 0.09$, uncued $\Phi = 0.43 \pm 0.07$, $p=0.006$, Wilcoxon signed-rank test, non-triggered trials) (see Methods section on *Analysis of cueing-induced gain of SSVEP power*); the difference was not significant in the pre-cue epoch (cued $\Phi = 0.44 \pm 0.08$, uncued $\Phi = 0.44 \pm 0.07$, $p=0.967$). These results provided electrophysiological evidence indicating that subjects were paying attention to the cued side during this task. We quantified, next, behavioral performance metrics for discriminating orientations at the target location (Fig. 1A, inset), for trials triggered based on the (cued) target’s SSVEP power index.”

Reviewer #2 (Remarks to the Author):

Tracking momentary fluctuations in human attention with an “interventional” cognitive brain-machine interface

In a study based on two experiments, the authors examined the effects of spatial attention on behavior. They used a novel brain-machine-interface approach that online extracted SSVEP amplitudes for left and right presented stimuli as an ongoing marker of concurrently deployed spatial attention and triggered the presentation of task-relevant, tilted gratings during high-SSVEP-amplitude (i.e. high attention) or low-amplitude (i.e. low attention) states. The authors present results suggesting a better (but not faster) detection of the orientation of the target grating when stimuli are presented during high-amplitude states as compared to low-amplitude states.

While I found the general aim of the study quite exciting and found many aspects of it thoughtful and well implemented and the figures very pleasing,

Thank you.

there are quite a few aspects that hinder the interpretation of the robustness and validity of the findings.

We have addressed these aspects, point-by-point, in the following responses.

One major aspect concerns the experimental parameters of the studies. The authors reported of many aspects of the study that were changed during the course of the data acquisition and thus differed across uneven subgroups of participants. Given the low number of participants, such changes and their impact on potential results were not evaluated and cannot be accounted for.

We acknowledge the variations in task and stimulus parameters in our study. In the revised manuscript, we have:

- a) Precisely indicated the subset of participants used for each analysis at each place in the Results (e.g. lines 247, 264, 296, 325-326, 432-433, 502, 1047-1054)
- b) Clarified the number of participants used in every analysis in each figure panel and sub-panel (e.g. Figs. 2A-C, 3A-B and SI Figs. S3A-C, S4A-C, S5A-B)
- c) Clarified the reasons behind the task and stimulus parameter variations at all appropriate places (e.g. lines 830-848, 961-970, 976-978, 1153-1156, 1183-1193, also see next).

Nonetheless, rather than being a limitation of the study, we propose that these variations underscore a key strength: They show that our findings are robust to routine modifications in task and stimulus parameters in these behavioral experiments.

We elaborate on the reasons for each of these changes, and their potential impact on our results and interpretation, in the subsequent responses.

• *In experiment 1: for many participants left and right stimuli were presented differently (13 and 15 Hz vs 25 Hz or static presentation) than the intended presentation with 14 and 18 Hz for left and right or right and left stimuli.*

We seek to clarify that we did not intend to fix the stimulation frequencies on the left and right hemifield at 14 Hz and 18 Hz, respectively across participants. We regret that the specific numerical values for flicker frequencies shown in Figure 1B gave this misleading impression.

In the experiment, we tested multiple different configurations for each participant in pilot sessions that preceded the main experiment, and chose the configuration that best provided SSVEP power SNR. For 16/26 participants (SI Table S1), we tested which combination – 14 Hz in the left hemifield and 18 Hz in the right hemifield, or vice versa – provided the best SSVEP SNR during the pilot session. The specific choice of SSVEP frequency for each hemifield was based on this pilot testing. For the remaining 10 participants, we could not reliably isolate good SSVEPs at 14 Hz and 18 Hz, in either configuration. For these participants, following pilot testing, we ran around 40 pilot trials by testing different SSVEP frequencies in the range of 13-18 Hz (step: 1 Hz), and selected the frequency that provided the strongest SSVEP response for that participant. For these participants, the stimulus in the other hemifield was either static (0 Hz, 6/10 participants) or flickering at a distal frequency (25 Hz, 4/10 participants). In this way, we ensured reliable isolation of SSVEP power from one stimulus without contamination from the other stimulus, for these participants.

These changes overall create physically and perceptually different conditions across participants. Especially the overall luminance of static stimuli is different as compared to flickering stimuli creating potential differences in the spatial biasing of attention due to luminance differences for left and right stimuli.

Static stimuli were used only in Paradigm A. In this paradigm, static stimuli do not systematically produce spatial biasing of attention toward any one hemifield for two reasons: i) static stimuli were presented with equal probability in the left and the right hemifields in a pseudorandom order across trials. ii) static stimuli were in the cued hemifield (target), or in the uncued hemifield (distractor), also with equal probability across trials. As a result we do not anticipate any spatial biasing of attention across trials, in a manner that systematically affects our results or their interpretation.

We clarify these points in the revised Methods. In addition, we have removed the specific numerical flicker frequency values in Figures 1B and 4B that could create a misleading impression.

Revised Methods

“In addition, the baseline session served to estimate individual-specific “best” SSVEP frequencies. For 5/15 participants we used fixed frequencies of 14 Hz and 18 Hz for the

SSVEPs (SI Table S1). These frequencies were counterbalanced between hemifields across participants. Perhaps because these were not ideal frequencies for evoking SSVEPs for these participants, signal-to-noise ratios (SNR) were poor for SSVEPs in 4/10 DSS dimensions estimated (SI Fig. S1D). Therefore, for the remaining 10 participants (who also ran distractor-triggered trials, see next section), we ran around 40 pilot trials by testing different SSVEP frequencies in the range of 13-18 Hz (step: 1 Hz), and selected the frequency that provided the strongest SSVEP response for that participant (SI Table S1); these were found to be either at 13 Hz or at 15 Hz. With this procedure, in all cases, DSS dimensions exhibited good SNR values for the SSVEPs (SI Fig. S1D). For this subset of participants, the stimulus on the other side was either non-flickering (0 Hz, 6/10 participants) or flickered at 25 Hz (4/10 participants); these latter frequencies were chosen to be sufficiently removed from the 13 Hz/ 15 Hz flicker frequency, so to ensure reliable isolation of SSVEP power at these frequencies. The stimulus tagged with individual-specific flicker frequencies (SSVEP stimulus) was presented with equal probability across left and right visual hemifields, and cues were counterbalanced across the SSVEP stimuli and the other (non-flickering or 25 Hz) stimulus such that either could be a target or distractor with equal probability. By this careful task design, we avoided systematic spatial biasing of attention toward any one side or any one type of stimulus in our experiments.”

Revised Figure 1B

Revised Figure 4B

• Stemming from the different stimulation frequencies: not all subjects contribute equally to the data plotted in figure 2A. Data with presumably differing SNR is entered, depending on the

stimulation regime and thus on whether SSVEP time courses could be extracted for cued and uncued SSVEPs simultaneously or during different trials only or for the cued stimulus only.

Please note that for each participant – regardless of whether one or two flickering stimuli were presented – we designed the task so that the flickering stimulus at each frequency was on the cued hemifield for an equal number of trials, as it was in the uncued hemifield. Moreover, before statistical quantification, to avoid biases due to different SNRs for the different frequencies, SSVEP power traces from each frequency were normalized to its *respective* value at baseline (by computing the cumulative distribution function or CDF), before averaging. Because of this design, we are essentially comparing and testing for differences in SSVEP power between cued and uncued conditions estimated with an identical number of trials, across corresponding (identical) frequencies (see revised Methods, above).

- *In addition, for 5 out of 15 participants of experiment 1, the stimulus presentation was triggered by the SSVEP amplitude for the cue side only, while no trials were present for which the stimulus presentation was triggered by the distractor SSVEP amplitude. This should affect the total trial number analyzed (and thus the SNR) or the temporal structure of the experiment.*

For 5/15 participants, the task design included only target-side triggered trials. We recognize that this yielded 5 additional participants with target-side triggered trials (15/15), as compared to distractor-side triggered trials (10/15).

Recognizing this difference in the number of participants and, consequently, differences in the total number of trials analyzed, we do not directly compare metrics for target-side triggered trials against those for distractor-side triggered trials, anywhere in the manuscript. In every case, we perform paired comparisons of these metrics across high- Φ and low- Φ conditions, but separately for each type of trial (target-side or distractor-side triggered).

- *As described in line 888 onwards: the ratio of high- Φ and low- Φ thresholds was altered across subjects. Why was it done, if it had no effect, as stated in line 890 and how was a potential difference tested?*

We clarify that it was not the ratio of high- Φ and low- Φ thresholds that was altered, but the thresholds themselves. For the first few (4/15) participants thresholds were more stringent (0.8 for high- Φ and 0.2 for low- Φ), but resulted in a high proportion of non-triggered trials in pilot experiments. As a result, these were relaxed (0.7 for high- Φ and 0.3 for low- Φ) for the remaining participants (11/15). We have now performed a statistical test (ANOVA) for potential differences between the two sub-groups, and added the following clarification in the revised Methods:

“For the first 4/15 participants (sub-group I), high- Φ and low- Φ thresholds were set to 0.8 and 0.2, respectively, but resulted in a high proportion of non-triggered trials in pilot experiments. Consequently, thresholds were relaxed to 0.7 and 0.3 for the remaining 11/15 participants (sub-group II). No significant interaction effects between behavioral parameters and sub-groups were

observed (d' : main effect of high- Φ vs low- Φ : $F(1,48)=7.58$, $p=0.008$, main effect of group: $p=0.752$, interaction effect: $p=0.088$; criterion: main effect of high- Φ vs low- Φ : $F(1,48)=0.09$, $p=0.761$, main effect of group: $p=0.529$, interaction effect: $p=0.890$; RT: main effect of high- Φ vs low- Φ : $F(1,48)=0.01$, $p=0.916$, main effect of group: $p=0.005$, interaction effect: $p=0.806$), indicating that modulations (across high- Φ and low- Φ trials) were not different across the two sub-groups for any behavioral parameter. Consequently, parameters across all 15 participants were combined for subsequent analyses.”

• *In experiment 2 the timing of the grating presentation was changed for some subjects. As a side note: the authors mention, potential differences were tested here but don't report the nature of these tests.*

We have now clarified the reasoning for this, and also performed statistical testing with an ANOVA as well as a permutation test. The text has been revised, as follows:

“For 3/9 participants presented in the main text, grating presentation occurred immediately upon $\Delta\Phi$ crossing threshold; for the remaining 6/9 participants, grating presentation occurred following a 60 ms delay of this threshold crossing. We introduced this delay for empirical reasons in the latter set of participants: to avoid stimulus onset from being coincident with the cessation of the auditory neurofeedback, because the latter terminated precisely at the time of threshold crossing. For both sets of participants, behavioral effects on normalized d' were closely similar: there was a significant main effect of task condition (high- $\Delta\Phi$ versus low- $\Delta\Phi$, $F(1,14)=21.06$, $p=0.0004$, ANOVA), but not of the timing ($p>0.99$), nor an interaction effect ($p=0.107$). A permutation test, by shuffling the timing label 1000 times across participants while retaining other labels intact, also revealed no significant evidence for an effect of timing on d' differences between high- $\Delta\Phi$ and low- $\Delta\Phi$ trials ($p=0.169$). Consequently, we pooled together these two groups of subjects for subsequent analyses.”

Related to the point above: there are text passages that imply consistent experimental parameters for the whole cohort, while in fact, this is not the case. Or it is not clear how missing or different data from subjects with different parameters are treated. This is misleading:

We regret the lack of clarity. We have now revised the text passages, qualifying each of these statements (see next).

• *line 148: “two flickering pedestals appeared”; not true for all subjects*

Removed. Page 9, Line 145;

• *line 150: “each flickered at a distinct frequency”: not true for all subjects*

Amended as: “Either one (6/15 subjects) or both (9/15 subjects) pedestals...”. Page 9, Lines 147-148;

- line 198/199 “Target (and distractor) presentation was triggered”: not true for all subjects

The original statement in the text – “Target (and distractor) presentation was triggered based on SSVEP power levels in real-time.” – is entirely accurate for all subjects. Both target and distractor stimuli were indeed triggered at the same time in all cases.

Per the reviewer’s suggestion, we have amended this statement for further clarity, as follows:
“Target (and distractor) presentation was triggered simultaneously based on SSVEP power levels of either one of the two stimuli, in real-time.” Page 11, Lines 207-209;

In the next section when presenting the results we specify exactly for which subset of subjects triggering happened based on target Φ levels, and for which based on distractor Φ levels.

- line 254: “Mean Φ on the distractor side was not significantly different across the target high- Φ and low- Φ ”: Calculation not for all subjects possible. Here it is quite critical as the statistics calculated, was calculated only for a subset of participants but generalized to the effects of the whole experiment.

Clarified as follows:

“Mean Φ on the distractor side was not significantly different across the target high- Φ and low- Φ trials (distractor Φ value: high- Φ trials = 0.53 ± 0.06 , low- Φ trials = 0.51 ± 0.07 , $p=0.125$, Wilcoxon signed-rank test; SI Fig. S2C; $n=5$ subjects).”

In the Methods we clarify also that:

“For these analyses, and those in SI Fig. S2C, we used data from $n=5$ participants (subject indices A1, A12, A13, A14, and A15, SI Table S1) for whom the SSVEP on both sides exhibited high SNR on both sides and could, therefore, be reliably analyzed. Although 9 participants were tested with flickering stimuli on both sides, for 4 participants, the SSVEP on either the target or the distractor SSVEP DSS dimension exhibited low SNR (SI Fig. S1D) precluding the reliable estimation of $\Delta\Phi$; therefore, data from the remaining 5 participants were included in these analyses. Data from 4 out of these 5 participants were used for target-triggered trials, as one participant (subject A1, SI Table S1) was not tested with distractor side triggering.”

- Figure 2: A “pooled across all subjects ($n=15$)” not the same for all subjects

Clarified above. Every SSVEP frequency for every subject – regardless of whether one or two flickering stimuli were tested – occurred in the cued or uncued hemifield for an equal number of trials, enabling direct comparison of SSVEP power across these trial types.

- Figure 3: Parts of the figure (A, B) are based on data from 5 subjects and some for 10. Why is this the case?

We have revised this analysis significantly; please see revised Figure 3, and associated revisions in the manuscript.

Revised Results

“Local SSVEP fluctuations rather than global SSVEP power differences correlate with accuracy effects

There is some debate in the literature about whether attention resources across the two visual hemifields are independent^{31,32}. While some studies suggest that sensory processing resources for the attended hemifield (e.g., as indexed by target SSVEP power) are allocated independently of the unattended hemifield (e.g., as indexed by distractor SSVEP power)^{31,33}, other studies suggest that attention produces a global competition for sensory resources across hemifields^{32,34}. While these hypotheses are not mutually exclusive, the results of Paradigm A support the former hypothesis: distractor side Φ states did not systematically predict target d' variations. In other words, the behavioral effects on d' were mediated, at least in part, by local, not global, SSVEP power fluctuations. To further confirm this hypothesis, we performed two additional analyses.

First, we plotted the Φ traces, separately for the target and the distractor sides, for the high- Φ and low- Φ target SSVEP triggered trials (Fig. 3A; $n=5$ subjects with flickering stimuli and high SNR on both sides, Methods). We observed no systematic modulation of mean Φ on the distractor side across the target high- Φ and low- Φ trials (distractor Φ value: high- Φ trials = 0.53 ± 0.06 , low- Φ trials = 0.51 ± 0.07 , $p=0.125$, Wilcoxon signed-rank test; $BF=1.36$) (SI Fig. S3B). In addition, trial-wise analysis revealed no evidence for correlation between target- Φ and distractor- Φ either in the high- Φ trials ($r=-0.01$, $p=0.90$) or in the low- Φ trials ($r=0.06$, $p=0.13$) (Methods, section on *Trial-wise analysis of correlation between target- Φ and distractor- Φ* ; see also trial-averaged correlations shown in SI Fig. S3C).

Second, we tested if target discrimination accuracy would vary with the strength of global competition between target and distractor representations, at a fixed level of target (or distractor) SSVEP power. For this, we computed the difference between the SSVEP power indices across hemifields ($\Delta\Phi = \Phi_{\text{target}} - \Phi_{\text{distractor}}$), just before stimulus presentation (at the time of threshold crossing), separately for the target triggered and distractor triggered trials, and separately for the high- Φ and low- Φ trials (Fig. 3B). For each trial type, we divided the trials into two subsets, based on a median split (Methods section on *Analysis of difference of SSVEP power across hemifields*). For the target triggered high- Φ trials and the distractor triggered low- Φ trials, the first subset of trials (values above the median) comprised trials typically with large positive values of $\Delta\Phi$ (“extreme $+\Delta\Phi$ ”) whereas the second subset (values below the median) comprised of $\Delta\Phi$ values around zero (“moderate $\Delta\Phi$ ”) (Fig. 3B, upper-left and lower-right respectively). Conversely, for the target triggered low- Φ trials and the distractor triggered high- Φ trials, the first subset of trials (values above the median) comprised of $\Delta\Phi$ values around zero (“moderate $\Delta\Phi$ ”) whereas the second subset (values below the median) comprised typically with large negative values of $\Delta\Phi$ (“extreme $-\Delta\Phi$ ”) (Fig. 3B, upper-right and lower-left respectively). We confirmed that the $\Delta\Phi$ distributions for every pair of subsets was significantly different from each other (Fig. 3B, insets; $p<0.001$, for 4 pairwise comparisons, Kolmogorov-Smirnov test). We then tested if d' modulations would co-vary systematically with $\Delta\Phi$ for each trial type.

Accuracies did not vary systematically with $\Delta\Phi$ across the first and second subsets, both when the data were analyzed together ($F(1,56) = 0.85$, $p=0.360$, ANOVA, $n=5$ subjects for target

triggered trials, and $n=4$ for distractor triggered trials), and in *post hoc* comparisons, when each of the four trial types were compared separately ($p>0.05$, estimation statistics).

In other words, modulations of target SSVEP power were not accompanied by congruent modulations of distractor SSVEP power in the opposite hemifield, a result consistent with earlier studies^{34,35}. Moreover, once local SSVEP power was fixed at specific (threshold) values, d' did not modulate with global differences in SSVEP power across the visual hemifields. In other words, sensory processing for the stimulus in the attended hemifield appears to be, at least, in part, independent of processes occurring in the opposite hemifield.”

Figure 3

Figure 3. Difference of SSVEP power across hemifields ($\Delta\Phi$) indexes d' enhancement

- Mean Φ traces for high- Φ trials (blue), low- Φ trials (red) and non-triggered trials (gray), for Target and Distractor triggered trials, time-locked to the triggering of the grating stimuli (dashed vertical line). Shaded region: s.e.m.
- (*Top Left*) d' values (dots) based on a median split of $\Delta\Phi$ for in high- Φ , target side triggered trials. Extreme (Ext) + $\Delta\Phi$ and moderate (Mod) $\Delta\Phi$ refer to $\Delta\Phi$ values greater than and less than the median, respectively. (*Inset*) $\Delta\Phi$ distribution for Extreme + $\Delta\Phi$ (light, solid line) and Moderate $\Delta\Phi$ (light, dashed line) subsets for high Φ trials. Dark lines: Curve fits based on a kernel density estimate. (*Top Right, Bottom Left and Bottom Right*) Same as in the top left panel, but showing median split results for low- Φ target side triggered trials, high- Φ distractor side triggered trials and low- Φ distractor side triggered trials, respectively. All panels: Other conventions are the same as in Figure 2B.”

Revised Methods

“Analysis of difference of SSVEP power across hemifields ($\Delta\Phi$). We tested whether the difference in SSVEP power indices between the target (cued) and the distractor (uncued) hemifield ($\Delta\Phi = \Phi_{\text{Target}} - \Phi_{\text{Distractor}}$), rather than the power index (Φ) would provide a more reliable marker for modulations of behavioral accuracy in paradigm A. This analysis was performed offline, as a precursor to the next online experiment (paradigm B). EEG data were analyzed offline using a procedure identical to that applied for the real-time data, except that in this case, the difference of SSVEP power index values between the target and the distractor hemifields ($\Delta\Phi$) was computed for each trial at the time of threshold crossing, separately for each condition: i) target triggered vs distractor triggered, ii) high- Φ vs low- Φ , and iii) left vs right hemifield SSVEP (Fig. 3B). For each subject, we divided trials in each condition into two subsets, based on a median split of $\Delta\Phi$. For the target triggered high- Φ trials and the distractor triggered low- Φ trials, trials with $\Delta\Phi$ values above the median, typically with large positive values of $\Delta\Phi$, were labeled as “extreme + $\Delta\Phi$ ” trials, whereas the trials with $\Delta\Phi$ values below the median, typically $\Delta\Phi$ values around zero, were labeled as “moderate $\Delta\Phi$ ” trials (Fig. 3B, upper-left and lower-right respectively). Conversely, for the target triggered low- Φ trials and the distractor triggered high- Φ trials, the subset of trials with $\Delta\Phi$ values above the median were labeled as “moderate $\Delta\Phi$ ”; whereas the subset with $\Delta\Phi$ values below the median was labeled as “extreme - $\Delta\Phi$ ” (Fig. 3B, upper-right and lower-left respectively). We tested whether the $\Delta\Phi$ distributions for every pair of subsets were significantly different from each other with pairwise Kolmogorov-Smirnov (KS) tests. We then tested if d' modulations co-varied systematically with $\Delta\Phi$ for each trial type using two kinds of statistical tests. Firstly, we performed an n-way ANOVA with d' as the dependent variable and $\Delta\Phi$, Φ (high- Φ vs low Φ trials), hemisphere (left and right), and type of triggering (target vs distractor) as independent factors. Secondly, in post hoc comparisons, each of the four trial types were analysed separately using estimation statistics⁷⁸. d' values for left and right hemifields were pooled for each of the 4 trial types, before performing KS tests and computing estimation statistics.

For these analyses, and those in SI Fig. S2C , we used data from n=5 participants (subject indices A1, A12, A13, A14, and A15, SI Table S1) for whom the SSVEP on both sides exhibited high SNR on both sides and could, therefore, be reliably analyzed. Although 9 participants were tested with flickering stimuli on both sides, for 4 participants, the SSVEP on either the target or the distractor SSVEP DSS dimension exhibited low SNR (SI Fig. S1D) precluding the reliable estimation of $\Delta\Phi$; therefore, data from the remaining 5 participants were included in these analyses. Data from 4 out of these 5 participants were used for target-triggered trials, as one participant (subject A1, SI Table S1) was not tested with distractor side triggering.”

Concerning the discussion of the findings of both experiments, the author claim that a d' modulation is measurable when target stimuli presentation is based on the differences between SSVEPs in both hemisphere (hinting at some global competition) while this is not the case when stimuli are triggered by different SSVEP levels of the distractor. While this would be a very interesting finding, this claim cannot easily be made given the data:

We note that there are two sets of results in the study.

Experiment 1 (Paradigm A) tests whether d' modulation at the target location depends on local SSVEP power variations – i) at the target location, or ii) at the distractor location – independent of the other location.

Experiment 2 (Paradigm B) addresses a different question. It tests whether d' modulation occurs when subjects are trained, with neurofeedback, to increase the global difference in SSVEP power between the target and distractor locations (inter-hemifield competition).

• *The crucial test for this claim is not available in the experiment/results: while for experiment 1 the impact of a distractor can be evaluated for different levels of distractor SSVEP amplitudes,*

Indeed, by default, in a standard cued attention task, distractor SSVEP (Φ) levels do not appear to influence target d' values. Please see Figure 3A in the revised manuscript (above), as well as SI Figure S3C.

Revised SI Figure S3C

this crucial contrast is not possible in experiment 2 and is also not presented in figure 4 for instance. Here, if I understood correctly, the stimulus presentation was triggered when the SSVEP lateralization reached an upper threshold and stimuli were either presented on the side with the high SSVEP amplitude (target-cued trials) or on the other side (distractor-cued trials). Hence for distractor processing in experiment 2, there is no statement possible in whether it varies according to the level of SSVEP lateralization.

We now show distractor SSVEP (Φ) levels also in revised Figure 4 and SI Figure S4B. The results reveal that target- Φ and distractor- Φ are anti-correlated in Paradigm B. In other words, when subjects were trained to increase the global difference in SSVEP power between the target and distractor locations, competitive interactions were generated, and yielded d' modulations.

Revised Figure 4D (left) and SI Figure S4B (right)

We did not test target and distractor SSVEP levels separately in paradigm B, as this was not the goal of this paradigm, but this is now presented in revised Figure 5 (see response to a subsequent comment). Furthermore, in the revised Discussion, we have clarified that the claims regarding the lack of target d' modulations based on distractor SSVEP power variations are based on Paradigm A alone.

“Second, previous literature suggests that attentional resources across the left and right visual hemifields are separate and, potentially, independent; this dissociation has been reported in many studies including human psychophysics^{31,33}, human electrophysiology^{34,35} and non-human primate electrophysiology⁸. In line with these findings, in Paradigm A, target discrimination d' was not systematically modulated either by distractor SSVEP power states ($\Phi_{\text{distractor}}$), or by the global difference of SSVEP power across hemifields ($\Delta\Phi$), suggesting that SSVEP power modulations may occur independently across brain hemispheres, by default, during cued attention tasks. Yet, in Paradigm B, when subjects were trained using neurofeedback, to increase the difference of SSVEP power across hemifields ($\Delta\Phi$) – and stimuli were triggered based on this neural signature of stimulus competition^{32,65} – we observed robust effects on d' , with Bayes Factor values around 9.0.”

Most crucially in the second experiment, the SSVEP lateralization triggering the stimulus presentation was not manipulated orthogonally to the attentional focus, hindering the examination of crucial contrast of interest.

A minor clarification is in order. In Paradigm B, no explicit attention cue was provided (Fig. 4B). Rather, we define the terms “target” and “distractor”, operationally, as follows:

“For each trial in this experiment, one hemifield was selected pseudorandomly as a “target” side and the other, as the “distractor” side; target and distractor sides were counterbalanced, with equal probability, across the left and right hemifields. We triggered the presentation of the grating stimuli when the difference in SSVEP power index between the target and distractor

sides ($\Delta\Phi = \Phi_{\text{target}} - \Phi_{\text{distractor}}$) reached a prespecified, participant specific, threshold measured in an earlier practice session (Methods). Participants reported the tilt of the grating indicated by a *post hoc* response probe (Fig. 4B); the target and distractor sides were probed with equal probability.”

With this design, we measured the d' for both stimuli (target and distractor). The behavioural results based on d' show that the participant’s attention was indeed engaged at the location of higher (lateralized) SSVEP power (higher d' at the “target” compared to the “distractor” location). (Fig. 4E).

- *There are other differences in the design that could also be responsible for potential effects: different stimuli, a different way of having participants shift their attention (top-down cue vs ongoing auditory feedback), different lengths of trials*

As mentioned above, each paradigm was designed to answer a different question (localized SSVEP power variations versus global competition). Given these different design choices for the two paradigms we do not directly compare the magnitude of behavioural (d') effects between these paradigms, anywhere in the manuscript.

- *I was also wondering how a pattern of “extreme, global differences in SSVEP power across hemifields” [line 413] could be differentiated from a pattern with an upregulated SSVEP amplitude for the attended side and a non-regulated SSVEP amplitude for the unattended side (more capturing the findings from experiment 1).*

It is precisely this pattern – extreme, global differences in SSVEP power – that paradigm B was designed to generate. While we acknowledge that extreme, global differences can occur in the absence of global competition, in the revised manuscript, we have provided detailed evidence showing that global competition indeed occurs in Paradigm B: We show that target- Φ and distractor- Φ are anti-correlated, i.e. they vary in opposite directions (revised Fig. 4D and SI Figure S4B, below).

In addition, trial-wise analysis revealed clear evidence for anti-correlation between the target- Φ and distractor- Φ ($r = -0.37$, $p < 0.001$).

In other words – when subjects sought to achieve large differences between target- Φ and distractor- Φ , based on auditory neurofeedback – competitive interactions between the hemifields were generated.

Revised Figure 4D (left) and SI Figure S4B (right)

We have amended the text to reflect these findings, as follows:

“First, we tested if this neurofeedback paradigm induced competitive interactions across the visual hemifields. Indeed, in this paradigm, target- Φ and distractor- Φ varied in opposite directions – at trend clearly visible in trial-average traces over time (Fig. 4D, lower). In addition, trial-wise analysis revealed clear evidence for anti-correlation between the target- Φ and distractor- Φ (SI Fig. S4B) ($r = -0.37$, $p < 0.001$; Methods, section on *Trial-wise analysis of correlation between target- Φ and distractor- Φ*). In other words – when subjects sought to achieve large differences between target- Φ and distractor- Φ , based on auditory neurofeedback – robust competitive interactions were generated between the neural representations in each hemifield.”

Another aspect concerns the statistical analyses and interpretations based thereon:

- For the analysis of potential differences of behavioral parameters between the high and low SSVEP state via Wilcoxon signed-rank test (as displayed in figure 2) the values from left and right hemisphere-derived data are treated as independent samples. This, however, violates the test's assumption that all paired observations are drawn randomly and independently as they stem from the same subject and I have serious doubts that this is a valid procedure. Given that the main finding is not significant when not relying on this procedure (as seen in SI Figure S2), renders the main message of the manuscript shaky.

We thank the reviewer for this suggestion.

In the revision, rather than averaging the left and right hemisphere data (as done in SI Figures S2), we have now performed a 2-way ANOVA for Paradigm A with Φ levels (high- Φ and low- Φ) and hemisphere (left and right) as the two independent factors and each behavioral metric as a dependent variable.

For d' modulation in the target-side triggered trials, we find a significant main effect of Φ levels with d' for high- $\Phi >$ low- Φ ($F(1,48)=4.47$, $p=0.040$). In addition, we find no main effect of hemispheres ($p=0.563$), nor a significant interaction effect ($p=0.599$). On the other hand, for d' modulation in the distractor-side triggered trials, we find no main effect of Φ levels ($p=0.635$), no main effect of hemispheres ($p=0.881$), nor a significant interaction effect ($p=0.379$). These new statistical analyses confirm the validity of main effect on d' reported in the paper.

For all parameters, we now report these new p-values based on the ANOVA in the revised manuscript. In addition, we report the Bayes factor for many such comparisons (see final response).

Revised Results

Describing main effect of Φ on d' , c , and RT on target triggered trials (Fig. 2B).

“Discrimination accuracy (d') was significantly higher for the high- Φ compared to the low- Φ trials (Fig. 2B, left; d' : ANOVA, main effect, high- $\Delta\Phi$ versus low- $\Delta\Phi$, $F(1,48)=4.47$, $p=0.040$; Bayes Factor (BF)=2.33; $n=15$ subjects; see Methods section on *Statistical Analyses*). On the other hand, there was no significant difference in criteria across high- Φ and low- Φ trial types (Fig. 2B, middle; criterion: ANOVA, main effect, high- $\Delta\Phi$ versus low- $\Delta\Phi$, $F(1,48)=0.27$, $p=0.608$; BF=0.22). Moreover, we observed no significant differences in reaction times also across high- Φ and low- Φ trials (Fig. 2B, right; RT: ANOVA, main effect, high- $\Delta\Phi$ versus low- $\Delta\Phi$, $F(1,48)=0.08$, $p=0.772$; BF=0.22).”

Describing main effect of Φ on d' , c , and RT on distractor triggered trials (Fig. 2C).

“In this case, we observed no apparent difference in d' , criterion or reaction time across the high- Φ and low- Φ trials (Fig. 2C, left, distractor SSVEP triggered: d' : ANOVA, main effect, high- $\Delta\Phi$ versus low- $\Delta\Phi$, $F(1,36)=0.23$, $p=0.635$; Bayes Factor (BF)=0.23; middle, criterion: ANOVA, main effect, high- $\Delta\Phi$ versus low- $\Delta\Phi$, $F(1,36)=0.96$, $p=0.333$; Bayes Factor (BF)=0.38; right, RT: ANOVA, main effect, high- $\Delta\Phi$ versus low- $\Delta\Phi$, $F(1,36)=0.11$, $p=0.740$; Bayes Factor (BF)=0.24; $n=10$ subjects).”

As a side note: why is $N = 26$, shouldn't it be 30?

The 4 data points were removed due to poor SNR (<0.25) (SI Fig. S1D). We indicated this in the Methods (Section on SSVEP SNR calculation, lines 768-771).

• A similar procedure for the statistical analysis is shown in figure 3C

Statistical analyses for Fig. 3 are now replaced with an ANOVA.

Revised Results describing main effect of $\Delta\Phi$ on d' (Fig. 3).

“Accuracies did not vary systematically with $\Delta\Phi$ across the first and second subsets, both when the data were analyzed together ($F(1,56) = 0.85, p=0.360$, ANOVA, $n=5$ subjects for target triggered trials, and $n=4$ for distractor triggered trials), and in *post hoc* comparisons, when each of the four trial types were compared separately ($p>0.05$, estimation statistics).”

• Given the differences between both experiments (see above), I'm also unsure whether pooling data from both experiments for the slope analysis across time bins is a good idea.

Per the reviewer's suggestion, we now include only trials from Paradigm B in this analysis. The results shown in Figure 5 – and accompanying Supplementary Figure S5 – are now revised with data from Paradigm B alone.

Revised Figure 5

Figure 5

Revised SI Figure S5A

SI Figure S5

“Slowly ramping dynamics of competitive interactions yield stronger behavioral effects

The previous results demonstrate that robust effects on discrimination accuracy occur when global (cross-hemifield) difference in SSVEP power ($\Delta\Phi$) reach large values. Yet, large $\Delta\Phi$ values also entail a high Φ value at the target location on target-side probed trials, and a low distractor Φ value at the distractor location on distractor-side probed trials. It is possible, then, that the effects observed in Paradigm B occurred due to extreme (high or low) co-fluctuations in Φ for the stimulus on the respective side (target or distractor, respectively), and were independent of Φ for the stimulus on the other side, as in Paradigm A.

To distinguish these possibilities further, we analyzed the dynamics of Φ and $\Delta\Phi$ in Paradigm B immediately prior to stimulus triggering. Typically, endogenous attention is deployed at a timescale of about 200-300 ms⁻¹ and sustaining attention for many seconds is known to impair orientation discrimination performance⁴⁰. If either Φ or $\Delta\Phi$ dynamics were to represent a neural signature of attention, we hypothesized that attention’s behavioral timescales would be faithfully reflected in the respective metric’s timescales as well. For this, we tested whether specific patterns in the dynamics of local SSVEP power (Φ), or its global difference across hemifields ($\Delta\Phi$), at distinct timescales would yield systematically different perceptual accuracies (Fig. 5A).

First, we divided trials into two categories – “fast-ramp” versus “slow-ramp” – based on the rate of change of Φ , just before threshold crossing, across trials of Paradigm B (Methods). The average Φ traces for each of these sets of trials are shown in Figure 5B (solid lines: slow ramp; dashed lines: fast ramp); the left panel shows target- Φ dynamics from target-side probed trials whereas the right panel shows distractor- Φ dynamics from distractor-side probed trials (for the converse dynamics, see SI Fig. S5A). We then computed the difference in d' – $\Delta d'$ – between these two sets of trials. Note that this quantity, $\Delta d'$, measures the difference in d' for the probed stimulus between trials when the SSVEP power index was high, compared to when it was low, at the probed location. We tested if this d' modulation was different between fast-ramp and slow-ramp trials.

Interestingly, when we divided trials based on ramping dynamics of Φ , we observed no significant differences between the two categories (Fig. 5C, $\Delta d'$: fast-ramp $\Phi = 0.43 \pm 0.54$, slow-ramp $\Phi = 0.40 \pm 0.69$, $p=0.496$, Wilcoxon paired signrank test; $BF=0.42$; rate of change computed in a time-window extending from $t_s - 250$ ms to t_s , where t_s is the stimulus onset time). In fact, regardless of the duration of the time-windows tested for computing the rate of change of Φ , there was no systematic difference in d' modulation across fast-ramp and slow-ramp trials (Fig. 5D, SI Fig. S5C).

Next, we divided trials based on ramping dynamics of $\Delta\Phi$ (global difference of SSVEP power indices; average $\Delta\Phi$ traces in Fig. 5E); in this case, the left panel shows positive $\Delta\Phi$ ($\Phi_{\text{target}} - \Phi_{\text{distractor}}$) dynamics when the target side was probed whereas the right panel shows negative $\Delta\Phi$ ($\Phi_{\text{distractor}} - \Phi_{\text{target}}$) dynamics when the distractor side was probed (for target and distractor Φ dynamics based on $\Delta\Phi$ split, see SI Fig. S5B). Remarkably, in this case, we observed a significantly higher $\Delta d'$ for slow-ramp trials as compared to fast-ramp trials (Fig. 5F; $\Delta d'$: fast-ramping = -0.11 ± 0.44 , slow-ramping = 0.61 ± 0.66 , $p=0.039$, Wilcoxon paired signrank test;

BF=2.80). These $\Delta d'$ differences showed graded changes across time-windows used for computing rate of change: d' modulations were strongest for time-windows of 200-250 ms duration before stimulus onset (Fig. 5G) and weakened at much shorter (e.g. 50 ms) or much larger (e.g. 600-800 ms) durations (Fig. 5G, SI Fig. S5D).

In sum, distinct dynamics of global, cross-hemifield differences in SSVEP power ($\Delta\Phi$) were strongly predictive of accuracy (d') modulations. In particular, slow-ramping dynamics of $\Delta\Phi$ were predictive of d' differences between the target and distractor locations, with strongest effects occurring at a timescale matching that of the deployment of endogenous attention⁴⁰. Taken together, these results indicate that a gradual change in SSVEP power differences across hemifields represents a robust neural signature of attention's effects on discrimination accuracy."

• Throughout the absence of a significant difference is treated and interpreted as the absence of a significant effect. Null-effects in a conventional significance testing do, however, not provide actual evidence for the null hypothesis (in contrast for instance to Bayesian analysis approaches, see Rouder et al., 2009) as the design may just be underpowered to detect effects below a specific effect size. In general, using a Bayesian framework (or other likelihood approaches) for the statistical tests could help to substantiate whether the null findings support the null hypothesis or whether evidence is pointing in neither direction of the null- or the alternative hypothesis

We thank the reviewer for the suggestion. We have included the Bayes factor, when reporting each of the results in the revised manuscript. We list some examples, below:

- Lines 245-247: "Discrimination accuracy (d') was significantly higher for the high- Φ compared to the low- Φ trials (Fig. 2B, left; d' : ANOVA, main effect, high- $\Delta\Phi$ versus low- $\Delta\Phi$, $F(1,48)=4.47$, $p=0.040$; **Bayes Factor (BF)=3.36**; $n=15$ subjects; see Methods section on *Statistical Analyses*)."
- Lines 248-250: "there was no significant difference in criteria across high- Φ and low- Φ trial types (Fig. 2B, middle; criterion: ANOVA, main effect, high- $\Delta\Phi$ versus low- $\Delta\Phi$, $F(1,48)=0.27$, $p=0.608$; **BF=0.20**)."
- Lines 250-252: "we observed no significant differences in reaction times also across high- Φ and low- Φ trials (Fig. 2B, right; RT: ANOVA, main effect, high- $\Delta\Phi$ versus low- $\Delta\Phi$, $F(1,48)=0.08$, $p=0.772$; **BF=0.96**)."
- Lines 431-434: "This cBMI neurofeedback paradigm revealed robust effects of $\Delta\Phi$ on behavioral accuracies. Discrimination d' was significantly higher for the target compared to the distractor (Fig. 4E, left, SI Fig S4B, $p=0.006$, d' : target = 1.80 ± 0.08 , distractor = 1.51 ± 0.09 , $p=0.006$, **BF=18.28**, $n=9$ subjects, paired test, estimation statistics)."
- Lines 435-437: "As before, no significant differences were observed in reaction times for target versus distractor responses (Fig. 4E, right, RT: target = 976 ± 84 ms, distractor = 970 ± 82 ms, $p=0.335$, **BF=0.17**)."

One aspect that may be of interest for the interpretation of the data: were the target stimuli for all different conditions (cued high vs cued low vs uncued high vs uncued low) presented at comparable time points throughout each trial?

For the revision, we have plotted the distribution of cue-target intervals for target-side triggered trials and distractor-side triggered trials in SI Fig. S3A. We did not see any significant difference between the median triggering times across high- Φ (blue) and low- Φ (red) trials for either condition (target-side triggered: $p=0.252$; distractor-side triggered: $p=0.105$, Wilcoxon paired sign rank test).

SI Figure S3

Minors:

- page 4 line 46: it is not mentioned for which measure the Fano Factor decreased
Clarified.

“... attention induces modulations of various neuronal metrics, including an increase in firing rate, a decrease in firing rate Fano factor, or a decrease in noise correlations across the population ...”

- page 9 line 153: what is a brief interval?

The interval was determined by the SSVEP power dynamics, i.e. the time it took for Φ to cross either high or low thresholds. For clarity, we have removed the phrase “brief interval” and replaced it with the phrase “variable interval” (line 151).

- page 10 line 179 onwards: while I think it’s valid to say that data seems to be transferred and processed near real-time with neglectable delay, the SSVEP power calculation relies on the integration of data from a time window of 500 ms, if I understood correctly, and is thus delayed.
Clarified.

“... SSVEP power index evoked by the pedestal on the cued (target) or uncued (distractor) side was computed and tracked, for each trial, in real-time (closed-loop delays: mean \pm std: 21.1 \pm 5.9 ms, SI Fig. S1E; SSVEP power computed in a 500 ms window) ...”

- Figure 3: What does “PDF” stand for?

Removed in the revised Figure 3.

• Discussion line 592 onwards: I don't get why the cBMI approach does not suffer from multiple mechanisms engaged in creating behavior. This point could be elaborated.

This is discussed in detail now in the Supplementary Information and new SI Figure S2.

Revised Supplementary Information

SI Figure S2

Figure S2. Results auxiliary to Figure 1 (main text).

(Topmost) Time course of source signal $A(t)$ (left) and intervened signal $I(t)$ in (right).

Equations describing the relationships between the neural processes M , N , Φ and source signal A for the Correlational approach (left) and Interventional approach (right). Here, $A(t)$ represents the attention signal, $\Phi(t)$ is the SSVEP power (CDF normalized, see text for details), M and N are alternative neural processes distinct from Φ , d' is discrimination accuracy, and RT is reaction time. $e_X(t)$ represent additive noise, for the respective neural process (X) drawn from a unit normal distribution independently at each time instant.

E. (Top row) Schematic depicting the No-effect model of attention's effect on d' and RT . (Middle row) Scatter plot showing relationship between Φ and RT (left sub-panel, red data) or between Φ

and d' (right sub-panel, blue) for the Correlational approach. (*Bottom row*) Same as in the middle panel, but for the Interventional approach. In each panel, correlation coefficients (r) and significance values (p) are indicated above each panel.

F. Same as in panel A, but for the RT-effect model.

G. Same as in panel A, but for the d' -effect model.

H. Same as in panel A, but for both-effects model.

(B-D). Other conventions are the same as in panel A.

“Modeling interventional versus correlational approaches. We present 4 directed graphical models -- each of which reflects a distinct, possible mechanism for attention’s effect on d' and RT modulation (SI Fig. S2, top row). In these models, A represents the cognitive process of attention, Φ represents SSVEP power modulation, M and N represent alternative neural processes (e.g. alpha or beta oscillations), distinct from SSVEP generating mechanisms, and d' and RT reflect behavioral effects on sensitivity and reaction times, respectively. Note that, in general, each neural process can be engaged by multiple cognitive processes other than attention (e.g. arousal) and can also be susceptible to internal, neural noise. Similarly, neural processes, other than those shown here, could influence each behavioral metric.

Each of these 4 models differs in the way it links SSVEP power (Φ), as well as the other neural processes, to the behavioral metrics of d' and RT. In the first model, SSVEP power changes do not directly influence either d' or RT. In the second and third models, SSVEP power changes drive RT and d' effects, respectively. In the fourth model, SSVEP power changes directly affect both d' and RT. For ease of reference, we call these the “no effect” model, the “RT effect” model, the “ d' effect” model, and the “both effects” model.

These 4 models cannot be distinguished with correlational approaches alone. In conventional tasks that employ correlational approaches, the common source process (A) represents the strongest source of common variation among all of the neural processes (M, N, Φ) and, thereby, the behavioral metrics (d' , RT). As a result, correlations occur between neural processes and behavioral metric that are causally influenced by attention. These are the correlations, for example, between Φ and d' as well as between Φ and RT, that previous studies have reported^{1,2}. We demonstrate this result with simulations based on a simple mathematical formulation in SI Fig. S2 (middle row).

In an interventional approach, SSVEP power (Φ) is forced to be at fixed, predetermined high or low values on each trial. If a change in Φ were *sufficient* to induce a change in a behavioral metric (d' or RT), we would expect to observe an obligatory change in the respective behavioral metric value across high- Φ and low- Φ trials. The specific behavioral metric affected, then, enables us to distinguish between the 4 models. Again, we demonstrate this result with simulations in SI Fig. S2 (bottom row).

Although in our experiments, we did not have the ability to directly intervene on SSVEP power and set it to arbitrary values, we followed an indirect approach to achieve interventional control. We triggered stimulus presentation when Φ (Paradigm A) or $\Delta\Phi$ (Paradigm B) reached specific, predetermined high or low values. Because of the causal relationship between the source

process A (e.g. attention), and neural processes (M, N, Φ) (SI Fig. S2, top row), fixing Φ to a particular value does not guarantee that A is also fixed to a specific value. Consequently, other neural processes, like M and N, are free to vary even if Φ is at a fixed value at the time of grating presentation. In this manner, our experiment examines the effect of decoupling fluctuations in Φ from fluctuations in the other neural processes on behavioral metrics. Based on this “intervention”, our results support the d’ effect model -- changes in Φ produce obligatory changes in d’, but not in neural process N and, consequently, not in RT.”

Revised Results

“Such an “interventional” cBMI provides two distinct advantages over correlational *post hoc* analysis approaches. First, the interventional cBMI enables decoupling and isolating SSVEP effects on behavioral metrics – accuracy (d’) and reaction times (RT) – from those of other neural processes. We examine 4 directed, graphical models, each of which depicts a distinct mechanism for the influence of SSVEP power (or Φ) on the behavioral metrics of d’ and RT (SI Fig. S2). If a change in Φ were sufficient to induce a change in a behavioral metric (d’ or RT), we would expect to necessarily observe a change in the respective behavioral metric across high- Φ and low- Φ trials. The specific behavioral metric affected, then, enables us to distinguish between the 4 models; further details are provided in the SI (section on Modeling interventional versus correlational approaches; SI Figure S2).”

• *line 654: What happened to the noisy electrodes?*

Clarified as follows:

“... noisy electrodes, thus identified, were removed from further analysis.”

• *line 706: What is meant by clockwise and counterclockwise orientations. I only know this term from describing motion directions*

Clarified as follows:

“In our experiment, participants had to report whether the target grating, on each trial, was oriented clockwise (rightward tilt) or counterclockwise (leftward tilt), relative to the vertical meridian.”

• *line 764: participants with poor SNR are still in the dataset?*

No, these were removed. Please refer SI Fig S1D; also the description in the Methods (lines 767-770).

• *line 926: How was this analysis window derived?*

This analysis has now been revised. Please see Fig. 2A and associated results (lines 1006-1010, revised manuscript).

• *line 1001: What does "challenging" mean?*

Clarified.

“we observed that triggering stimuli based on extreme values of $\Delta\Phi$ was more challenging than that based on Φ alone. For example, mean trigger times for Paradigm A were significantly lesser than that for Paradigm B (Paradigm A: 2754 +/- 263 ms; Paradigm B: 3410 +/- 383 ms,

p<0.001, Mann Whitney U test) despite a comparable percentage of trials being triggered across both paradigms (Paradigm A: 68.49 +/- 14.38%; Paradigm B: 76.07 +/- 12.98%, p=0.1, Mann Whitney U test).”

- *line 1050: What does “had to be relaxed” mean?*

The word “relaxed” has been replaced with “reduced”, for clarity (details in SI Fig. S4D).

- *SI Figure S2 D: why are there only 8 green data points instead of 10?*

Some points in the figure panel were largely overlapping. This SI figure panel is now removed, due to significant changes to main Figure 3.

Reviewers' comments:

Reviewer #1 (Remarks to the Author):

The authors have thoroughly addressed all the issues I raised. I have no further issues or comments.

Reviewer #2 (Remarks to the Author):

Tracking momentary fluctuations in human attention with an "interventional" cognitive brain-machine interface

The authors have made big efforts in incorporating the suggestions made in the previous reviews. This is much appreciated. I overall feel that many issues have been resolved and the story of the experiment is more clear-cut.

I'm, however, still quite hesitant with the overall low number of participants enrolled in the study even more cut down for some specific analyses (the transparent indication of the number of participants contributing to each analysis is much appreciated). While the presentation of the BFs helps to evaluate whether the data is more likely to support the null or the alternative hypothesis, the BF estimation still depends on the effect size and the sample size.

Given the low number of subjects contributing to the test (the hypotheses tested are overall quite interesting), the generalizability and potential robustness of the findings remain hard to judge and may thus require additional measurements of participants.

In addition, the BFs provided and their interpretation often does not match with the claim: line 250: for the reaction times across high- Φ and low- Φ trials no significant differences are reported, yet the BF = 0.9, is rather inconclusive.

Similarly in line 297 and onwards: a BF of 1.36 again is rather inconclusive.

In the same paragraph, a correlation coefficient for $N = 4$ is reported, if I understood the procedure correctly (as described here: Trial-wise analysis of the correlation between target- Φ and distractor- Φ). This sample size does, however, not allow for a reliable calculation of correlation coefficients (see Button et al., 2013, Nature Reviews Neuroscience; see also Yarkoni, 2009, Perspectives in Psychological Science).

Given the low power, potential effects, now depicted as a trend (see figure 3 B top left, $p = .051$) may simply be undetected.

I don't understand why for the trial-wise analysis of correlation ... (line 1012 onwards) the Φ -values were calculated for different time windows and also plotted for these time windows [supp. fig S3] if the correlation is based on one data point for each subject? And how can the SSVEP amplitude be evaluated with a 31.25 ms long time window if one cycle of a 13 Hz SSVEP signal is around 77 ms long?

One of the main findings of the experiments may require a more thorough discussion: What is the reason, why experiment 1 suggests local relevance of SSVEP while experiment 2 suggests modulation of signals across both hemispheres? Does this suggest different mechanisms: in study 1 only sensory gain modulation is seen for spatially selected stimuli irrespective of a modulation of the spatially deselected stimulus, affecting behavior. In study 2, it's the difference between the spatially selected and deselected stimulus that seems to be behaviorally relevant. Are participants in study 2 recruiting a suppressive mechanism in addition to the sensory gain modulation or are participants implementing a different mechanism for selecting task-relevant stimuli in study 2 as compared to study 1? Maybe, the authors could extend the discussion on the potential differences between both studies and the potential implications for attentional/selection mechanisms at hand.

I found the way the authors used the term global competition sometimes misleading. The term competition, as it is used in attention frameworks such as the Biased Competition account, refers to the fact that stimuli, falling for instance into a single receptive field of a neuron/neuron population, compete for processing resources/neural representation/behavioral selectin/ etc. A consequence of this is the mutual suppression of their neural response by competition. Now, attention helps in selecting one of the stimuli by biasing the competition in favor of the selected stimulus, leading to a reduced competitive suppression for the attended/selected stimulus.

My feeling is that the authors in this manuscript use the term global competition when they refer to Biased competition. As in line 306 for instance, global competition would lead to a suppression of the neural representation of the stimuli in both hemispheres. Attention in a second step would resolve this competition by releasing the suppression for the selected stimulus in the sense of Biased Competition. In this vein, as phrased in line 611 onwards, attention does not "create competition" but resolves it. The phrasing and correct use of global competition should be checked throughout the manuscript and adapted where appropriate to be in line with the general idea of the Biased Competition framework.

Minors:

Figure S2 I like the figure illustrating the potential relationships. Maybe for grasping it more easily, the columns of the RT and d' scatter plots (for A, B, C, D) could be swapped (d' left and RT right) so that they are right underneath the respective d' and RT fields of the schematics.

line 217, typo: compared that \diamond compared to that
line 842, maybe better "to be sufficiently apart from"

#####

Button, K. S., Ioannidis, J. P. A., Mokrysz, C., Nosek, B. A., Flint, J., Robinson, E. S. J., et al. (2013). Power failure: why small sample size undermines the reliability of neuroscience. *Nat. Rev. Neurosci.* 14, 365–376. doi:10.1038/nrn3475.

Yarkoni, T. (2009). Big Correlations in Little Studies: Inflated fMRI Correlations Reflect Low Statistical Power-Commentary on Vul et al. (2009). *Perspect. Psychol. Sci.* 4, 294–298. doi:10.1111/j.1745-6924.2009.01127.x.

Reviewers' comments:

Reviewer #1 (Remarks to the Author):

The authors have thoroughly addressed all the issues I raised. I have no further issues or comments.

Thank you.

Reviewer #2 (Remarks to the Author):

Tracking momentary fluctuations in human attention with an “interventional” cognitive brain-machine interface

The authors have made big efforts in incorporating the suggestions made in the previous reviews. This is much appreciated. I overall feel that many issues have been resolved and the story of the experiment is more clear-cut.

Thank you.

I'm, however, still quite hesitant with the overall low number of participants enrolled in the study even more cut down for some specific analyses (the transparent indication of the number of participants contributing to each analysis is much appreciated). While the presentation of the BFs helps to evaluate whether the data is more likely to support the null or the alternative hypothesis, the BF estimation still depends on the effect size and the sample size.

We appreciate the reviewer's viewpoint.

Unlike conventional EEG studies that correlate electrophysiological markers with behavioral parameters, *post hoc*, our real-time cBMI experiments are considerably more challenging to conduct and require significant time for participants to complete (total ~5-6 hours per participant). In order to successfully complete each trial, participants had to wait for SSVEP power states to reach extreme values (Paradigm A) or had to actively increase the difference in SSVEP power across hemifields with neurofeedback (Paradigm B). Despite these challenges, we were able to test a total of $n=24$ participants for these experiments.

The four key findings of this manuscript are well supported with high numbers of participants, as well as with robust Bayes Factors.

- i) Substantial evidence for ($BF>3$) a systematic relationship between target Φ (instantaneous SSVEP power) and target d' (Paradigm A, Fig. 2, $n=15$ participants, **BF=3.36**)
- ii) Substantial evidence against ($BF<0.3$) a systematic relationship between target Φ and distractor d' (Paradigm A, Fig. 2, $n=10$ participants, **BF=0.17**)
- iii) Strong evidence for ($BF>10$) a systematic relationship between $\Delta\Phi$ (difference in SSVEP power across hemifields) and target d' (Paradigm B, Fig. 4, $n=9$ participants, **BF=18.28**)
- iv) Substantial evidence for ($BF>3$) a systematic relationship between $\Delta\Phi$ dynamics and $\Delta d'$ (d' differences between the target and distractor locations) (Paradigm B, Fig. 5, $n=9$ participants, **BF=5.52**)

It is these findings that are highlighted in the Abstract as the key contributions of the study. Moreover, our participant numbers are typical of well-cited EEG studies of this type - Morgan, S. T., et. al., 1996 ($n=12$); Störmer, V. S., et. al., 2013 ($n=15$); Störmer, V. S., et. al., 2014 ($n=12$); Kelly, S. P., et. al., 2005 ($n=10$); Waldhauser, G. T., et. al., 2012 ($n=18$); and Davidson, M. J., et. al., 2020 ($n=19$).

Given the low number of subjects contributing to the test (the hypotheses tested are overall quite interesting), the generalizability and potential robustness of the findings remain hard to judge and may thus require additional measurements of participants.

Figure 3 is the only main figure with a very low number of participants (n=5 for target-side triggered trials and n=4 for distractor-side triggered trials). This analysis was intended merely to form a “bridge” between Paradigm A and Paradigm B -- to motivate the need for the latter, “neurofeedback” paradigm.

The first key conclusion from this figure – that target and distractor SSVEPs were uncorrelated across hemifields – was based on several hundred datapoints (see next response for clarification on Methods). As such, this is a robust result that is consistent with previous findings from literature (Walter, S., et al., 2014; Störmer, V. S., et al., 2014).

The second key finding – that there was no significant variation of target d' with $\Delta\Phi$ – is supported not only by the ANOVA ($p=0.36$), but also by the fact that the variation of d' with $\Delta\Phi$ did not yield even a trend toward significance for three out of the four *post hoc* tests ($p=0.775$, 0.638 , 0.916 ; Fig 3B). The fourth *post hoc* test (upper left panel of Fig. 3B) yielded $p=0.051$, which did not evidence a trend, once we perform multiple comparisons correction. We have now indicated the Benjamini-Hochberg multiple comparisons corrected p-values in the figure, to avoid the potential for confusion.

Figure 3

We have rephrased the conclusions from this section as follows:

“In other words, modulations of target SSVEP power were not accompanied by congruent modulations of distractor SSVEP power in the opposite hemifield. Moreover, once local SSVEP power was fixed at specific (threshold) values, there was no evidence for a significant change in d' with global differences in SSVEP power across the visual hemifields. In other words, sensory processing for the stimulus in the attended hemifield appears to be, at least, in part, independent of processes occurring in the opposite hemifield, a result consistent with earlier studies^{34,35}.”

If the reviewer feels that removing Figure 3 (or moving it to the Supplementary Information), in order to focus on the most robust effects, would help improve the quality of the manuscript, we are willing to accept this suggestion.

In addition, the BFs provided and their interpretation often does not match with the claim: line 250: for the reaction times across high- Φ and low- Φ trials no significant differences are reported, yet the $BF = 0.9$, is rather inconclusive.

Similarly in line 297 and onwards: a BF of 1.36 again is rather inconclusive.

Based on the reviewer's suggestion we have amended these claims as follows:

“Moreover, we observed no evidence for or against significant differences in reaction times across high- Φ and low- Φ trials (Fig. 2B, right; RT: ANOVA, main effect, high- Φ versus low- Φ , $F(1,48)=0.08$, $p=0.772$; $BF=0.96$).”

“Based on this analysis, we observed no evidence for or against systematic modulation of mean Φ on the distractor side across the target high- Φ and low- Φ trials (distractor Φ value: high- Φ trials = 0.53 ± 0.06 , low- Φ trials = 0.51 ± 0.07 , $p=0.125$, Wilcoxon signed-rank test; $BF=1.36$).”

But please note that in Paradigm B, the Bayes Factor revealed substantial evidence against differences in reaction times between the target (high $\Delta\Phi$) and distractor (low $\Delta\Phi$) probed trials (Fig. 4E, right, RT: target = 976 ± 84 ms, distractor = 970 ± 82 ms, $p=0.335$, **$BF=0.17$**).

In the same paragraph, a correlation coefficient for $N = 4$ is reported, if I understood the procedure correctly (as described here: Trial-wise analysis of the correlation between target- Φ and distractor- Φ). This sample size does, however, not allow for a reliable calculation of correlation coefficients (see Button et al., 2013, Nature Reviews Neuroscience; see also Yarkoni, 2009, Perspectives in Psychological Science).

We believe there is a misunderstanding here. We have clarified the analysis steps, below, as well as in the Methods of the revised manuscript.

I don't understand why for the trial-wise analysis of correlation ... (line 1012 onwards) the Φ - values were calculated for different time windows and also plotted for these time windows [supp. fig S3] if the correlation is based on one data point for each subject? And how can the SSVEP

amplitude be evaluated with a 31.25 ms long time window if one cycle of a 13 Hz SSVEP signal is around 77 ms long?

We regret the lack of clarity. First, we estimated target and distractor Φ on each trial in 5 moving windows of 500 ms duration each. The centers of successive windows were shifted by 62.5 ms (8 samples at $fs=128Hz$) so that the first of the five windows was centered at 250 ms before stimulus onset. Then the target- Φ and distractor- Φ values were z-scored, trial-wise. Following this, these values were pooled across participants ($n=4$) and trials ($n>25$ per participant). This resulted in a total of 695 datapoints for low- Φ (525 datapoints for high- Φ) over which the Pearson correlation was computed. Due to the large number of data points, for clarity of visualization, in SI Fig. S3C-D (Paradigm A) we trial-averaged the target and distractor- Φ values for each time window and separately for each participant (different symbols in the figure). The same procedure was followed for analysis of Paradigm B data also (SI Fig. S4B).

We have clarified this in the Methods section as follows:

“Trial-wise analysis of correlation between target- Φ and distractor- Φ . We computed trial-wise correlations between target- Φ and distractor- Φ . For this analysis, we considered target and distractor- Φ values computed in 4 windows of 250 ms duration centered at uniformly spaced time points ranging from 0-1000 ms prior to stimulus triggering (duration: ± 125 ms or ± 16 samples at $fs=128Hz$). For statistical analyses, the Pearson correlation coefficient was computed between the z-scored (trial-wise) target- Φ and distractor- Φ values, pooled over all trials and participants. Since the trial-wise correlation analysis has a large number of data points (i.e., >400 data points, 4 points per trial and >25 trials per participant), only the trial-averaged target and distractor- Φ values were plotted for these time windows across participants in SI Fig. S3C, for clarity of visualization.”

Given the low power, potential effects, now depicted as a trend (see figure 3 B top left, $p = .051$) may simply be undetected.

As clarified above, the finding that there was no significant variation of target d' with $\Delta\Phi$ is supported not only by the ANOVA ($p=0.36$), but also by the fact that the variation of d' with $\Delta\Phi$ did not yield even a trend toward significance for three out of the four *post hoc* tests ($p=0.775$, 0.638 , 0.916 ; Fig 3B). The fourth *post hoc* test (upper left panel of Fig. 3B) yielded $p=0.051$, which did not evidence a trend, once we perform multiple comparisons correction. We have now indicated the Benjamini-Hochberg multiple comparisons corrected p-values in the figure, to avoid confusion.

One of the main findings of the experiments may require a more thorough discussion: What is the reason, why experiment 1 suggests local relevance of SSVEP while experiment 2 suggests modulation of signals across both hemispheres? Does this suggest different mechanisms: in study 1 only sensory gain modulation is seen for spatially selected stimuli irrespective of a modulation of the spatially deselected stimulus, affecting behavior. In study 2, it's the difference between the spatially selected and deselected stimulus that seems to be behaviorally relevant. Are participants in study 2 recruiting a suppressive mechanism in addition to the sensory gain modulation or are participants

implementing a different mechanism for selecting task-relevant stimuli in study 2 as compared to study 1? Maybe, the authors could extend the discussion on the potential differences between both studies and the potential implications for attentional/selection mechanisms at hand.

In the revised Discussion, we discuss the related, but distinct, findings from the two task paradigms. Specifically, we hypothesize that the spatial competition and global selection in Paradigm B likely arises, as a special case, due to neurofeedback in this paradigm.

“... , previous literature suggests that attentional resources across the left and right visual hemifields are separate and, potentially, independent; this dissociation has been reported in many studies including human psychophysics^{31,33}, human electrophysiology^{34,35} and non-human primate electrophysiology⁸. In line with these findings, in Paradigm A, target discrimination d' was not systematically modulated either by distractor SSVEP power states ($\Phi_{\text{distractor}}$), or by the global difference of SSVEP power across hemifields ($\Delta\Phi$), suggesting that SSVEP power modulations may occur independently across brain hemispheres, by default, and independently affect performance across the hemifields during cued attention tasks. Yet, in Paradigm B, when subjects were trained using neurofeedback, to increase the difference of SSVEP power across hemifields ($\Delta\Phi$) – and stimuli were triggered based on this neural signature of stimulus competition^{32,65} – we observed robust, differential modulations of d' across the hemifields, with Bayes Factor values around 18.0.

We propose the following hypothesis that reconciles our findings with those in previous literature. SSVEP power is modulated by a combination of processes: These include both local selection processes that operate independently across visual hemifields – by engaging dissociable neural processing resources in each brain hemisphere – as well as global selection mechanisms that involve biased competition for neural resources between hemispheres. Under normal conditions, the former set of processes (local selection) dominate. But, when subjects were explicitly provided with neurofeedback based on the difference in SSVEP power across hemifields, global selection processes became strongly engaged. In other words, our neurofeedback paradigm engendered global, spatial selection by inducing biased competition for neural resources across brain hemispheres, and thereby produced robust effects on behavioral accuracies. Whether such a neurofeedback cBMI can be employed to train subjects to produce behavioral effects that are considerably stronger than those reported in standard attention tasks, remains to be explored.”

I found the way the authors used the term global competition sometimes misleading. The term competition, as it is used in attention frameworks such as the Biased Competition account, refers to the fact that stimuli, falling for instance into a single receptive field of a neuron/neuron population, compete for processing resources/neural representation/behavioral selectin/ etc. A consequence of this is the mutual suppression of their neural response by competition. Now, attention helps in selecting one of the stimuli by biasing the competition in favor of the selected stimulus, leading to a reduced competitive suppression for the attended/selected stimulus. My feeling is that the authors in this manuscript use the term global competition when they refer to Biased competition.

This is an interesting idea. As the reviewer correctly indicates, the term biased competition – especially in spatial attention tasks – has been generally reserved for scenarios when multiple stimuli appear in close proximity (e.g. within a neuron’s RF). In this case, attention biases the competition in favor of the selected stimulus and enhances neural responses, as long as the stimulus in the RF exhibits a “preferred” feature. By contrast attention suppresses neural

responses for stimuli with “non-preferred” features (Feature Similarity Gain Principle, Martinez-Trujillo, J. C., & Treue, S., 2004).

In this study, our stimuli appeared far apart, in distinct visual hemifields. Moreover, we did not analyze feature tuning in the SSVEP responses (e.g. orientation preferences) and did not measure the effect of attention on preferred versus non-preferred stimuli. We, hence, used the term “global competition”, as distinct from the conventional use of biased competition (but see next response).

As in line 306 for instance, global competition would lead to a suppression of the neural representation of the stimuli in both hemispheres. Attention in a second step would resolve this competition by releasing the suppression for the selected stimulus in the sense of Biased Competition. In this vein, as phrased in line 611 onwards, attention does not “create competition” but resolves it. The phrasing and correct use of global competition should be checked throughout the manuscript and adapted where appropriate to be in line with the general idea of the Biased Competition framework.

Thank you for the suggestion. We have replaced the term “global competition” with the term “biased competition” or “global selection”, to refer to a selection mechanism that operates across the visual field in Paradigm B. We reproduce below a select few sentences that have been revised; the full set of revisions are available in the red-lined document.

Abstract:

“Next, we trained participants on an auditory neurofeedback paradigm to generate **biased, cross-hemispheric competitive interactions** between target and distractor SSVEPs”

Results:

“Our findings, so far, suggest that attentional resources in the two hemifield are, at least partly, independent. Yet, previous behavioral studies suggest that the behavioral effect of spatial attention – in particular, those on d' enhancement – are mediated by **biasing competitive selection** for sensory resources, globally, across the visual field^{1,36–39}.”

“Second, we tested if target discrimination accuracy would vary with the strength of **biased, global competition** between target and distractor representations, at a fixed level of target (or distractor) SSVEP power.”

“In other words, **biasing global competitive interactions** among stimulus representations across the visual hemifields provided a reliable marker for behavioral effects on discrimination accuracy, but not on reaction times.”

Discussion:

“Second, it is well known that – at least in some settings – attention can **bias competition** for resources across the visual hemifields^{37,64,65}. We, therefore, tested the hypothesis that **generating biased competition** for neural representations across the visual hemifields would also produce attention-like effects on perceptual accuracy.”

In addition, we have now clarified the wording to indicate that, while $\Delta\Phi$ reflects interhemispheric competition between visual representations in the two hemispheres, the behavioral effect on $\Delta d'$ (higher d' at the target location) induced by extreme values of $\Delta\Phi$ is consistent with the biased competition model of attention.

“We propose the following hypothesis that reconciles our findings with those in previous literature. SSVEP power is modulated by a combination of processes: These include both local selection processes that operate independently across visual hemifields – by engaging dissociable neural processing resources in each brain hemisphere – as well as global selection mechanisms that involve biased competition for neural resources between hemispheres. Under normal conditions, the former set of processes (local selection) dominate. But, when subjects were explicitly provided with neurofeedback based on the difference in SSVEP power across hemifields, global selection processes became strongly engaged. In other words, our neurofeedback paradigm engendered global, spatial selection by inducing biased competition for neural resources across brain hemispheres, and thereby produced robust effects on behavioral accuracies. Whether such a neurofeedback cBMI can be employed to train subjects to produce behavioral effects that are considerably stronger than those reported in standard attention tasks, remains to be explored.”

Minors:

Figure S2 I like the figure illustrating the potential relationships. Maybe for grasping it more easily, the columns of the RT and d' scatter plots (for A, B, C, D) could be swapped (d' left and RT right) so that they are right underneath the respective d' and RT fields of the schematics.

The figure is now reorganized, as suggested by the reviewer.

line 217, typo: compared that \diamond compared to that

Amended.

line 842, maybe better “to be sufficiently apart from”

Modified.

References:

Morgan, S. T., Hansen, J. C., & Hillyard, S. A. (1996). Selective attention to stimulus location modulates the steady-state visual evoked potential. *Proceedings of the National Academy of Sciences*, 93(10), 4770-4774.

Störmer, V. S., Alvarez, G. A., & Cavanagh, P. (2014). Within-hemifield competition in early visual areas limits the ability to track multiple objects with attention. *Journal of Neuroscience*, 34(35), 11526-11533.

Störmer, V. S., Winther, G. N., Li, S. C., & Andersen, S. K. (2013). Sustained multifocal attentional enhancement of stimulus processing in early visual areas predicts tracking performance. *Journal of Neuroscience*, 33(12), 5346-5351.

Kelly, S. P., Lalor, E. C., Reilly, R. B., & Foxe, J. J. (2005). Visual spatial attention tracking using high-density SSVEP data for independent brain-computer communication. *IEEE Transactions on Neural Systems and Rehabilitation Engineering*, 13(2), 172-178.

Waldhauser, G. T., Johansson, M., & Hanslmayr, S. (2012). Alpha/beta oscillations indicate inhibition of interfering visual memories. *Journal of Neuroscience*, 32(6), 1953-1961.

Davidson, M. J., Mithen, W., Hogendoorn, H., Van Boxtel, J. J., & Tsuchiya, N. (2020). The SSVEP tracks attention, not consciousness, during perceptual filling-in. *Elife*, 9, e60031.

Walter, S., Quigley, C., & Mueller, M. M. (2014). Competitive interactions of attentional resources in early visual cortex during sustained visuospatial attention within or between visual hemifields: Evidence for the different-hemifield advantage. *Journal of cognitive neuroscience*, 26(5), 938-954.

Martinez-Trujillo, J. C., & Treue, S. (2004). Feature-based attention increases the selectivity of population responses in primate visual cortex. *Current biology*, 14(9), 744-751.

REVIEWERS' COMMENTS:

Reviewer #2 (Remarks to the Author):

The authors have thoroughly addressed all the issues raised and motivated and explained their approaches. Thank you for your work. I feel the central message of the manuscript is more clear-cut now. There are no further issues on my behalf.